# Back to the Continuous Attractor

**Ábel Ságodi, Guillermo Martín-Sánchez, Piotr Sokół, Il Memming Park**

Champalimaud Centre for the Unknown
Champalimaud Foundation, Lisbon, Portugal
`{abel.sagodi,guillermo.martin,memming.park}@research.fchampalimaud.org`
`piotr.sokol@protonmail.com`

## Abstract

Continuous attractors offer a unique class of solutions for storing continuous-valued variables in recurrent system states for indefinitely long time intervals. Unfortunately, continuous attractors suffer from severe structural instability in general—they are destroyed by most infinitesimal changes of the dynamical law that defines them. This fragility limits their utility especially in biological systems as their recurrent dynamics are subject to constant perturbations. We observe that the bifurcations from continuous attractors in theoretical neuroscience models display various structurally stable forms. Although their asymptotic behaviors to maintain memory are categorically distinct, their finite-time behaviors are similar. We build on the persistent manifold theory to explain the commonalities between bifurcations from and approximations of continuous attractors. Fast-slow decomposition analysis uncovers the existence of a persistent slow manifold that survives the seemingly destructive bifurcation, relating the flow within the manifold to the size of the perturbation. Moreover, this allows the bounding of the memory error of these approximations of continuous attractors. Finally, we train recurrent neural networks on analog memory tasks to support the appearance of these systems as solutions and their generalization capabilities. Therefore, we conclude that continuous attractors are *functionally robust* and remain useful as a universal analogy for understanding analog memory.

## 1 Introduction

Biological systems exhibit robust behaviors that require neural information processing of analog variables such as intensity, direction, and distance. Virtually all neural models of working memory for continuous-valued information rely on persistent internal representations through recurrent dynamics. The continuous attractor structure in their recurrent dynamics has been a pivotal theoretical tool due to their ability to maintain activity patterns indefinitely through neural population states[1–4]. They are hypothesized to be the neural mechanism for the maintenance of eye positions, heading direction, self-location, target location, sensory evidence, working memory, and decision variables, to name a few[5–7]. Observations of persistent neural activity across many brain areas, organisms, and tasks have corroborated the existence of continuous attractors[7–15].

Despite their widespread adoption as models of analog memory, continuous attractors are brittle mathematical objects, casting significant doubts on their ontological value and hence suitability in accurately representing biological functions. Even the smallest arbitrary change in recurrent dynamics can be problematic, destroying the continuum of fixed points essential for continuous-valued working memory. In neuroscience, this vulnerability is well-known and often referred to as the "fine-tuning problem"[5,16–20]. There are two primary sources of perturbations in the recurrent network dynamics: (1) the stochastic nature of online learning signals that act via synaptic plasticity, and (2) spontaneous fluctuations in synaptic weights[21,22]. Thus, additional mechanisms are necessary to compensate for the degradation in particular implementations, by bringing the short-term behavior closer to that

of a continuous attractor [16,23–29]. However, we lack the theoretical basis to understand how much this matters in practice, i.e. what are the effects of different levels of degradation on memory. This is fundamental to justify relying on the brittle concept of continuous attractors for understanding biological analog working memory.

In this study, we explore perturbations and approximations of continuous attractors in the space of dynamical models. We first report on the differences and similarities between the various structurally stable dynamics in the vicinity of continuous attractors in the space of dynamical systems models. Our analysis reveals the presence of a "ghost" continuous attractor (a.k.a. slow manifold) in all of them (Sec. 2). By assuming normal hyperbolicity we separate the time scales to obtain a decomposition of the dynamics by separating out the fast flow normal to and the slow flow within the slow manifold. We derive theoretical results that ensure the existence of a slow manifold and determine its closeness to a continuous attractor (Sec. 3). We explore task-trained recurrent neural networks (RNNs) to show that these systems appear naturally as solutions to the task (Sec. 4) and that their generalization capabilities can easily be studied as the distance to the continuous attractor (Sec. 5). The proposed decomposition applied to theoretical models and task-trained RNNs reveals a "universal motif" of analog memory mechanism with various potential topologies. This leads to the connection of different systems with different topologies as **approximate continuous attractors** (Sec. 6). Our theory guarantees that systems close to a continuous attractor (in the space of vector fields) will have similar behavior to it, implying that the concept of continuous attractors remains a crucial framework for understanding the neural computation underlying analog memory (Sec. 3.4).

## 2 A critique of pure continuous attractors

We will first lay out a number of observations about the dynamics of bifurcations and approximations of continuous attractors used in theoretical neuroscience. Ordinary differential equations (ODEs) are commonly used to describe the dynamical laws governing the temporal evolution of firing rates or latent population states [1]. In this framework, neural systems are viewed as implementing the continuous time evolution of neural states to perform computations. We will consider a continuous attractor as a mechanism that implements analog memory computation: carrying a particular memory representation over time. To define it formally, let $\mathbf{x}(t) \in \mathbb{R}^d$ denote the neural state, and $\dot{\mathbf{x}} = \mathbf{f}(\mathbf{x})$ represent its dynamics. Let $\mathcal{M} \subset \mathbb{R}^d$ be a manifold. We say $\mathcal{M}$ is a continuous attractor, if (1) every state on the manifold is a fixed point, $\forall \mathbf{x} \in \mathcal{M}, \mathbf{f}(\mathbf{x}) = 0$, and (2) the fixed points are marginally stable tangent to the manifold and stable normal to the manifold. In other words, the continuous attractor is a continuum of equilibrium points such that the neural state near the manifold is attracted to it, and on the manifold, the state does not move. Marginal stability implies that continuous systems are *structurally unstable*, meaning that small perturbations or variations in the system's parameters lead to significant changes in the system's behavior or stability [30–33]. We will now study some examples of continuous attractors and how perturbations change their dynamics.

### 2.1 Motivating example: bounded line attractor

As an illustrative example, we can construct a line attractor (a continuous attractor with a line manifold) as follows:

$$\dot{\mathbf{x}} = -\mathbf{x} + [\mathbf{W}\mathbf{x} + \mathbf{b}]_+ \tag{1}$$

where $\mathbf{W} = [0, -1; -1, 0]$ and $\mathbf{b} = [1; 1]$, and $[\cdot]_+ = \max(0, \cdot)$ is the threshold nonlinearity per unit. We get $\dot{\mathbf{x}} = 0$ on the $x_1 = -x_2 + 1$ line segment in the first quadrant as the manifold (Fig. 1A, left; black line). Linearization of the fixed points on the manifold exhibits two eigenvalues, $0$ and $-2$; the $0$ eigenvalue allows the continuum of fixed points, while $-2$ makes the flow normal to the manifold attractive (Fig. 1A, left; flow field).

In general, continuous attractors are not only structurally unstable [34], they bifurcate almost certainly for an *arbitrary* perturbation of $\mathbf{f}$. In this example, small changes to the parameters $(\mathbf{b}, \mathbf{W})$ perturb the eigenvalues and any to the $0$ eigenvalue destroys the continuous attractor: it bifurcates to either a single stable fixed point (Fig. 1A, top) or two stable fixed points separated with a saddle-node in between (Fig. 1A, bottom). However, interestingly, after bifurcation, continuous attractors seemingly tend to leave a "ghost" manifold topologically equivalent to the original continuous attractor (note the slow speed). Furthermore, the flow after the bifurcation is contained in the ghost manifold, i.e., it is an invariant manifold. This phenomenon, wherein a continuous attractor is approximated by a manifold within the neural space along which the drift occurs at a very slow pace, has previously been discussed [35–37].

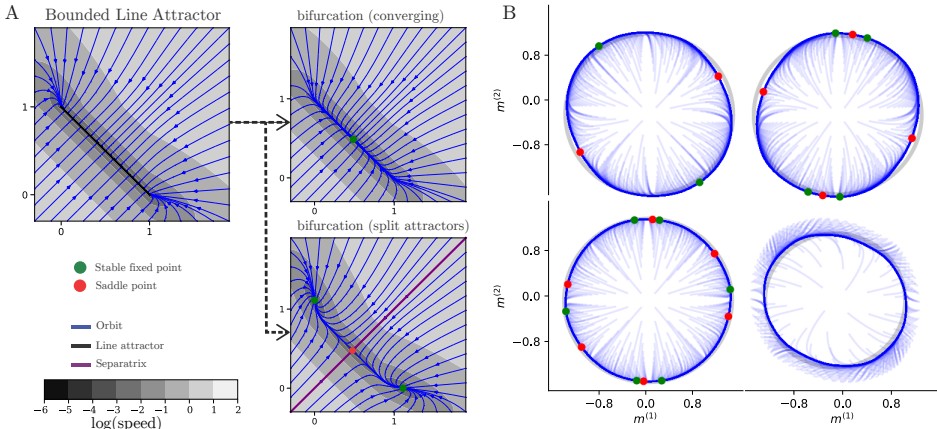

Figure 1: The critical weakness of continuous attractors is their inherent brittleness as they are rare in the parameter space, i.e., infinitesimal changes in parameters destroy the continuous attractor implemented in RNNs[5,16]. Some of the structure seems to remain; there is an invariant manifold that is topologically equivalent to the original continuous attractor. **(A)** Phase portraits for the bounded line attractor (Eq. (1)). Under perturbation of parameters, it bifurcates to systems without the continuous attractor. **(B)** The low-rank ring attractor approximation (Sec. (S3.4)). Different topologies exist for different realizations of a low-rank attractor: different numbers of fixed points (4, 8, 12), or a limit cycle (right bottom). Yet, they all share the existence of a ring invariant set.

## 2.2 Theoretical models of ring attractors

For circular variables such as the goal-direction (e.g. for navigation[38,39] and working memory for communication in bees[40]) or head direction, the temporal integration, and working memory functions are naturally solved by a ring attractor (continuous attractor with a ring topology)[41?–49]. Other examples include integration of evidence for continuous perceptual judgments, e.g. a hunting predator that needs to compute the net direction of motion of a large group of prey[50]. In this section, we investigate the bifurcations of various implementations of continuous attractors. Continuous attractor network models of the head direction are based on the interactions of neurons that (anatomically) form a ring-like overlapping neighbor connectivity[8,51–57]. Similarly to the line attractor, the ring attractor bifurcates with almost any perturbation to the network dynamics. However, the resulting dynamics continue to follow a familiar pattern: they remain confined to a ghost manifold that closely approximates the original continuous attractor.

**Piecewise-linear ring attractor model of the central complex:** Firstly, we discuss perturbations of a continuous ring attractor recently proposed as a model for the head direction representation in fruit flies[8]. This model is composed of $N$ heading-tuned neurons with preferred headings $\theta_j \in \{2\pi i/N\}_{i=1...N}$ radians (Sec. S3.2). For sufficiently strong local excitation (given by the parameter $J_E$) and broad inhibition ($J_I$), this network will generate a stable bump of activity corresponding to the head direction. This continuum of fixed points forms a $N$-sided polygon.

We evaluate the effect of parametric perturbations of the form $\mathbf{W} \leftarrow \mathbf{W} + \mathbf{V}$ with $\mathbf{V}_{i,j} \overset{iid}{\sim} \mathcal{N}(0, 1/100)$ on a network of size $N = 6$ (forming an hexagon, see also Sec. S3). We found that the continuum of fixed points can collapse to between 2 and 12 isolated fixed points (Fig. 2A). As far as we know, this bifurcation from a ring of equilibria to a saddle and node has not been described previously in the literature. The probability of each type of bifurcation was numerically estimated (Sec. S3.2). Surprisingly, the number of fixed points is maintained throughout a range of perturbation sizes and hence depends only on the direction of the perturbation (Fig. S8).

**Bump attractor model:** A well-established approach to form a ring attractor in the limit of large network is with a connection matrix $W$ with entries that follow a circular Gaussian function of $i - j$[58,60–62] (Sec. S3.3). This type of ring attractor network can support a stable "activity bump" that can move around the ring of nonlinear neurons in correspondence with changes in head direction[63]. For finite-sized networks, the dynamics are constrained to an attractive invariant ring, covered with $N$ stable fixed points for a network of size $N$ (Fig. 2B). For such networks the number of fixed points can change with the size of the perturbation (Fig. S8).

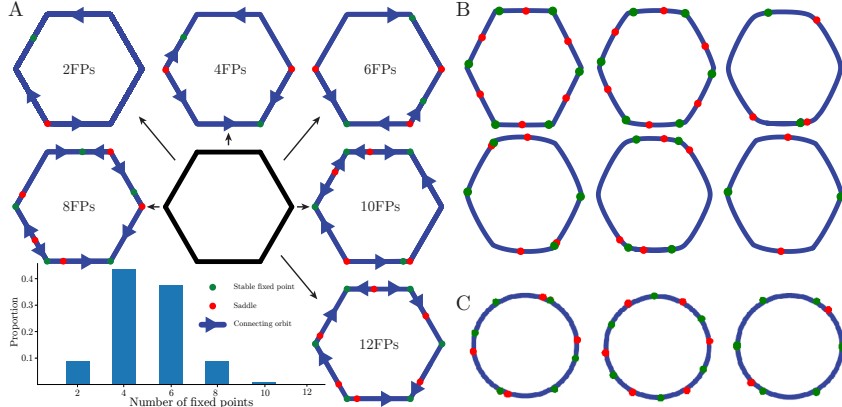

Figure 2: Perturbations of different implementations and approximations of ring attractors lead to bifurcations that all leave the ring invariant manifold intact. For each model, the network dynamics is constrained to a ring manifold with stable fixed points (green) and saddle nodes (red). (A) Perturbations to Noorman et al.[8]. The ring attractor can be perturbed in systems with an even number of fixed points (FPs) up to $2N$ (stable and saddle points are paired). (B) Perturbations to a tanh approximation of a ring attractor Seeholzer et al.[58]. (C) Different Embedding Manifolds with Population-level Jacobians (EMPJ) approximations of a ring attractor[59].

**Low-rank ring attractor model:** Low-rank networks can be used to approximate ring attractors[64,65]. In the limit of infinite-size networks, one can construct a ring attractor through a rank 2 network by constraining the overlap of the right- and left-connectivity vectors (see Sec. S3.4). However, in simulations of finite-size networks with this constraint, the dynamics instead always converge to a small number of stable fixed points arranged on a ring (Fig. 1B).

**Embedding Manifolds with Population-level Jacobians:** Approximate ring attractors can be constructed by constraining the connectivity so that the networks Jacobian satisfies certain requirements for a ring attractor to exist[59] (see Sec. S3.5). The models constructed with this method also can contain an invariant ring manifold on which the dynamics contain stable and saddle fixed points (Fig. 2C). It has been observed that approximate continuous attractor emerge in networks trained on sampled points with other methods as well[66].

**Similarity between all bifurcations and approximations of continuous attractors:** In all discussed models of ring attractors, we verify that they suffer from the fine-tuning problem. However, importantly, we also observe in all the systems the existence of ghosts of the continuous attractor (either through bifurcation or from finite-size effects) in the form of an *attractive invariant manifold*. Therefore, while they are not strictly a continuous attractor in the mathematical sense, they are *approximate* ring attractors in the sense that the fixed points and connecting orbits still form a circle.

Is this lawful degradation a universal phenomenon? And if so, how does it relate to the size of the perturbation? And what are the implications for the memory performance of these approximations? (Sec. 3). Do these approximations appear as natural solutions to the memory storage problem? (Sec. 4). And if so, how well do they generalize to longer time requirements? (Sec. 5). Finally, are continuous attractors in practice still useful as an idealized model of how animals represent continuous variables? (Sec. 6).

## 3 Theory of Approximate Continuous Attractors

In this section, we theoretically answer in an implementation-agnostic manner the degradation questions posed from the exploration. To do so, we apply invariant manifold theory to continuous attractor models and translate the results for the neuroscience audience (see also Sec. S1).

### 3.1 Persistent Manifold Theorem

First, we argue that the lawful degradation into a system with a slow manifold is universally guaranteed (as long as the perturbation is small, and the continuous attractor was normally hyperbolic). Let $l$ be the intrinsic dimension of the manifold of equilibria that defines the continuous attractor. Given a perturbation $\mathbf{p}(\mathbf{x})$ to the ODE that induces a bifurcation,

$$\dot{\mathbf{x}} = \mathbf{f}(\mathbf{x}) + \epsilon\,\mathbf{p}(\mathbf{x}) \tag{2}$$

where $\|\mathbf{p}(\cdot)\|_\infty = 1$ and $\epsilon > 0$ is the bifurcation parameter, we can reparameterize the dynamics around the manifold with coordinates $\mathbf{y} \in \mathbb{R}^l$ and the remaining ambient space with $\mathbf{z} \in \mathbb{R}^{d-l}$. To describe an arbitrary bifurcation of interest, we introduce a sufficiently smooth function $\mathbf{g}$ and $\mathbf{h}$, such that the following system is equivalent to the original ODE:

$$\dot{\mathbf{y}} = \epsilon \mathbf{g}(\mathbf{y}, \mathbf{z}, \epsilon) \qquad \text{(tangent)} \qquad (3)$$

$$\dot{\mathbf{z}} = \mathbf{h}(\mathbf{y}, \mathbf{z}, \epsilon) \qquad \text{(normal)} \qquad (4)$$

where $\epsilon = 0$ gives the condition for the continuous attractor $\dot{\mathbf{y}} = \mathbf{0}$. We denote the corresponding manifold of $l$ dimensions as $\mathcal{M}_0 := \{(\mathbf{y}, \mathbf{z}) \mid \mathbf{h}(\mathbf{y}, \mathbf{z}, 0) = \mathbf{0}\}$.

We say the flow around the manifold is *normally hyperbolic*, if the flow normal to the manifold is hyperbolic, i.e. the Jacobians $\nabla_\mathbf{z}\mathbf{h}$ evaluated at any point on the $\mathcal{M}_0$ has $d - l$ eigenvalues with their real part uniformly bounded away from zero, and $\nabla_\mathbf{y}\mathbf{g}$ has $l$ eigenvalues with zero real part. More specifically, for continuous attractors, the real parts of the eigenvalues of $\nabla_\mathbf{z}\mathbf{h}$ are strictly negative, representing sufficiently strong attractive flow toward the manifold. Equivalently, for the ODE, $\dot{\mathbf{x}} = \mathbf{f}(\mathbf{x})$, the variational system is of constant rank and has exactly $(d - l)$ eigenvalues with negative real parts uniformly away from zero and $l$ eigenvalues with zero real parts everywhere along the continuous attractor.

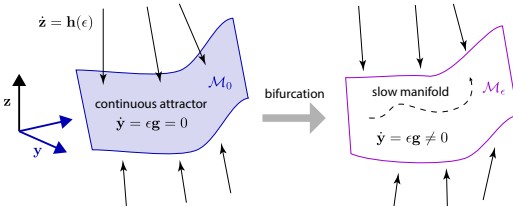

Figure 3: Persistent manifold theorem applied to compact continuous attractor guarantees the flow on the slow manifold $\mathcal{M}_\epsilon$ is invariant and continues to be attractive. The dashed line is a trajectory "trapped" in the slow manifold (which has the same topology as the continuous attractor).

For any parameterization $\mathbf{g}$, $\epsilon > 0$ induces a bifurcation of the continuous attractor. What can we say about the fate of the perturbed system? The continuous dependence theorem[67] says that the trajectories will change continuously as a function of $\epsilon$ without a guarantee on how quickly they change. However, the topological structure and the asymptotic behavior of trajectories change discontinuously due to the bifurcation. Yet, surprisingly, there is a strong connection in the geometry due to Fenichel's theorem[68].* We informally present a special case due to Jones[73]:

**Theorem 1** (Persistent Manifold). *Let $\mathcal{M}_0$ be a connected, compact†, normally hyperbolic manifold of equilibria originating from a sufficiently smooth ODE. For a sufficiently small perturbation $\epsilon > 0$, there exists a manifold $\mathcal{M}_\epsilon$ diffeomorphic to $\mathcal{M}_0$ and invariant under the flow of Eq.* (3)-(4).

The manifold $\mathcal{M}_\epsilon$ is called the *slow manifold* which is no longer necessarily a continuum of equilibria. However, the invariance implies that trajectories remain within the manifold except potentially at the boundary. Furthermore, the non-zero flow on the slow manifold is slow and given in the $\epsilon \to 0$ limit as $\frac{d\mathbf{y}}{d\tau} = \mathbf{g}(c^\epsilon(\mathbf{y}), \mathbf{y}, 0)$ where $\tau = \epsilon t$ is a rescaled time and $c^\epsilon(\cdot)$ parameterizes the $l$ dimensional slow manifold. In addition, the stable manifold of $\mathcal{M}_0$ is similarly persistent[73], implying that the manifold $\mathcal{M}_\epsilon$ remains attractive. Finally, the persisting invariant manifold is very close in space to the original continuous attractor (see also Theorem 3).

These conditions are met for the examples in Fig. 1 (see Sec. S2.1 for the corresponding fast-slow reparametrization‡).As the theory predicts, it bifurcates into a 1-dimensional slow manifold (Fig. 1, dark-colored regions) that contains fixed points and connecting orbits and is overall still attractive. Furthermore, Fenichel's Persistent Manifold theorem explains the bifurcation structure of the theoretical models discussed in Sec. 2.2. Because continuous ring attractors are bounded, they persist as an invariant manifold and remain attractive under small perturbations[78].

---

*The Persistent Manifold Theorem has been successfully applied previously in neuroscience[69,70], for example to reduce the dimensionality of the Hodgkin-Huxley model[71,72].

†See Eldering[74] for results of persistence of noncompact invariant manifolds.

‡As a technical note, for the theory to apply to a continuous piecewise-linear system, it is required that the invariant manifold is globally attracting[75], which is also the case for the BLA (see also[76,77] for a discussion of geometric singular perturbation theory for piecewise linear dynamical systems). Therefore, we consider systems that are at least continuous, but some extra conditions apply if a system is not differentiable.

## 3.2 Fast-slow decomposition and the revival of continuous attractors

Consider a behaviorally relevant timescale for working memory, for example, roughly up to a few tens of seconds. If the relevant dynamical system is orders of magnitude slower, for example, 1000 sec or longer, its effect is too slow to have a practical impact on the behavior. This clear gap in the fast and slow time scales can be recast as *normal hyperbolicity* of the slow manifold by relaxing the zero real part to a separation of time scales (reciprocal of eigenvalues or Lyapunov exponents). In other words, the attractive flow normal to the manifold needs to be uniformly faster than the flow on the slow manifold. By taking the limit of the slow flow on the manifold to arbitrarily long time constant (i.e., to zero flow), we achieve the reversal of the persistent manifold theorem.

**Proposition 1** (Revival of continuous attractor). *Let $\mathcal{M}_\epsilon$ be a connected, compact, attractive, normally hyperbolic slow manifold (as parametrized by Eq.* (3)-(4)*). Let the uniform norm of the flow tangent to the manifold be $\|\dot{\mathbf{y}}\|_\infty = \eta$. There exists a perturbation with uniform norm at most $\eta$ that induces a bifurcation to a continuous attractor manifold.*

An explicit perturbation is derived in Sec. S5.1. This makes the uniform norm of the vector field on a (slow) manifold a useful measure of the distance of an approximation to a continuous attractor. Prop. 1 can be extended to the case where the invariant manifold has additional dynamics to which the output mapping is invariant (see Theorem 7). These systems can be perturbed onto a decomposable system where one of the subsystems has a slow flow.

## 3.3 Relevance of dynamics on the memory performance of the slow manifold

Third, we relate the flow of the manifold (and, through Prop. 1, the *size* of the perturbation) to the memory error of the approximation in short-time scale. We also discuss the implications of the theoretical insights on the memory error in the asymptotic time scale.

In the short-time scale the memory performance is bounded by the uniform norm of the flow tangent to the manifold. Let $\mathbf{x}_0 \in \mathcal{M}$, and $\varphi = \mathbf{p}(\cdot)|_{\mathcal{M}}$ be the vector field restricted to the manifold (following the notation in Eq. 2). The average deviation from the initial memory $\mathbf{x}_0$ over time is bounded linearly (derivation in Sec. S6):

$$\frac{1}{\text{vol}\,\mathcal{M}} \int_{\mathcal{M}} |\mathbf{x}(t, \mathbf{x}_0) - \mathbf{x}_0|\, d\mathbf{x}_0 \le t\,\|\varphi\|_\infty \qquad \text{(error bound)} \qquad (5)$$

Note that this bound is the worst case and tighter for sufficiently small $t \ge 0$. Furthermore, for compact invariant manifolds the error is bounded by the diameter of the manifold and hence this bound becomes irrelevant for $t$ large.

While the uniform norm gives insight on the short-time scale behavior of the perturbed ODE, we expect that working memory tasks generalize to longer durations [19]. The long-time scale behavior on the slow manifold is dominated by the stability structure, i.e., the topology of the dynamics. Although we have seen numerous topologies in Sec. 2, the Persistent Manifold Theorem says that this variability is fundamentally limited, especially in low dimensions (see for more details Sec. S4). This is especially relevant as previous works have identified a low-dimensional organization of neural activity to explain the brain's ability to adapt behavioral responses to changing stimuli and environments [79–81]. For a ring attractor, this implies that the stability structure of the invariant manifold is either (1) composed of an equal number of stable fixed points and saddle nodes, placed alternatingly and with connecting orbits, or (2) a limit cycle. These different stability structures have different generalization properties (see Sec. 5). In more complex scenarios, such as two-dimensional attractors, fixed points can coexist with limit cycles, creating a rich tapestry of possible attractors.

## 3.4 Implications on experimental neuroscience

Animal behavior exhibits strong resilience to changes in their neural dynamics, such as the continuous fluctuations in the synapses or slight variations in neuromodulator levels or temperature. Hence, any theoretical model of neural or cognitive function that requires fine-tuning, such as the continuous attractor model for analog working memory, raises concerns, as they are seemingly biologically irrelevant. This challenge is further compounded by the structural constraints imposed by the connectome, which defines the network's architecture and limits the possible configurations of synaptic and circuit dynamics [82,83]. Moreover, unbiased data-driven models of time series data and task-trained recurrent network models cannot recover such continuous attractor theories precisely. Our theory shows that this apparent fragility is not as devastating as previously thought: despite the "qualitative differences" in the phase portrait, the "effective behavior" of the system can be arbitrarily

close, especially in the behaviorally relevant time scales. We show that as long as the attractive flow to the memory representation manifold is fast and the flow on the manifold is sufficiently slow, it represents an **approximate continuous attractor**[§]. Furthermore, our theory bounds the error in working memory incurred over time for such approximate continuous attractors. Therefore, the concept of continuous attractors remains a crucial framework for understanding the neural computation underlying analog memory, even if the ideal continuous attractor is never observed in practice. Experimental observations that indicate the slowly changing population representations during the "delay periods" where working memory is presumably required, do not necessarily contradict the continuous attractor hypothesis. Perturbative experiments can further measure the attractive nature of the manifold and their causal role through manipulating the memory content.

## 4  Numerical Experiments on Task-optimized Recurrent Networks

While our theory describes the abundance of approximate continuous attractors in the vicinity of a continuous attractor, it does not imply that there are no approximate solutions away from continuous attractors. In this section, we use task-optimized RNNs as a means to search for plausible solutions for analog memory for a circular variable. We train a diverse set of RNNs, and then identify the solution type of trained RNNs to gain insights into its performance, error patterns, generalization capabilities, and, ultimately, proximity to a continuous attractor.

Understanding the implemented computation in neural systems in terms of dynamical systems is a well-established practice[1,5,86]. Researchers have analyzed task-optimized RNNs through non-linear dynamical systems analysis[87,?,-95] and to compare those artificial networks to biological circuits[36,96,97]. Previously, systematic analysis of the variability in network dynamics has been surveyed in vanilla RNNs, and variations in dynamical solutions over architecture and nonlinearity have been quantified[36,87,90,91,98]. Furthermore, working memory mechanisms in RNNs had tendencies to find sequential or persistent representations through training depending on the task specification[99]. We therefore investigated to what extent training RNNs on a task uniquely determines the low-dimensional dynamics, independent of neural architectures. We see that all the solutions have a slow invariant manifold, making all of them an instantiation of approximate continuous attractors.

### 4.1  Model Architectures and Training Procedure

Building upon prior work, which has shown their capabilities on such tasks, we trained RNNs to either (1) estimate head direction through integration of angular velocity[92,93] (Fig. 4A1) or (2) perform a memory-guided saccade task for targets on a circle[100,101] (Fig. 4A2, details in Sec. S7.1 and see Sec. S7.3 for how RNNs relate to Eq. 2). We numerically minimized the mean squared error loss $L_{\mathrm{MSE}}$ between the network output and the target output. For each activation function and each network architecture (vanilla RNN with ReLU, `tanh`, and rectified `tanh` activation functions, LSTM, and GRU), we trained 10 networks per hidden size: 64, 128, and 256 with (hidden) state noise.

### 4.2  Numerical Fast-Slow Decomposition

For each trained network, we find the slow manifold by integrating the autonomous dynamics, then selecting the parts of the trajectories that have speed slower than a threshold (Sec. S7.9.1). We identify the points on the invariant manifold from the simulated trajectories that are projected closest to a set of points in the output space relevant to the task after convergence, i.e. on the target ring. We parametrize the one-dimensional invariant manifold by fitting a cubic spline with periodic boundary constraints to these points (black line in Fig. 4B & C). Normal hyperbolicity is measured by a gap in the timescales of the system (the eigenvalue spectrum of the linearization along points on the invariant manifold, Fig. 4E and F).

We find the fixed points on the invariant ring by identifying regions where the direction of the flow flips (Sec. S7.6.3). Stable fixed points are identified where the flow directions are both pointing towards this flip point, while saddle nodes are identified where they are pointing away (Fig. 4B & C.)

### 4.3  Variations in the Topologies of the Slow Manifold Solutions

To understand what solutions the RNNs found to solve the task, we investigate their memory mechanism. For this, we dissect the dynamics of RNNs by segregating time scales to delineate the rapid flow normal to the slow manifold, and the flow on the manifold (Sec. S7.6.3). All solutions

---

[§]This correspond a type of "ideal pattern" in the vocabulary of Chirimuuta[84]. Our framework proposes a general approach to abstract away irrelevant details in models for analog memory[85].

involve a slow manifold with the same topology as the relevant variable in the task. The different solutions are different in their asymptotic dynamics (Fig. 4). The most often found solution is of the type *fixed point ring manifold* (Fig. 4B and C). These solutions are consistent with observations that persistent activity relies on discrete attractors[102,103]. Less commonly found topologies includes the slow torus around a repulsive ring invariant manifold (Fig. 4D). This solution in turn is consistent with both observations of the possibility of using non-constant dynamics for memory storage[19,104] and neuronal circuits underlying persistent representations despite time-varying activity[105]. All stability structures (fixed points and limit cycles) are mapped close to the target output circle (Figs. S15, S19, S20). We also verify the normal hyperbolicity of the trained networks shown in Fig. 4B and C. The largest eigenvalue of the Jacobian fluctuates around zero (the invariant manifold is not a continuous attractor), but it is removed from the second largest (Fig. 4E & F).

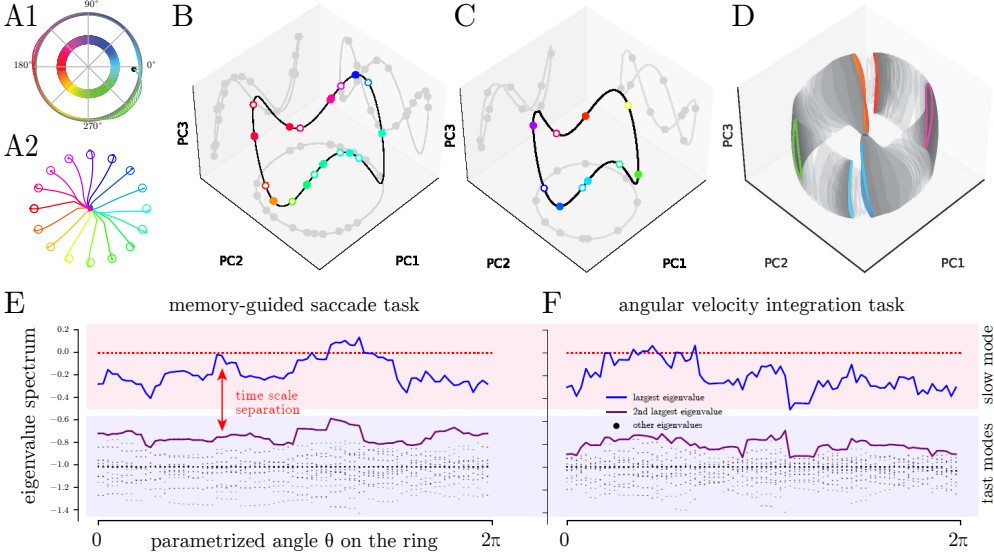

Figure 4: Slow manifold approximation of different trained networks on the memory-guided saccade and angular velocity integration tasks. (A1) Output of an example trajectory on the angular velocity integration task. (A2) Output of example trajectories on the memory-guided saccade task. (B) An example fixed-point type solution to the memory-guided saccade task. Circles indicate fixed points of the system (filled for stable, empty for saddle) and the decoded angular value on the output ring is indicated with the color according to A1. (C) An example of a found solution to the angular velocity integration task. (D) An example slow-torus type solution to the memory-guided saccade task. The colored curves indicate stable limit cycles of the system. (E+F) The eigenvalue spectra for the trained networks in B and C show a gap between the first two largest eigenvalues.

## 4.4 Universality amongst Good Solutions

The fixed point topologies show a lot of variation across networks (Fig. 4B,C, Fig. 5 and Fig. S20), much like the systems next to continuous attractors (Fig. 1 and Fig. 2). Previously, it has been observed that fixed point analysis has a major limitation, namely, that the number of fixed points must be equal across compared networks[91]. Our methodology effectively addresses and overcomes this limitation. The universal structure of continuous attractor approximations as slow invariant manifolds allows us to connect different topologies as **approximate continuous attractors** (Sec. 3.3). For results on LSTMs and GRUs and a higher dimensional task, see Sec. S7.7 and Sec. S7.9, respectively.

## 5   Generalization Analysis

In this section, we use task-trained RNNs to study the relationship between dynamics and generalization capabilities. When neuroscientists study neural computations in animals, tasks have finite durations, leaving it unclear whether animals learn the intended computation or a finite-time approximation. The same issue applies to trained neural networks. We will explore whether the networks possess the necessary memory for perfect recall or only perform the task within the timescale of their training.

The two possible approximations of a ring attractor, a limit cycle or a fixed point ring manifold (Sec. 3.3), exhibit markedly distinct generalization characteristics. Approximating the system as a limit cycle results in a memory trace that gradually diminishes over time (c.f. Park et al. [19]). Conversely, the alternative approximation's memory states are contingent upon the quantity and positioning of stable fixed points within the system. We describe in detail the generalization properties of the trained networks[¶] on the angular velocity integration task at two different time scales: asymptotic and finite time.

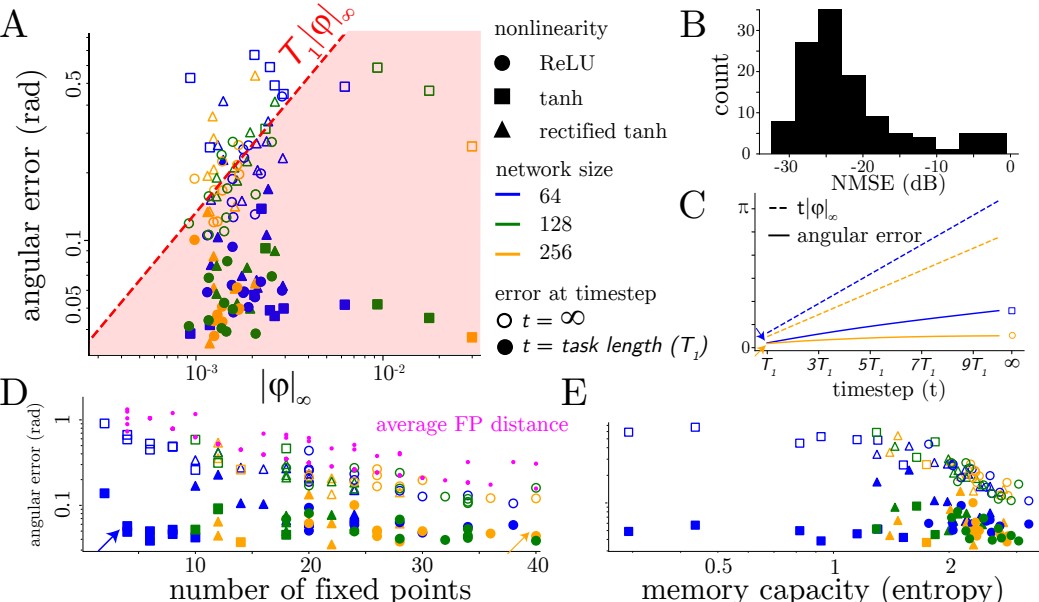

Figure 5: Temporal generalization validates theoretical predictions regardless of implementation detail. (A) Average accumulated angular error versus the maximum flow on the manifold (Eq. 5), shown for finite time (task duration that the networks were trained on, $T_1$; filled markers) and at asymptotic time (hollow markers). (B) Normalized validation loss of all trained networks. (C) Average error and theoretical upper bound over time for two selected networks (corresponding to arrows in panel D). (D) Average asymptotic error is roughly inversely proportional to the number of fixed points. (E) Memory capacity is predictive of the average error.

**Finite time:** Along with the angular velocity integration component of the task, the trained networks learn to store a memory of an angular variable. We assess the performance of the network to store the memory of the angle over time. The networks typically perform well on the timescale on which they have been trained, $T_1 = 256$ time steps (Fig. 5C). The memory error for $T_1$ is, as theoretically predicted (Eq. 5, see Prop. 2), bounded by the uniform norm of the vector field on the invariant manifold, and therefore by the distance to a continuous attractor (Prop. 1, Fig. 5A, Sec. S7.6.3 & S6).

**Asymptotic time:** Looking beyond the finite timescale provides valuable insights into the network's ability to store information. For the asymptotic time scale, we capture the asymptotic behavior of the system by identifying to what part of the state space the system evolves to in the limit $t \rightarrow \infty$ (see also Sec.S7.6.2). For a one-dimensional system, this will either be fixed points or a limit cycle. For the fixed-point type solution, the maximal error is given by the maximal distance to the next fixed point, while for a limit cycle, this will always be $\pi$. We calculate the *average fixed point distance* by taking the average of the inter-fixed-point interval for each neighboring pair of fixed points.

We can also quantify the loss of information. Assuming a uniform distribution over the angles, we define the *memory capacity* as the negative conditional entropy of the continuous memory given the asymptotic state, i.e. the stable fixed points (see Sec. S7.6.2 and Eq. 68).

---

[¶]We tested all networks with a validation set and took a cutoff for the normalized MSE for the networks we consider for the analysis at -20 dB (Fig. 5B).

**Error Accumulation in Neural Networks:** The mean accumulated error at the time at which the task was trained has an exponential relationship with the number of fixed points (Fig. 5A). Furthermore, this error is bounded by the mean distance between stable and unstable fixed points (red dots in Fig. 5D). This is another indication that the networks rely on a ring invariant manifold to implement the task. Networks with different numbers of fixed points might have the same performance on the finite time scale (bounded by $T_1 \|\varphi\|_\infty$) but have vastly different generalization properties because they differ in the number of fixed points (Fig. 5C).

## 6 Approximate Slow Manifolds are near Continuous Attractors

In Sec. 2.2, we presented a theory of approximate solutions in the neighborhood of continuous attractors. When are approximate solutions to the analog working memory problem near a continuous attractor? We posit that there are four conditions (see for more detail Sec. S5.3): **(C1)** sufficiently smooth approximate bijection between neural activity and memory content, **(C2)** the speed of drift of memory content is bounded, **(C3)** robustness against state (S-type) noise, and **(C4)** robustness against dynamical (D-type) noise [19]. The correspondence implied by **(C1)** translates to the existence of a manifold in the neural activity space with the same topology as the memory content.[‖] Persistence **(C2)** requires that the flow on the manifold is slow and bounded. S-type robustness **(C3)** implies non-expansive flow, i.e., non-positive Lyapunov exponents. Along with D-type robustness **(C4)**, it implies the manifold is "attractive", and normally hyperbolic (see also Sec. S5.3.1).

If these four conditions hold, for example for task-trained RNNs, there exists a smooth function with a uniform norm matching the slowness on the manifold such that when added, the slow manifold becomes a continuous attractor (Prop. 1 and Theorem 7, see also Sec. S5.4). For the RNN experiments, we added state-noise while training using stochastic gradient descent, satisfying **(C3)** and **(C4)**. We have also verified that **(C2)** holds (Fig. 5A). Although the stochastic optimization cannot lead to *the* continuous attractor solution, it gets to the neighborhood where all approximate solutions share the same main feature: having a subsystem that has a slow flow.

## 7 Discussion

Continuous attractors are highly prone to bifurcation under arbitrary perturbations unless they exist in special parametric forms. This sensitivity to perturbations has traditionally made them seem unsuitable for modeling neural computation in noisy biological systems, according to conventional views on robustness. Nevertheless, we demonstrate that continuous attractors can exhibit functional robustness, making them a crucial concept in explaining the neural computation underlying analog memory. We show that approximations of analog memory (i.e., theoretical models that satisfy conditions **(C1)**-**(C4)**) must possess slow manifold dynamics, placing them near continuous attractors within the space of dynamical systems. This implies that both biological systems and artificial neural networks only need to be near a continuous attractor to effectively solve problems in a manner similar to the ideal theoretical model, on behaviorally relevant timescales.

Although we expressed our theory in a non-parametric manner with an arbitrary perturbation $\mathbf{p}(\cdot)$, we can easily extend it to particular parametric forms such as biophysical models or an RNN using a sensitivity of the flow to the parameters (e.g. synaptic weight). Our theory can be applied to latent dynamical systems estimated from neural recordings [107]. As a framework, it can abstract out the details in imperfect dynamical implementations, however, it is an open problem to directly recover the continuous attractor from neural recordings or extend it to other ideal computational motifs.

**Limitations**    Although, we only explicitly describe the topology and dimensionality of the identified invariant manifolds for a representative set, the results indicate that most solutions have a ring invariant manifold with a slow flow. Our numerical analysis relies on identifying a time scale separation from simulated trajectories. If the separation of time scales is too small, it may inadvertently identify parts of the state space that are only forward invariant (i.e., transient). However, this did not pose a problem in our analysis of the trained RNNs, which is unsurprising, as the separation is guaranteed by state noise robustness (due to injected state noise during training).

The possible solutions that the networks can find are restricted by having a linear output mapping. In Park et al. [19], an alternative dynamical solution using oscillators (or quasi-periodic toroidal attractor) was described, however, a nonlinear readout may be necessary.

---

[‖]Note that effectively feedforward solutions [106] do not satisfy **(C1)**.

## Code Availability

The code for this project is publicly available at `https://github.com/catniplab/back_to_the_continuous_attractor`.

## Acknowledgments and Disclosure of Funding

We would like to extend our heartfelt thanks to Sukbin Lim, Srdjan Ostojic and Daniel Durstewitz for their invaluable feedback and thoughtful suggestions which significantly refined the work. This work was supported by NIH RF1-DA056404 and the Portuguese Recovery and Resilience Plan (PPR), through project number 62, Center for Responsible AI, and the Portuguese national funds, through FCT—Fundação para a Ciência e a Tecnologia—in the context of the project UIDB/04443/2020.

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

## Supplemental Material

## S1    Intuitive definitions of several key concepts used in our paper

**Manifold:**    A part of the state-space that locally resembles a flat, ordinary space (such as a plane or a three-dimensional space, but more generally -dimensional Euclidean space) but can have a more complicated global shape (such as a donut).

**Invariant set:**    A property of a set of points in the state space where, if you start within the set, all future states remain within the set and all past states belong to the set as well.

**Normally Hyperbolic Invariant Manifold:**    A behavior of a dynamical system where flow in the direction orthogonal to the manifold converges (or diverges) to the manifold significantly faster than the direction that remains on the manifold.

**Diffeomorphism:**    A diffeomorphism is a stretchable map that can be used to transform one shape into another without tearing or gluing. A differentiable map with differentiable inverse.

$C^1$ **neighborhood of a** $C^1$ **function:**    A set of functions that are close to the function in terms of both their values and their first derivatives.

**Compact Set:**    A set where every sequence of points has a subsequence that converges to a point within the set. Intuitively, it means the set is closed and bounded, making it "finite" in a certain sense.

**Connecting Orbit:**    A trajectory in a dynamical system that connects two different equilibrium points or periodic orbits. Specifically, a heteroclinic orbit connects distinct equilibrium points.

## S2 The bounded line attractor

In order to demonstrate the implications of the theory of the persistence of continuous attractors, we rigorously test the predictions of the theory on the stability of the Bounded Line Attractor (BLA). Our objective is to assess the practical implications of the theoretical findings of bounded continuous attractors in a small and tractable system, and second, to contribute empirical evidence that can help refine and extend existing theoretical frameworks.

The BLA has a parameter that determines step size along line attractor $\alpha$. Analogously as for UBLA, these parameters determine the capacity of the network. The inputs push the input along the line attractor in two opposite directions, see below. The BLA needs to be initialized at $\beta(1, 1)$ and $\frac{\beta}{2}(1, 1)$, respectively, for correct decoding, i.e., output projection.

$$\mathbf{W}_{\text{in}} = \alpha \begin{pmatrix} -1 & 1 \\ 1 & -1 \end{pmatrix}, \ \mathbf{W} = \begin{pmatrix} 0 & -1 \\ -1 & 0 \end{pmatrix}, \ \mathbf{W}_{\text{out}} = \frac{1}{2\alpha} \begin{pmatrix} 1 \\ -1 \end{pmatrix}, \ \mathbf{b} = \beta \begin{pmatrix} 1 \\ 1 \end{pmatrix}, \ \mathbf{b}_{\text{out}} = 0. \quad (6)$$

Parameter that determines step size along line attractor $\alpha$. The size determines the maximum number of clicks as the difference between the two channels. This pushes the input along the line "attractor" in two opposite directions, see below.

The results from such low-dimensional system can be extended to higher-dimensional systems through reduction methods from center manifold theory. On the center manifold the singular perturbation problem (as is the case for continuous attractors) restricts to a regular perturbation problem [108]. Furthermore, relying on the Reduction Principle [109], one can always reduce all systems (independent of dimension) to the same canonical form, given that they have the same continuous attractor.

### S2.1 Fast-slow form

First of all, we will show how to transform the BLA network to the slow-fast form in Eq. 3-4 to explicitly demonstrate that the theory applies to it. To achieve this, we transform the state space so that the line attractor aligns with the $y$-axis. So, we apply the affine transformation $R_\theta(x - \frac{1}{2})$ with the rotation matrix $R_\theta = \begin{bmatrix} \cos\theta & -\sin\theta \\ \sin\theta & \cos\theta \end{bmatrix} = \frac{1}{\sqrt{2}} \begin{bmatrix} 1 & 1 \\ -1 & 1 \end{bmatrix}$ where we have set $\theta = -\frac{\pi}{4}$. So we perform the transformation $x \to x' = R_\theta(x - \frac{1}{2})$ and so we have $x = R_\theta^{-1} x' + \frac{1}{2}$ with $R_\theta^{-1} = R_{-\theta}$. Then we get that

$$R_\theta^{-1} \dot{x}' = \text{ReLU}\left(W(R_\theta^{-1} x^{\backprime} + \frac{1}{2}) + 1\right) - R_\theta^{-1} x' - \frac{1}{2}. \quad (7)$$

For a perturbed connection matrix $W = \begin{bmatrix} \epsilon & -1 \\ -1 & 0 \end{bmatrix}$ we get

$$R_\theta^{-1} \dot{x}' = \text{ReLU}\left(\frac{1}{\sqrt{2}} \begin{bmatrix} \epsilon & -1 \\ -1 & 0 \end{bmatrix} \left(\begin{bmatrix} 1 & -1 \\ 1 & 1 \end{bmatrix} x^{\backprime} + \frac{1}{2}\right) + 1\right) - \frac{1}{\sqrt{2}} \begin{bmatrix} 1 & -1 \\ 1 & 1 \end{bmatrix} x' - \frac{1}{2} \quad (8)$$

$$\dot{x}' = \begin{bmatrix} -1 & 1 \\ 1 & 1 \end{bmatrix} \left(\frac{1}{2} \begin{bmatrix} \epsilon - 1 & -\epsilon - 1 \\ -1 & 1 \end{bmatrix} x' + \frac{1}{2\sqrt{2}} \begin{bmatrix} \epsilon - 1 \\ -1 \end{bmatrix} + \begin{bmatrix} 1 \\ 1 \end{bmatrix} - \frac{1}{2} \begin{bmatrix} 1 \\ 1 \end{bmatrix}\right) - x' \quad (9)$$

$$\dot{x}' = \left(\begin{bmatrix} -2 & 0 \\ 0 & 0 \end{bmatrix} + \frac{\epsilon}{2} \begin{bmatrix} 1 & -1 \\ -1 & 1 \end{bmatrix}\right) x' + \frac{1}{2\sqrt{2}} \begin{bmatrix} \epsilon \\ -\epsilon \end{bmatrix} \quad (10)$$

### S2.2 Bifurcation analysis

We will now identify all possible bifurcations from the BLA, to show that indeed all perturbations preserve the continuous attractor as an invariant manifold.

We consider all parametrized perturbations of the form $\mathbf{W} \leftarrow \mathbf{W} + \mathbf{V}$ for a random matrix $\mathbf{V} \in \mathbb{R}^{2 \times 2}$ to the BLA. The BLA can bifurcate in the following systems, characterized by their invariant sets: a system with single stable fixed point, a system with three fixed points (one unstable and two stable) and a system with two fixed points (one stable and the other a half-stable node) and a system with a (rotated) line attractor. Only the first two bifurcations (Fig. 1A) can happen with nonzero chance for the type of random perturbations we consider. The perturbations that leave the line attractor intact or to lead to a system with two fixed points have measure zero in the parameter space. The perturbation that results in one fixed point happen with probability $\frac{3}{4}$, while perturbations lead to a system with three fixed points with probability $\frac{1}{4}$, see Sec. S2.2.2. The (local) invariant manifold manifold is indeed persistent for the BLA and homeomorphic to the original (the bounded line).

**Stabilty of the fixed point with full support** We investigate how perturbations to the bounded line affect the Lyapunov spectrum. We calculate the eigenspectrum of the Jacobian:

$$\det[W' - (1 + \lambda)\mathbb{I}] = (\epsilon_{11} - 1 - \lambda)(\epsilon_{22} - 1 - \lambda) - (\epsilon_{12} + 1)(\epsilon_{21} + 1)$$
$$= \lambda^2 - (2 + \epsilon_{11} + \epsilon_{22})\lambda - \epsilon_{11} - \epsilon_{22} + \epsilon_{11}\epsilon_{22} - \epsilon_{12} - \epsilon_{21} - \epsilon_{12}\epsilon_{21}$$

Let $u = -(2 + \epsilon_{11} + \epsilon_{22})$ and $v = -\epsilon_{11} - \epsilon_{22} + \epsilon_{11}\epsilon_{22} - \epsilon_{12} - \epsilon_{21} - \epsilon_{12}\epsilon_{21}$

There are only two types of invariant set for the perturbations of the line attractor. Both have as invariant set a fixed point at the origin. What distinguishes them is that one type of perturbations leads to this fixed point being stable while the other one makes it unstable.

**Stability of the fixed points on the axes**    We perform the stability analysis for the part of the state space where $Wx > 0$. There, the Jacobian is

$$J = - \begin{pmatrix} 1 & 1 \\ 1 & 1 \end{pmatrix} \tag{11}$$

We apply the perturbation

$$W' = \begin{pmatrix} 0 & -1 \\ -1 & 0 \end{pmatrix} + \epsilon \tag{12}$$

with

$$\epsilon = \begin{pmatrix} \epsilon_{11} & \epsilon_{12} \\ \epsilon_{21} & \epsilon_{22} \end{pmatrix} \tag{13}$$

The eigenvalues are computed as

$$\det[W' - (1+\lambda)\mathbb{I}] = (\epsilon_{11} - 1 - \lambda)(\epsilon_{22} - 1 - \lambda) - (\epsilon_{12} - 1)(\epsilon_{21} - 1)$$
$$= \lambda^2 + (2 - \epsilon_{11} - \epsilon_{22})\lambda - \epsilon_{11} - \epsilon_{22} + \epsilon_{11}\epsilon_{22} + \epsilon_{12} + \epsilon_{21} - \epsilon_{12}\epsilon_{21}$$

Let $u = 2 - \epsilon_{11} - \epsilon_{22}$ and $v = -\epsilon_{11} - \epsilon_{22} + \epsilon_{11}\epsilon_{22} + \epsilon_{12} + \epsilon_{21} - \epsilon_{12}\epsilon_{21}$

$$\lambda = \frac{-u \pm \sqrt{u^2 - 4v}}{2} \tag{14}$$

Case 1: $\mathrm{Re}(\sqrt{u^2 - 4v}) < -u$, then $\lambda_{1,2} < 0$

Case 2: $\mathrm{Re}(\sqrt{u^2 - 4v}) > -u$, then $\lambda_1 < 0$ and $\lambda_2 > 0$

Case 3: $v = 0$, then $\lambda = \frac{1}{2}(-u \pm u)$, i.e., $\lambda_1 = 0$ and $\lambda_2 = -u$

$$\epsilon_{11} = -\epsilon_{22} + \epsilon_{11}\epsilon_{22} + \epsilon_{12} + \epsilon_{21} - \epsilon_{12}\epsilon_{21} \tag{15}$$

We give some examples of the different types of perturbations to the bounded line attractor. The first type is when the invariant set is composed of a single fixed point, for example for the perturbation:

$$\epsilon = \frac{1}{10} \begin{pmatrix} -2 & 1 \\ 1 & -2 \end{pmatrix} \tag{16}$$

The second type is when the invariant set is composed of three fixed points:

$$\epsilon = \frac{1}{10} \begin{pmatrix} 1 & -2 \\ -2 & 1 \end{pmatrix} \tag{17}$$

The third type is when the invariant set is composed of two fixed points, both with partial support.

$$b' = \frac{1}{10} \begin{pmatrix} 1 & -1 \end{pmatrix} \tag{18}$$

The fourth and final type is when the line attractor is maintained but rotated:

$$\epsilon = \frac{1}{20} \begin{pmatrix} 1 & 10 \\ 10 & 1 \end{pmatrix} \tag{19}$$

### S2.2.1    Bifurcation landscape

We will now state our previous observations as a Theorem that characterizes the possible bifurcations of the BLA.

**Theorem 2.** *All perturbations of the bounded line attractor are of the types as listed above.*

*Proof.* We enumerate all possibilities for the dynamics of a ReLU activation network with two units. First of all, note that there can be no limit cycle or chaotic orbits.

Now, we look at the different possible systems with fixed points. There can be at most three fixed points[110] Corollary 5.3. There has to be at least one fixed point, because the bias is non-zero.

General form (example):

$$\epsilon = \frac{1}{10} \begin{pmatrix} -2 & 1 \\ 1 & -2 \end{pmatrix} \tag{20}$$

One fixed point with full support:

In this case we can assume $W$ to be full rank.

$$\dot{x} = \mathrm{ReLU}\left[\begin{pmatrix} \epsilon_{11} & \epsilon_{12} \\ \epsilon_{21} & \epsilon_{22} \end{pmatrix}\begin{pmatrix} x_1 \\ x_2 \end{pmatrix} + \begin{pmatrix} 1 \\ 1 \end{pmatrix}\right] - \begin{pmatrix} x_1 \\ x_2 \end{pmatrix} = 0$$

Note that $x > 0$ iff $z_1 := \epsilon_{11}x_1 + (\epsilon_{12} - 1)x_2 - 1 > 0$. Similarly for $x_2 > 0$.

So for a fixed point with full support, we have

$$\begin{pmatrix} x_1 \\ x_2 \end{pmatrix} = A^{-1}\begin{pmatrix} -1 \\ -1 \end{pmatrix} \tag{21}$$

with

$$A := \begin{pmatrix} \epsilon_{11} - 1 & \epsilon_{12} - 1 \\ \epsilon_{21} - 1 & \epsilon_{22} - 1 \end{pmatrix}.$$

Note that it is not possible that $x_1 = 0 = x_2$.

Now define

$$B := A^{-1} = \frac{1}{\det A}\begin{pmatrix} \epsilon_{22} - 1 & 1 - \epsilon_{12} \\ 1 - \epsilon_{21} & \epsilon_{11} - 1 \end{pmatrix}$$

with

$$\det A = \epsilon_{11}\epsilon_{22} - \epsilon_{11} - \epsilon_{22} - \epsilon_{12}\epsilon_{21} + \epsilon_{12} + \epsilon_{21}.$$

Hence, we have that $x_1, x_2 > 0$ if $B_{11} + B_{12} > 0$, $B_{21} + B_{22} > 0$ and $\det A > 0$ or if $B_{11} + B_{12} < 0$, $B_{21} + B_{22} < 0$ and $\det A < 0$.

This can be satisfied in two ways, If $\det A > 0$, this is satisfied if $\epsilon_{22} > \epsilon_{12}$ and $\epsilon_{11} > \epsilon_{21}$, while if $\det A < 0$, this is satisfied if $\epsilon_{22} < \epsilon_{12}$ and $\epsilon_{11} < \epsilon_{21}$. This gives condition 1.

Finally, we investigate the condition that specify that there are fixed points with partial support. If $x_1 = 0$ then $(\epsilon_{22} - 1)x_2 + 1 = 0$ and $z_1 < 0$. From the equality, we get that $x_2 = \frac{1}{1 - \epsilon_{22}}$. From the inequality, we get $(\epsilon_{12} - 1)x_2 + 1 \geq 0$, i.e. $\frac{1}{1 - \epsilon_{12}} \geq x_2$. Hence,

$$\frac{1}{1 - \epsilon_{12}} \geq \frac{1}{1 - \epsilon_{22}}$$

and thus

$$\epsilon_{22} \leq \epsilon_{12}. \tag{22}$$

Similarly to have a fixed point $x^*$ such that $x_2^* = 0$, we must have that

$$\epsilon_{11} \leq \epsilon_{21}. \tag{23}$$

Equation 22 and 23 together form condition 2.

Then, we get the following conditions for the different types of bifurcations:

1. If condition 1 is violated, but condition 2 is satisfied with exactly one strict inequality, there are two fixed points on the boundary of the admissible quadrant.

2. If condition 1 is violated, and only one of the subconditions of condition 2 is satisfied, there is a single fixed point on one of the axes.

3. If condition 2 is violated, there is a single fixed point with full support.

4. If both conditions are satisfied, there are three fixed points.

We now look at the possibility of the line attractor being preserved. This is the case if $v = 0$. It is not possible to have a line attractor with a fixed point off of it for as there cannot be disjoint fixed points that are linearly dependent[110] Lemma 5.2 □

## S2.2.2 Probability of bifurcation types

We will now calculate which proportion proportion of the bifurcation parameter space is results in the different bifurcation types. The conditions that result in three fixed points are

$$0 < \epsilon_{11}\epsilon_{22} - \epsilon_{11} - \epsilon_{22} - \epsilon_{12}\epsilon_{21} - \epsilon_{12} - \epsilon_{21},$$

$$\epsilon_{22} \leq \epsilon_{12},$$

$$\epsilon_{11} \leq \epsilon_{21}.$$

Therefore, because

$$\epsilon_{22} \leq \epsilon_{12},$$
$$\epsilon_{11} \leq \epsilon_{21}.$$

we always have that

$$0 < \epsilon_{11}\epsilon_{22} - \epsilon_{11} - \epsilon_{22} - \epsilon_{12}\epsilon_{21} - \epsilon_{12} - \epsilon_{21}.$$

This implies that this bifurcation happens with probability $\frac{1}{4}$ in a $\epsilon$-ball around the BLA neural integrator with $\epsilon < 1$. We conclude that the single stable fixed point type perturbation happens with probability $\frac{3}{4}$.

## S2.3 Structure of the parameter space

We will present the structure of the bifurcation space through a slice in which we fix $\epsilon_{11}$ and $\epsilon_{12}$. First, we summarize which conditions result in which bifurcation in Table 1. We derive that the local bifurcation in this slice has the structure as shown in Fig. S6.

Table 1: Summary of the conditions for the different bifurcations.

|  | 1FP (full) | 1FP (partial) | 3FPs | 2FPs | LA |
|---|---|---|---|---|---|
| C1 | ✓ | ✗ | ✓ | ✗ | ✗ |
| C2 | ✗ | only Eq22 or 23 | ✓ | ✓ | ✗ |

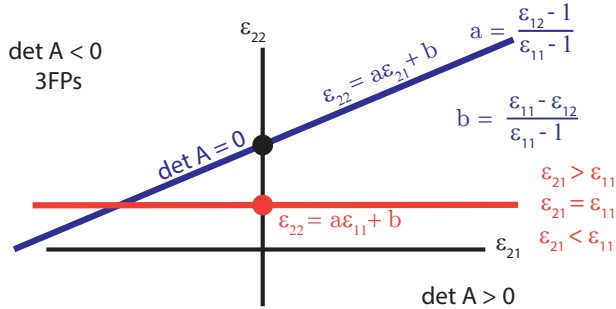

Figure S6: A slice of the parameter space of the BLA for a fixed $\epsilon_{11}$ and $\epsilon_{12}$.

## S2.4 Smoother activation functions

It is well-known that activation functions ($\sigma$ in Eqs. 61 and 64), which can take many forms, play a critical role in propagating gradients effectively through the network and backwards in time[111–113]. Activation functions that are $C^r$ for $r \geq 1$ are the ones to which the Persistence Theorem applies. The Persistence Theorem further specifies how the smoothness of the activation can have implications on the smoothness of the persistent invariant manifold. For situations where smoothness of the persistent invariant manifold is of importance, smoother activation functions might be preferable, such as the Exponential Linear Unit (ELU)[114] or the Continuously Differentiable Exponential Linear Units (CELU)[115].

## S3 Ring perturbations

To computationally investigate the neighborhood of recurrent dynamical systems that implement continuous attractors, we investigate 5 RNNs that are known a priori to form 1 or 2 dimensional continuous attractors.

We define a local perturbation (i.e., a change to the ODE with compact support) through the bump function $\Psi(x) = \exp\left(\frac{1}{\|x\|^2 - 1}\right)$ for $\|x\| < 1$ and zero outside, by multiplying it with a uniform, unidirectional vector field. All such perturbations leave at least a part of the continuous attractor intact and preserve the invariant manifold, i.e. the parts where the fixed points disappear a slow flow appears.

The parametrized perturbations are characterized as the addition of a random matrix to the ODE.

### S3.1 Simple ring attractor

We further analyzed a simple (non-biological) ring attractor, defined by the following ODE: $\dot{r} = r(1 - r)$, $\dot{\theta} = 0$. This system has as fixed points the origin and the ring with radius one centered around zero, i.e., $(0,0) \cup \{(1,\theta) \mid \theta \in [0, 2\pi)\}$. We investigate bifurcations caused by parametric and bump perturbations of the ring invariant manifold (see Sec. S3), which is bounded and boundaryless. All perturbations maintain the topological structure of the invariant manifold.

### S3.2 Heading direction network

The networks proposed in [8] are composed of $N$ heading-tuned neurons whose preferred headings $\theta_j$ uniformly tile heading space, with an angular separation of $\Delta\theta = \frac{2\pi}{N}$ radians. These neurons can be arranged topologically in a ring according to their preferred headings, with neurons locally exciting and broadly inhibiting their neighbors. The total input activity $h_j$ of each neuron is governed by:

$$\tau \dot{h}_j = -h_j + \frac{1}{N}\sum_k (W_{jk}^{sym} + v_{in} W_{jk}^{asym})\phi(h_k) + c_{ff}, j = 1, \ldots, N, \tag{24}$$

with

$$W_{jk}^{sym} = J_I + J_E \cos(\theta_j - \theta_k), \tag{25}$$

where $J_E$ and $J_I$ respectively control the strength of the tuned and untuned components of recurrent connectivity between neurons with preferred headings $\theta_j$ and $\theta_j$ and $v_{in}$ is an angular velocity input which the network receives through asymmetric, velocity-modulated weights

$$W^{asym} = \sin(\theta_j - \theta_k). \tag{26}$$

**Fixed points** In the absence of an input ($v_{in} = 0$) fixed points of the system can be found analytically by considering all submatrices $W_\sigma^{sym}$ for all subsets $\{\sigma \subset [n]\}$ with $[n] = \{1, \ldots, N\}$. A fixed point $x^*$ needs to satisfy

$$x^* = -(W_\sigma^{sym})^{-1}c_{ff} \tag{27}$$

and

$$x_i^* < 0 \text{ for } i \in \sigma. \tag{28}$$

We bruteforce check all possible supports to find all fixed points. We use the eigenvalues of the Jacobian to identify the stability of the found fixed points. We evaluate the effect of parametric perturbations of a network of size $N = 6$ with $J_E = 4$ and $J_I = -2.4$ by identifying all bifurcations (Fig. 2A).

**Connecting orbits** We approximate connecting orbits through numerical integration of the ODE intialized in close to the identified saddle points along the unstable manifold.

**Measure zero co-dimension 1 bifurcations**    Measure zero co-dimension 1 bifurcations of the ring attractor network fall into two types, see Fig. S7.

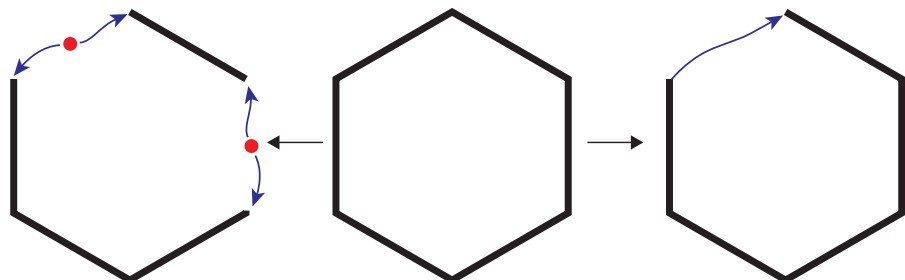

Figure S7: Measure zero co-dimension 1 bifurcations of the ring attractor network [8].

**Measure zero co-dimension $N$ bifurcation**    The limit cycle is the only bifurcation that we found that can be achieved on only a measure zero set of parameter values around the parameter for the continuous attractor.

**Independence of norm of perturbation on bifurcation**    As we can see in Fig. S8, the topology of the system is maintained through a range of bifurcation sizes when the bifurcation direction is fixed.

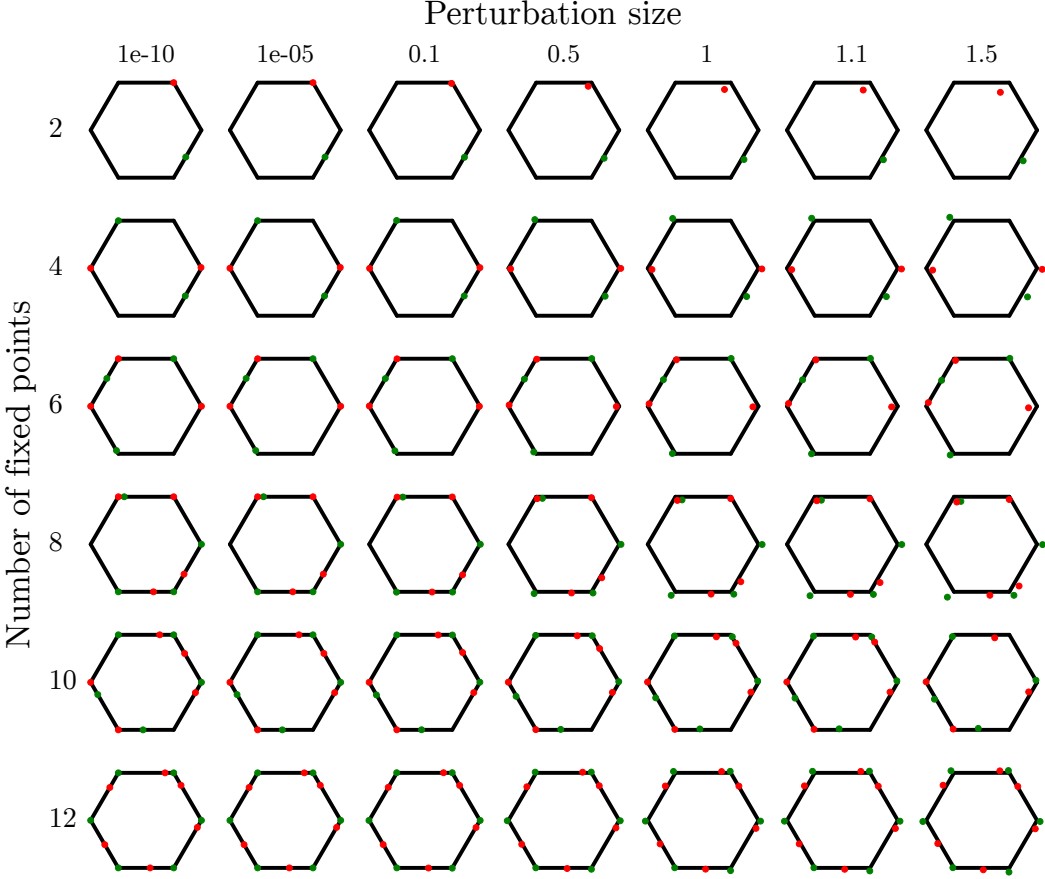

Figure S8: Rows show the bifurcations resulting from perturbations from the matrices with the same direction in Fig. 2A but with different norms (columns).

## S3.3 Ring attractor approximation with tanh neurons

We investigated the bifurcations around the approximate ring attractor constructed with a symmetric weight matrix for a tanh network[58,62]. The functional form of $W$ is the sum of a constant term plus a Gaussian centered at $\theta_i - \theta_j = 0$:

$$W(\theta_i - \theta_j) = J^- + (J^+ - J^-)\exp\left[-\frac{(\theta_i - \theta_j)^2}{2\sigma^2}\right],\tag{29}$$

with the dimensionless parameter $J^-$ representing the strength of the weak crossdirectional connections, $J^+$ the strength of the stronger isodirectional connections, and $\sigma$ the width of the connectivity footprint.

Such ring attractor approximations are similar to the ones in[60,61,116,117]. However, some have other nonlinearities, e.g., the sigmoid is used in[61]. Another line of related models can be found in[118],[119] and[120].

**Loss of function: Sensitivity of continuous attractors to perturbations** We will show that there are differences at how well approximations perform at different timescales. We measure how performance of different models for the representation of an angular variable drop as a function of perturbation size Fig. S9 through the memory capacity metric (Sec.S7.6.2). For each perturbation size, we sample a low rank (rank 1,2 or 3) random matrix with norm equal to that perturbation size. We determine the location of the fixed points through the local flow direction criterion as described in Sec. 4.2 This invariant manifold was found to be consistently close the the original invariant ring attractor. The initial ring had $2N$ fixed points ($N$ stable, $N$ saddle) on this invariant ring manifold. The memory capacity of this initial configuration is $N\log(N)$ for the $2N$ uniformly spaced fixed points.

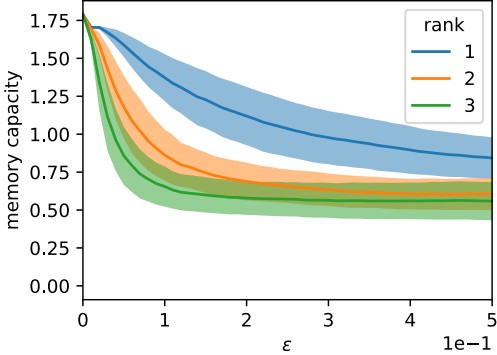

Figure S9: Degradation of performance across perturbation sizes. System behavior at the asymptotic time scales measured through memory capacity.

### S3.3.1 Mean field approaches

Another line of models[121–124] also relies on connection weights with translational invariance

$$J(x, x') = \frac{A}{\sqrt{2\pi a}}\exp\left[-\frac{(x - x')^2}{2a^2}\right]\tag{30}$$

where $J(x, x')$ is the neural interaction from $x'$ to $x$ and the ensemble of infinite neurons are lined up so that $x \in (-\infty, \infty)$.

Weak random spatial fluctuations in the connection strength are to be expected when learning the coupling function with Hebbian plasticity. In the 1D case, it is well known that the presence of such synaptic heterogeneity causes a drift of an input-induced activity pattern to one of a finite number of attractor positions which are randomly spread over representational space[16,20]. Similarly, in 2D, spatial fluctuations in the connection strengths cause a slight perturbation of the bump shape[125]. Frozen stabilisation has been proposed as alternative method to construct a neural networks to self-organise to a state exhibiting (high-dimensional) memory manifolds with arbitrarily large time constants[126].

[127] analyzes an Ising network perturbed with a specially structured noise at the thermodynamic limit. Although their analysis elegantly shows that the population activity of the perturbed system does not destroy the Fisher information about the input, they do not consider a scenario where the ring attractor is used as a working memory mechanism, it is rather used to encode instantaneous representation. In contrast, our analysis involves understanding how the working memory content degrades over time due to the dynamics. We are not aware of any mean field analysis that covers this aspect.

## S3.4 Ring attractor approximation with a low-rank network

The networks consisted of $N$ firing rate units with a sigmoid input-output transfer function[64]:

$$\dot{\xi}_i(t) = -\xi_i(t) + \sum_{j=1}^{N} J_{ij}\phi(x_j(t)) + I_i, \tag{31}$$

where $x_i(t)$ is the total input current to unit $i$, $J_{ij} = g\chi_{ij} + P_{ij}$ is the connectivity matrix, $\phi(x) = \tanh(x)$ is the current-to-rate transfer function, and $I_i$ is the external, feedforward input to unit $i$. The random component $g\chi$ is considered unknown except for its statistics (mean 0, variance $g^2/N$). A general structured component of rank $r \ll N$ can be written as a superposition of $r$ independent unit-rank terms

$$P_{ij} = \frac{m_i^{(1)}n_j^{(1)}}{N} + \cdots + \frac{m_i^{(r)}n_j^{(r)}}{N}, \tag{32}$$

and is in principle characterized by $2r$ vectors $m^{(k)}$ and $n^{(k)}$.

To approximate a ring attractor we can consider structured matrices where the two connectivity pairs $m^{(1)}$ and $n^{(1)}$, $m^{(2)}$ and $n^{(2)}$ share two different overlap directions, defined by vectors $y_1$ and $y_2$. We set:

$$m^{(1)} = \sqrt{\Sigma^2 - r_1^2}\, x_1 + r_1 y_1, \tag{33}$$

$$m^{(2)} = \sqrt{\Sigma^2 - r_2^2}\, x_2 + r_2 y_2, \tag{34}$$

$$n^{(1)} = \sqrt{\Sigma^2 - r_1^2}\, x_3 + r_1 y_1, \tag{35}$$

$$n^{(2)} = \sqrt{\Sigma^2 - r_2^2}\, x_4 + r_2 y_2, \tag{36}$$

where $\Sigma^2$ is the variance of the connectivity vectors and $r_1^2$ and $r_2^2$ quantify the overlaps along the directions $y_1$ and $y_2$.

We keep the following parameters for the analysis: $\Sigma = 2$, $\rho = 1.9$ and $g = 0.1$.

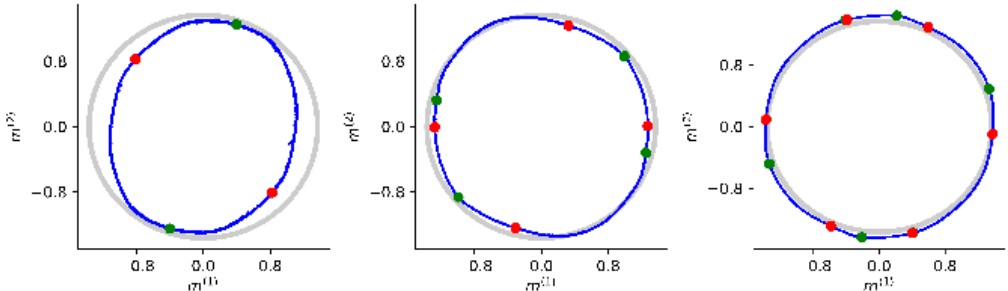

Figure S10: Some examples of networks dynamics for sizes $N = 10, 100, 1000$.

Our theory explains the phenomenon of the existence of a ring invariant manifold in these low-rank networks as follows. We can think of the finite size realization as a small perturbation to the infinite size network on the reduced dynamics in the $m^1, m^2$ plane (independent of the parameter $g$ for the random part of the matrix) (Fig. 2B). For very small networks the ring structure is destroyed and only the plane persists as a slow manifold.

## S3.5 Embedding Manifolds with Population-level Jacobians

We fit three networks with the Embedding Manifolds with Population-level Jacobians (EMPJ) method[59].

The networks are RNNs of $N = 10$ neurons with a $tanh$ activation function and $\tau = 0.05$ time constant. To use EMPJ we need to specify our desired $k$ fixed points and $m$ eigenvectors and eigenvalues per fixed point.

To do this we choose $k = 3$ points $\{(x_i, y_i)\}_{i=1}^{N}$ in the 2D ring with radius $r = 1$ that are equally spaced from 0.1 to $2\pi$ radians. Then we specify the vectors orthogonal to the ring in those points ($V_o = \{(x_i, y_i)\}_{i=1}^{N}$, same as the points) and tangent ($V_t = \{(-y_i, x_i)\}_{i=1}^{N}$). Finally we associate the negative eigenvalue $-1/\tau$ to the orthogonal eigenvectors and different eigenvalues to the tangent eigenvectors. We use $-1, 0$ or $1$ depending on whether we want the points to be stable, center or unstable in the direction of the slow ring manifold.

Since these determined dynamics are only in a 2D plane, we use a random linear mapping $D\colon \mathbb{R}^2 \to \mathbb{R}^N$ to map the fixed points and the eigenvectors to a 10-dimensional space. First, we sample a random from orthogonal matrix $A$ from the $O(N)$ Haar distribution and then we take the first two columns of this matrix to be $D = [A]_{12}$.

EMPJ then returns a network parameters that satisfy these constraints. Furthermore, due to the particular solver used, that regularizes the magnitude of the parameters, we get that all the other eigenvalues not specified are also set to $-1/\tau$ (for details see[59]).

**Finding fixed points**  As we remark in Sec. S3.5.1, EMPJ networks are not robust to S-type noise, therefore we cannot apply our analysis of identifying the invariant set through the convergence criterion of numerically integrated trajectories. We therefore find fixed points through the Newton-Raphson method. We iteratively solve

$$\mathbf{J}(\mathbf{x}_i)\mathbf{dx}_i = \mathbf{x}_i \tag{37}$$

where $\mathbf{J}(\mathbf{x}_i)$ is the Jacobian of the system at $\mathbf{x}_i$ and $\mathbf{x}_{i+1} = \mathbf{x}_i - \mathbf{dx}_i$. The iteration stops when $|\mathbf{dx}| < \delta$ for a tolerance threshold $\delta$. We initialize $\mathbf{x}_0$ on the invariant ring uniformly. The maximum number of iterations was set to 10.000 and the tolerance level to $\delta = 10^{-8}$.

### S3.5.1   Lack of S-type robustness

We remark that the resulting invariant manifold is not robust to S-type perturbations. Although the fixed points that are constrained in the fitting procedure are attractive in all directions, some of the points along the ring might not be. For on-manifold perturbations (in the plane in which the ring is embedded), S-type perturbations do lead to flow towards the invariant ring (Fig. S11A). However, for (small) off-manifold perturbations, the trajectories typically diverge away from the invariant ring (Fig. S11B). This indicates that the basin of attration is very small and hence this approximation is not robust to S-type noise.

All of the perturbations were sampled as $x(0) = x(0) + \eta$ with $x(0) \in \mathrm{span}(D)$. For the on-manifold perturbations $\eta \in \mathrm{span}(D)$ and $\|\eta\|_2 = 10^{-2}$. For the off-manifold perturbations $\eta \in \mathbb{R}^{10}$ and $\|\eta\|_2 = 10^{-5}$.

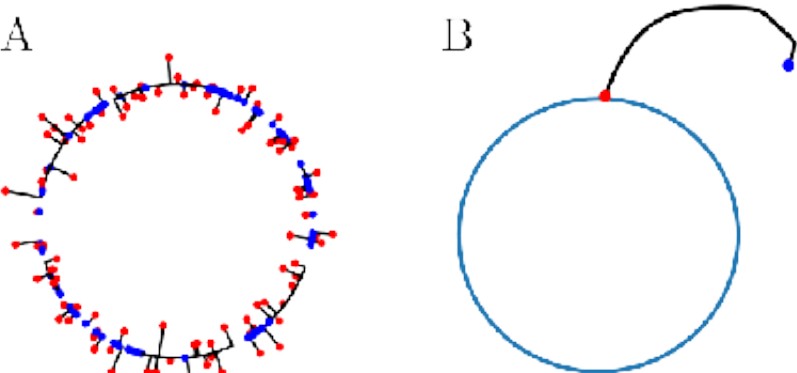

Figure S11: Trajectories of the third network in Fig. 2C. Starting point in red, end of trajectory in blue. (A) On-manifold S-type perturbations from the ring. (B) An example of an off-manifold S-type perturbation from the ring.

### S3.5.2   Higher dimensional manifolds

We furthermore fit a torus and a sphere continuous attractor with the EMPJ. The networks we used have $N = 100$ neurons. For finding fixed points with the Newton-Raphson method, we used 1.000 as the maximum number of iterations and a tolerance level of $\delta = 10^{-5}$.

**Torus**  Figure S12 illustrates the stability structures of the approximate torus attractor fitted with EMPJ. The ratio of the radii of the two rings is adjusted for visualization purposes.

**Sphere**  Fig. S13 illustrates the stability structures of the approximate sphere attractor fitted with EMPJ. Similar to the torus, points initialized off the sphere converge onto the sphere attractor (Fig. S13).

### S3.5.3   Other fixed point fitting methods

Storing multiple continuous attractors has been worked out in[128], with a similar approximation as our theory suggests. The different stored patterns form a discrete approximation of the continuous map (resulting in quasicontinuous maps, which corrspond to the approximate contintuous attractors in our theory).

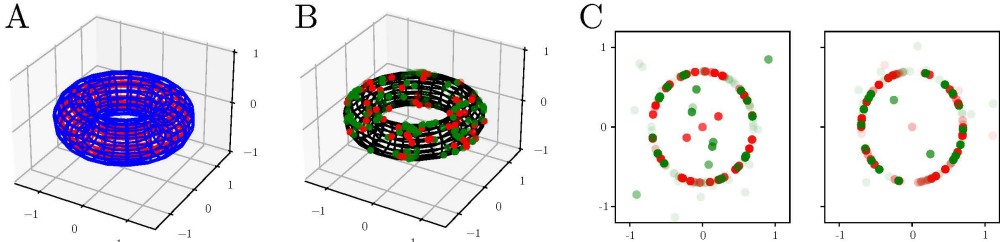

Figure S12: The approximate torus attractor. (A) Points initialized on a grid off of the torus (blue) converge onto the torus attractor (red). (B) The found fixed points on the approximate torus (green: stable, red: saddle). (C) The found fixed points projected onto the two 2D subspaces that defined the two rings of the torus.

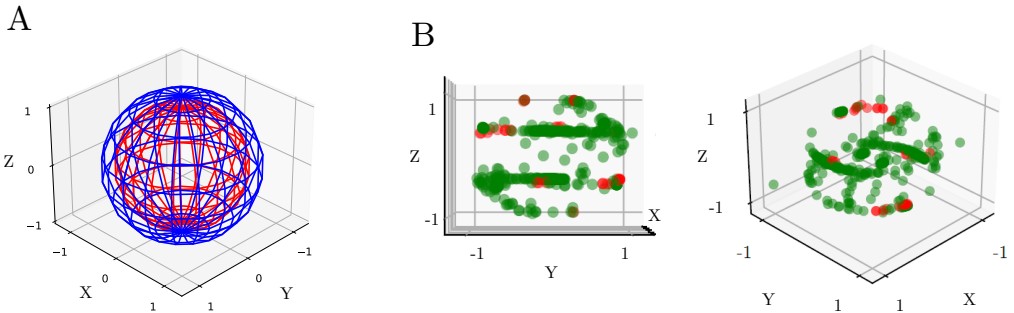

Figure S13: The approximate sphere attractor. (A) Points initialized on a grid off of the sphere (blue) converge onto the torus attractor (red). (B) The found fixed points on the approximate sphere (green: stable, red: saddle). The two subfigures show rotated versions of the location of the fixed points on the sphere.

## S4 Persistence Theorem

Understanding the long-term behavior of dynamical systems is a fundamental question in various fields, including mathematical biology, ecological modeling, and neuroscience. Fenichel's Persistence Theorem provides critical insights into the behavior of such systems, particularly in relation to the stability and persistence of invariant manifolds under perturbations.

Fenichel's Persistence Theorem extends the classical theory of invariant manifolds, offering conditions under which normally hyperbolic invariant manifolds persist despite small perturbations. This theorem is particularly powerful in analyzing systems where perturbations are inevitable, providing a framework for understanding how qualitative features of the system's dynamics are maintained. In this section, we delve into the specifics of Fenichel's Persistence Theorem, outlining its key components, assumptions, and implications.

**Invariance**   One of the main concepts in the Persistence Theorem is the notion of an invariant manifold. Intuitively, this just means that trajectories stay inside the manifold for all time. Local invariance is a bit more involved and allows for the possibility of leaving the manifold, but only through it's boundary.

**Definition 1.** A set $\mathcal{M}$ is *locally invariant* under the flow from Eq. 3-4 if it has neighborhood $V$ so that no trajectory can leave $\mathcal{M}$ without also leaving $V$. In other words, it is locally invariant if for all $x \in \mathcal{M}, \varphi(x, [0, t] \subset V$ implies that $\varphi(x, [0, t]) \subset M$, similarly with $[0, t]$ replaced by $[t, 0]$ when $t < 0$.

**Definition 2.** A set $S$ is said to be *forward invariant* under a flow $\varphi_t$ if for every point $x$ in $S$ and for all $t \geq 0$, the image of $x$ under the flow at time $t$, denoted $\varphi_t(x)$, remains in $S$. This can be written as:

$$\varphi_t(x) \in S \quad \text{for all} \quad x \in S \quad \text{and} \quad t \geq 0.$$

**Small perturbation**   In the context of Fenichel's Persistence Theorem, a "sufficiently small $C^r$ perturbation of the vector field $f$" refers to perturbations that are small in the $C^r$ norm. The $C^r$ norm measures the size of a function and its derivatives up to order $r$.

Formally, let $f : \mathbb{R}^n \to \mathbb{R}^n$ be the original smooth vector field and let $\tilde{f}$ be a perturbed vector field. The perturbation $\tilde{f}$ is a sufficiently small $C^r$ perturbation if the difference $\tilde{f} - f$ has a small $C^r$ norm. Mathematically, this can be expressed as:

$$\left\| \tilde{f} - f \right\|_{C^r} < \epsilon,$$

where $\epsilon$ is a small positive number, and the $C^r$ norm is defined as:

$$\left\| \tilde{f} - f \right\|_{C^r} = \max_{0 \le k \le r} \sup_{x \in \mathbb{R}^n} \left\| D^k(\tilde{f}(x) - f(x)) \right\|.$$

Which gives a constraint on how much each of the $k$-th derivatives $D^k$ of the perturbed vector field can differ from the original.

In summary, a perturbation $\tilde{f}$ is considered sufficiently small in the $C^r$ sense if the difference between $\tilde{f}$ and $f$, along with their derivatives up to order $r$, is uniformly small across the domain. For $C^1$ perturbations, this defines a $C^1$ neighborhood of functions, within which the persistence of the manifold is ensured by Fenichel's theorem. For $r = 0$, we get the uniform norm (defined as $\|f\|)\infty := \sup(|f|)$, see also Sec.S6).

**Closeness**    Finally, to formalize what is meant by that the persistent invariant manifold is very close to the original continuous attractor, we need a notion of distance between the manifolds.

**Definition 3.**  Let $(M, d)$ be a metric space. For each pair of non-empty subsets $X \subset M$ and $Y \subset M$, the *Hausdorff distance* between $X$ and $Y$ is defined as

$$d_H(X, Y) := \max \left\{ \sup_{x \in X} d(x, Y), \ \sup_{y \in Y} d(X, y) \right\},$$

where $\sup$ represents the supremum operator, $\inf$ the infimum operator, and where

$$d(a, B) := \inf_{b \in B} d(a, b)$$

quantifies the distance from a point $a \in X$ to the subset $B \subseteq X$.

In conclusion, the Hausdorff distance provides a rigorous mathematical framework to quantify the "closeness" or "similarity" between two sets.

## S4.1   Fenichel's Persistent Manifold Theorem

This section will introduce the original Fenichel's Persistent Manifold Theorem, laying the groundwork for understanding how normally hyperbolic invariant manifolds persist under perturbations in systems with distinct time scales. By examining this foundational theorem, we can build a deeper understanding of the stability and behavior of complex dynamical systems.

In the study of dynamical systems, particularly those involving multiple time scales, understanding the behavior of solutions near invariant manifolds is crucial. These manifolds often determine the long-term dynamics of the system and their stability properties. The Fenichel's Persistent Manifold Theorem provides a powerful framework for analyzing such systems by demonstrating the persistence of normally hyperbolic invariant manifolds under small perturbations. This theorem is particularly relevant in systems where variables evolve on different time scales— typically referred to as "slow" and "fast" dynamics. By reformulating the system with a change of time-scale, we can explore how the dynamics on these manifolds behave, especially when perturbed. Consider the system given by Equations 3-4, which can be rewritten using a rescaled time variable, $\tau$, to distinguish between the fast and slow dynamics as follows:

$$\begin{cases} \epsilon x' & = g(x, y, \epsilon) \\ y' & = h(x, y, \epsilon) \end{cases} \tag{38}$$

where $' = \frac{d}{d\tau}$ and $\tau = {}^t/_\epsilon$. The time scale given by $\tau$ is said to be fast whereas that for $t$ is slow, as long as $\epsilon \ne 0$ the two systems are equivalent.

The functions $g$ and $h$ in Eq. 3-4 are both assumed to be $C^r$ (for $r > 0$ on a set $U \times I$ where $U \subset \mathbb{R}^d$ is open, and $I$ is an open interval, containing 0.

Suppose that the set $\mathcal{M}_0$ is a subset of the set $\{h(x, y, 0) = 0\}$ and is a compact manifold, possibly with boundary, and is normally hyperbolic relative to 38.

**Theorem 3** (Theorem 1 in [73]). *If $\epsilon > 0$, but sufficiently small, there exists a manifold $\mathcal{M}_\epsilon$ that lies within $\mathcal{O}(\epsilon)$ of $\mathcal{M}_0$ and is diffeomorphic to $\mathcal{M}_0$. Moreover it is locally invariant under the flow of Eq. 3-4, and $C^r$, including in $\epsilon$, for any $0 < r < \infty$. Finally, $\mathcal{M}_\epsilon$ has $\mathcal{O}(\epsilon)$ Hausdroff distance to $\mathcal{M}_0$ and has the same smoothness as $g$ and $h$.*

If the invariant manifold $\mathcal{M}_0$ is attractive, then the only way trajectories can escape the invariant set $\mathcal{M}_\epsilon$ after perturbation, is in negative time through the boundaries. This guarantees that the persistent manifold $\mathcal{M}_\epsilon$ is still attractive.

## S4.2 Fundamental limitations of the topology of bifurcated continuous attractors

### S4.2.1 Flow on a line

**Theorem 4** (One Dimensional Equivalence)**.** *Two flows and in are topologically equivalent if and only if their equilibria, ordered on the line, can be put into one-to-one correspondence and have the same topological type (sink, source or semistable).*

### S4.2.2 Flow on a ring

The flow on a ring can be described by the differential equation[129]:

$$\frac{d\theta}{dt} = f(\theta), \tag{39}$$

with $\theta \in [0, 2\pi]$.

The simplest type of flow on a ring is the uniform circular flow, where each point moves with a constant angular velocity $f(\theta) = \omega$. If $|f(\theta)| > 0$ for all $\theta$, there is a circular flow with a variable speed.

There is a fixed point for each unique $\theta$ for which $f(\theta) = 0$. The nature of the flow around these fixed points can be classified into:

- **Stable fixed points**: Points where the flow tends to as time progresses ($f(\theta) > 0$ for all $\theta \in [0, \theta] \cap V$ and $f(\theta) < 0$ for all $\theta \in [\theta, 2\pi] \cap V$ for some open ball $V$ around $\theta$).
- **Unstable fixed points**: Points from which the flow diverges ($f(\theta) < 0$ for all $\theta \in [0, \theta] \cap V$ and $f(\theta) > 0$ for all $\theta \in [\theta, 2\pi] \cap V$ for some open ball $V$ around $\theta$).

For a detailed discussion of how bifurcations of a continuous attractors depend on the symmetry of the continuous attractor, see for example The Equivariant Branching Lemma (Lemma 1.31 in[130]). If the solutions are symmetric under rotations (a circular symmetry such as for a ring attractor). As you change the bifurcation parameter, you find that new solutions appear that also respect this rotational symmetry. The lemma tells you that these new solutions will align with specific symmetries (like different rotation angles), which are described by the irreducible representations of the symmetry group.

### S4.3 Consequences to system identification

Fenichel's Persistence Theorem has several significant implications for modelling and system identification in dynamical systems. Because the theorem provides a guarantee that small perturbations in the system do not lead to significant changes in the qualitative behavior of the system, we can be (slightly) wrong for example about the exact nonlinearity of a neuron's transform function. For example, if neurons are only approximately ReLU the theory developed in[131] still holds (at the behaviorally relevant timescales). More generally, when reconstructing computational system dynamics to understand how cognitive functions are implemented in the brain, our theory shows that small deviations in the identified system can still lead to behaviorally equivalent models for neural computation[4].

## S5  Near Perfect Analog Memory Systems are close to Continuous Attractors

We will give some clarifications and proofs of the claims on systems near perfect analog memory systems.

### S5.1  Revival of the continuous attractor from a slow manifold

We provide a proof of Prop. 1 which is dependent on the $\epsilon$-Neighborhood Theorem[**], which we state here.

**Theorem 5** ($\epsilon$-Neighborhood Theorem[132]). *Let $Y \subseteq \mathbb{R}^n$ be a smooth, compact manifold. Then for a small enough $\epsilon > 0$, each $w \in Y_\epsilon := \{w \in \mathbb{R}^n \mid \exists y \in Y : \|y - w\| < \epsilon\}$ has a unique closest point in $Y$, denoted $\pi(w)$. Moreover, $\pi : Y_\epsilon \to Y$ is a submersion with $\pi|_Y = id$.*

We now prove Prop. 1.

*Proof.* It is sufficient to show that there is a perturbation $\mathbf{p}$ that has zero flow off of $\mathcal{M}_\epsilon$ but for which $\mathbf{f} + \mathbf{p} = 0$ on $\mathcal{M}_\epsilon$ for the full system $\mathbf{f}$ as defined in Eq. 2. Define

$$\mathbf{p}(\mathbf{x}) = \int_{\mathcal{M}_\epsilon} -\delta(\mathbf{x} - \mathbf{y}) \mathbf{f}(\mathbf{y}) d\mathbf{y},$$

with the Dirac delta function $\delta(\mathbf{x}) = \delta(x_1)\delta(x_2)\ldots\delta(x_d)$. It is then easy to check that $\mathbf{f}(\mathbf{x}) + \mathbf{p}(\mathbf{x}) = 0$ for all $\mathbf{x} \in \mathcal{M}_\epsilon$ and $\mathbf{f}(\mathbf{x}) + \mathbf{p}(\mathbf{x}) = \mathbf{f}(\mathbf{x})$ for all $\mathbf{x} \notin \mathcal{M}_\epsilon$. Hence, we have a continuous attractor at $\mathcal{M}_\epsilon$. If smoothness is important, we can construct the following perturbation. From the $\epsilon$-Neighborhood Theorem[133], we get that there exists a smooth positive function $\delta : \mathcal{M}_\epsilon \to \mathbb{R}^+$, such that if we let $N_\delta$ be the $\delta$-neighborhood of $\mathcal{M}_\epsilon$,

$$M_\epsilon := \{y \in \mathbb{R}^n : |y - x| < \delta(x) \text{ for some } x \in M_\epsilon\},$$

then each $y \in N_\delta$ possesses a unique closest point $\pi_\delta(y)$ in $\mathcal{M}_\epsilon$ with the map $\pi_\delta : \mathcal{M}_\delta \to \mathcal{M}_\epsilon$ being a submersion.

We can then define a bump function

$$\psi(y) = \begin{cases} \exp\left(-\frac{1}{1-(\pi_\delta(y)-y)^2}\right) & \text{if } y \in (-\delta - \pi_\delta(y), \delta - \pi_\delta(y)) \\ 0 & \text{otherwise} \end{cases} \tag{40}$$

Then the perturbation

$$\mathbf{p}(\mathbf{y}) = -\psi(\mathbf{y})\mathbf{f}(\pi_\delta(\mathbf{y}))$$

is smooth and creates a continuous attractor at $\mathcal{M}_\epsilon$. $\qquad\square$

### S5.2  Output mapping

This paper focuses on a linear output mapping for simplicity. Errors can be minimized with a nonlinear mapping, if there is an output that is mapped off of the output manifold, we can always adjust the output mapping to correct for this, if the space of output mappings is general enough. However, this can only be applied for errors off of the output manifold. If there is memory degradation along the output manifold, it is not possible to choose another mapping that corrects for this error. Therefore, we choose a linear output mapping for our analysis. In some cases, linear output mappings are found to support neural computation, for example for motion direction-discrimination[134].

### S5.3  Approximate solutions to an analog working memory problem

We will now discuss the conditions for when approximate solutions to an analog working memory problem are near a continuous attractor. We consider approximate solutions to an analog working memory problem to be systems of the form 61 or 64 (in both cases following a linear decoder), which have a small memory error over time in output space.

#### S5.3.1  Robustness

Noise, practically defined as unpredictable components of the system's behavior, comes from many sources. The concept of S- and D-type noise is based on[19].

**S-type robustness**    S-type noise encapsulates reversible changes in the neural state such that the deterministic part of the dynamics itself remains unchanged. Neural dynamics must be robust to perturbations and stimuli that push the neuronal activity away from the continuous attractor[135].

---

[**]Not to be confused with the perturbation parameter $\epsilon$.

**D-type robustness** D-type noise, in the space of recurrent network dynamics parameterized by the synaptic weights. This corresponds to slight changes to the ODE, i.e. perturbations. Previously, it robustness to D-type noise was considered to correspond to structurally stability [19].

Here we shortly discuss what we consider to be a necessary and sufficient condition for D-type robustness. We can use the concept of Lipschitz persistence to define D-type robustness. Mañe showed that if an invariant manifold is Lipschitz persistent then it must be normally hyperbolic [136]. To understand the concept of Lipschitz persistence, we need to define the Lipschitz section and Lipschitz constant.

**Definition 4.** Let $M$ be a $C^\infty$ boundaryless manifold and $V \subset M$ a $C^1$ compact boundaryless submanifold. Assume that $M$ is a submanifold of $\mathbb{R}^n$. Let $NV$ be a $C^1$ subbundle of $TM|_V$ satisfying $TV \oplus NV = TM|_V$. If $\eta$ is a section of $NV$, define the Lipschitz constant of $\eta$ by

$$\mathrm{Lip}(\eta) = \sup \left\{ \frac{\|\eta(x) - \eta(y)\|}{\|x - y\|} \mid x, y \in V, x \neq y \right\}.$$

We say that $\eta$ is a *Lipschitz section* if $\mathrm{Lip}(\eta) < +\infty$. Let $\Gamma_{\mathcal{L}}(NV)$ be the space of Lipschitz sections of $NV$ endowed with the norm

$$\|\eta\|_{\mathcal{L}} = \sup \{\|\eta(x)\| \mid x \in V\} + \mathrm{Lip}(\eta).$$

Let $\mathrm{Diff}^1(M)$ be the space of $C^1$ diffeomorphisms with the topology of the $C^1$ convergence on compact subsets.

**Definition 5.** Let $f \in \mathrm{Diff}^1(M)$. We say that $V$ is a Lipschitz persistent invariant manifold of $f$ if there exists a neighborhood $U$ of $V$ such that for all $\delta > 0$ there exists a neighborhood $\mathcal{U}_\delta$ of $f$ such that if $g \in \mathcal{U}_\delta$ there exists $\eta \in \Gamma_{\mathcal{L}}(NV)$ with $\|\eta\|_{\mathcal{L}} < \delta$ satisfying $V_g = \mathrm{graph}(\eta)$, where $\mathrm{graph}(\eta) = \{\exp_x(\eta(x)) \mid x \in V\}$, $V_g = g(U)$.

Observe that this definition implies $V_f = V$, hence $f(V) = V$. Moreover, the Lipschitz persistence is independent of the bundle $NV$.

For a flow $\varphi_t$ (coming from the solutions of an ODE) we can fix $t = \tau \in \mathbb{R}_{>0}$ so that we get a homeomorphism $\varphi_\tau$. This allows us to apply this result to apply to our case.

In the case where $V$ is a point, then it is a hyperbolic fixed point and the persistence follows trivially from the implicit function theorem.

*Remark* 1. It is sufficient to take a persistent manifold that is uniformly locally maximal. If $V$ is persistent and uniformly locally maximal, then $V$ has to be normally hyperbolic [136].

*Definition* 6 (Uniformly locally maximal). There exist neighborhoods $U$ of $V$ in $M$ and $U$ of $f$ in the space $\mathrm{Diff}^1(M)$ of $C^1$-diffeomorphisms of $M$, such that for any $g \in U$, $N_g = \bigcap_{k \in \mathbb{Z}} g^k(U)$ is a $C^1$-submanifold close to $V$, with $N_f = N$. The latter property implies the uniqueness of the invariant submanifold.

## S5.4 Near Perfect Analog Memory Systems are close to Decomposable Systems with a Continuous Attractor

We now prove a more general statement about the kind of systems that are close to perfect analog memory systems. The theory guarantees that a system that satisfies conditions (C1)-(C4) will have a continuous attractor in the following sense. For such a system there exists a decomposition such that the system can be effectively decomposed into a continuous attractor (attractive invariant manifold with zero flow) and a component on which a (possibly "fast" flow) can exist buy which get quenched by the out put projection. These additional dynamics orthogonal to decoding have been observed for motor movement preparation [137,138].

For this part of the theory, we need to consider the output manifold $\mathcal{M}_{\mathrm{output}}$, the manifold on which we determine the error over time as in Sec. S6. For this section, we will consider the output mapping to be a smooth (possibly nonlinear) mapping $g \colon X \to Y$ between the neural state space $X = \mathbb{R}^{d_X}$ and the output space $Y = \mathbb{R}^{d_Y}$. For a circular variable this will be the ring $S^1$.

The construction of a perturbation relies on finding the necessary minimal structure in the invariant manifold for which we can guarantee closeness to a continuous attractor. Therefore, first of all, we need this closeness in terms of the geometry of the manifold, which we guarantee through the notion of a fibration of the output mapping. Second, we need to guarantee that the flow is bounded in a sense so that our perturbation is also bounded by this amount. We will characterize this by the vector field normal to the fibers of the fiber bundle. A fiber bundle is a mathematical structure that allows us to study spaces that are locally like a product space but globally may have a different structure. So we first state the definition of a fiber bundle and a trivial fibration.

**Definition 7.** A *fiber bundle* is a structure $(E, B, \pi, F)$ where:

- $E$ is the *total space*,

- $B$ is the *base space*,

- $\pi : E \to B$ is a continuous surjection called the *projection map*, and

- $F$ is a topological space called the *fiber*.

This structure must satisfy the local triviality condition: for each $b \in B$, there exists an open neighborhood $U$ of $b$ such that there is a homeomorphism

$$\varphi : \pi^{-1}(U_b) \to U_b \times F_b$$

that commutes with the projection onto $U$, meaning that the following diagram commutes:

$$
\begin{array}{ccc}
\pi^{-1}(U_b) & \xrightarrow{\ \varphi\ } & U_b \times F_b \\
\pi \downarrow & & \downarrow \mathrm{pr}_1 \\
U_b & \xrightarrow{\ id\ } & U_b
\end{array}
$$

where $\mathrm{pr}_1 : U_b \times F_b \to U_b$ is the projection onto the first factor.

So, around every point in the base space, you can "zoom in" and see that the bundle looks like a straightforward product of the base space and the fiber and $\varphi$ is providing this trivialization.

**Definition 8.** We say that a projection map $\pi : E \to B$ is *locally trivial* if each point $b \in B$ is contained in an open set $U$ having the property that $E_U := \pi^{-1}(U)$ is trivial over $U$.

We will rely on the concept of a submersion to characterize how the output space needs to relate to our invariant manifold.

**Definition 9** (Submersion). Let $M$ and $N$ be differentiable manifolds and $f : M \to N$ be a differentiable map between them. The map $f$ is a *submersion* at a point $p \in M$ if its differential $Df_p : T_pM \to T_{f(p)}N$ is a surjective linear map.

This allows us to characterize what kind of structure the invariant manifold needs to have, namely it can be see as the direct product of the output manifold and some other manifold which described over what part of the invariant manifold the output mapping is invariant.

**Theorem 6** (Ehresmann's lemma[139]). *If a smooth mapping $f : M \to N$, where $M$ and $N$ are smooth manifolds, is*

1. *a surjective submersion, and*

2. *a proper map*

*then it is a locally trivial fibration.*

*Remark* 2. If a manifold $M$ is compact, then the above smooth map is a proper map. If we do not have a submersion or that the output mapping is transversal to the output manifold, we have a situation in which the flow on the invariant manifold can be arbitrarily fast (even though memory is degrading slowly). This happens when the flow is in a singularity of the output mapping.

For our statement we want that the invariant manifold can be decomposed into a space that is diffeomorphic to the output manifold and another compact manifold: $\mathcal{M}_\epsilon = \mathcal{M}_{slow} \times \mathcal{M}_{null}$ with $\mathcal{M}_{slow} \simeq \mathcal{M}_Y$. We can relax this to allow for the possibility of some torsion along the invariant manifold. For this to hold, it is sufficient that the output mapping must be a submersion because this makes it a locally trivial fibration. So we get that $g : X \to Y$ defines a locally trivial fiber mapping.

The second assumption we need is to have a bound for the speed of trajectories along the fibers, which will correspond to a speed along the output manifold, resulting in memory degradation. We need to assume that there is a slow flow (in the direction of $\mathcal{M}_{slow}$). We characterize the relevant maximal size of the vector field that needs to be perturbed as the supremum over the uniform norms of the vector field normal to the fibers of the fiber bundle.

**Theorem 7.** *Let $\mathcal{M}_\epsilon$ be a connected, compact, normally hyperbolic slow manifold (as parametrized by Eq. (3)-(4)) with the real part of the eigenvalues of $\nabla_\mathbf{z}\mathbf{h}$ all negative. Further, assume that this manifold can be decomposed $\mathcal{M}_\epsilon = \mathcal{M}_{slow} \times \mathcal{M}_{null}$ with $\mathcal{M}_{slow} \simeq \mathcal{M}_Y$ and that the uniform norm of the flow tangent to $\mathcal{M}_\epsilon$ restricted to $\mathcal{M}_{slow}$ be $\|(\dot{\mathbf{y}})_{slow}\|_\infty = \eta$. Then, there exists a perturbation with uniform norm at most $\eta$ that induces a bifurcation to a system that is decomposable into a continuous attractor and a system with a non-zero flow.*

In other words, after applying this perturbation, the slow component of the perturbed system satisfies $\dot{\mathbf{x}}'|_{slow} = 0$ for the system $\dot{\mathbf{x}}' = \mathbf{f}(\mathbf{x}) + \mathbf{p}(\mathbf{x})$. Furthermore, the trajectories of the resulting system form a fiber bundle where the output projection serves as the bundle projection. Each fiber consists of trajectories that are mapped to the same value in $\mathcal{M}_{output}$, meaning that the fibers describe an invariance under the output projection.

*Proof.* Assume that the output mapping is a submersion from the invariant manifold $\mathcal{M}_\epsilon$ to the output manifold $\mathcal{M}_{\text{output}}$. Ehresmann's lemma implies that this implies that we have a locally trivial fibration $(\mathcal{M}_\epsilon, \mathcal{M}_{\text{output}}, g, \mathcal{M}_{\text{null}})$. We construct the perturbation **p** as the part of the vector field that us normal to the fibers $g^{-1}(y)$ for each $y \in \mathcal{M}_{\text{output}}$. By construction, this perturbation has uniform norm at most $\eta$. This perturbation makes the vector field normal to each fiber zero. That implies that each fiber is an invariant submanifold of $\mathcal{M}_\epsilon$. From the structure of the fiber bundle it follows that there is a continuum of such invariant submanifolds. Hence, the perturbed system is decomposable into a system that is a continuous attractor and a system with a non-zero flow. $\square$

**Remark** This perturbation can be made smoothness, along similar lines as above. We can take a $\epsilon$-Neighborhood around the invariant manifold on which we extend the above vector field with bump functions to get a smooth vector field that still results in an attractive invariant manifold.

**Example** An example of an approximate continuous attractor solution of the form with a decomposable system of which one of the subsystems is close to a continuous attractor is the torus solution in Fig. 4D. In this case, the system can be perturbed slightly such that there exists a continuum of limit cycles laid out over a ring.

# S6 Upper bound for the memory performance on a short time scale

We will now formalize the statement about an upper bound dependent on the uniform norm of the vector field on the slow manifold in Sec.3.2 and provide a proof.

**Proposition 2.** *Let $\mathcal{M}$ be a normally hyperbolic slow manifold as in Prop. 3.2. Let $\mathbf{x}_0 \in \mathcal{M}$, and $\varphi = \mathbf{f}|_{\mathcal{M}}$ be the flow restricted to the manifold. The average deviation from the initial memory $\mathbf{x}_0$ over time is bounded linearly*

$$\frac{1}{\operatorname{vol}\mathcal{M}} \int_{\mathcal{M}} |\mathbf{x}(t, \mathbf{x}_0) - \mathbf{x}_0| \, \mathrm{d}\mathbf{x}_0 \leq t \, \|\varphi\|_\infty. \tag{41}$$

*Proof.* Numerical integration of the ODE gives

$$x(t, \mathbf{x}_0) = \int_0^t \varphi(x(\tau)) d\tau + \mathbf{x}_0$$

$$\leq \int_0^t \|\varphi\|_\infty d\tau + \mathbf{x}_0$$

$$= t\|\varphi\|_\infty + \mathbf{x}_0$$

From this, we get

$$\frac{1}{\operatorname{vol}\mathcal{M}} \int_{\mathcal{M}} |\mathbf{x}(t, \mathbf{x}_0) - \mathbf{x}_0| \, \mathrm{d}\mathbf{x}_0 \leq \frac{1}{\operatorname{vol}\mathcal{M}} \int_{\mathcal{M}} t\|\varphi\|_\infty$$

$$= t \, \|\varphi\|_\infty.$$

$\square$

We formulate here a theory of continuous attractor approximations in terms of memory loss over time. It can be used uniform norm of vector field on the manifold to bound the memory performance on the short-time scale. Let $\mathbf{x}_0 \in \mathcal{M}$, and $\varphi = \mathbf{p}|_{\mathcal{M}}$ be the flow restricted to the manifold. We will show that the average deviation from the initial memory $\mathbf{x}_0$ over time is bounded linearly as in Eq. 5.

## S6.1 Ring attractor

For a ring attractor we can give more tight bounds on the accumulated error for the angular memory. Suppose we have a dynamical system $\boldsymbol{x} \in \mathbb{R}^N$ with autonomous dynamics $\dot{\boldsymbol{x}} = F_{\boldsymbol{\theta}}(\boldsymbol{x})$ and solutions $\mathbf{x}(t, \mathbf{x}_0)$ uniquely defined for each $\mathbf{x}_0$. Let us define the error (i.e. the average deviation from the initial memory) for an attractor as

$$\mathcal{L}(T) := \frac{1}{\operatorname{vol}\mathcal{M}} \int_{\mathcal{M}} |\mathbf{x}(t, \mathbf{x}_0) - \mathbf{x}_0| \, \mathrm{d}\mathbf{x}_0 \tag{42}$$

If we further assume that the memory is not simply the state of the network, we need to take into consideration a decoder of the memory. Suppose that there is an invertible decoder mapping $f : \mathcal{U} \to \mathbb{R}^N$. For a ring variable, we can take this to be the projection onto the plane $\mathbf{W}_{\text{out}}$ and then applying the $\operatorname{arctan}$:

$$g(x) = \operatorname{arctan}(\mathbf{W}_{\text{out}}x). \tag{43}$$

In the case of the ring attractor, the memory we would like to encode is $\alpha \in \mathcal{U} = [0, 2\pi)$, the error is defined as $|x - y|_o = o_\pi(|x - y|)$ where:

$$o_\pi(x) = \begin{cases} x & \text{if } x < \pi \\ 2\pi - x & \text{if } x \geq \pi \end{cases} \tag{44}$$

If we call $\hat{\alpha}_{\boldsymbol{\theta}}(\alpha_0, t) = g(\varphi_{\boldsymbol{\theta}}(f(\alpha), t))$ we get the expression of the memory loss for this kind of memory as:

$$\mathcal{L}(T) = \frac{1}{2\pi} \int_0^{2\pi} (|\hat{\alpha}_{\boldsymbol{\theta}}(\alpha_0, t) - \alpha_0|) \, d\alpha_0 \tag{45}$$

**General bounds** Define the following functions:

$$\epsilon^+(t) = \sup_{\alpha_0} o_\pi(|\hat{\alpha}_{\boldsymbol{\theta}}(\alpha_0, t) - \alpha_0|) \geq \frac{1}{2\pi} \int_0^{2\pi} o_\pi(|\hat{\alpha}_{\boldsymbol{\theta}}(\alpha_0, t) - \alpha_0|)$$

$$\epsilon^m(t) = \frac{1}{2\pi} \int_0^{2\pi} o_\pi(|\hat{\alpha}_{\boldsymbol{\theta}}(\alpha_0, t) - \alpha_0|) \, d\alpha_0 \tag{46}$$

$$\epsilon^-(t) = \inf_{\alpha_0} o_\pi(|\hat{\alpha}_{\boldsymbol{\theta}}(\alpha_0, t) - \alpha_0|) \leq \frac{1}{2\pi} \int_0^{2\pi} o_\pi(|\hat{\alpha}_{\boldsymbol{\theta}}(\alpha_0, t) - \alpha_0|)$$

Then we get the bounds for the loss $\mathcal{L}(T)$ as:

$$\frac{1}{T}\int_0^T \epsilon^-(t)dt \leq \mathcal{L}(T) = \frac{1}{T}\int_0^T \epsilon^m(t)dt \leq \frac{1}{T}\int_0^T \epsilon^+(t)dt \tag{47}$$

**Speed bounds**

We can define the maximum, average and minimum memory error speed as:

$$v_{\epsilon+} = \sup_{\alpha_0} \frac{d}{dt}(o_\pi(|\hat{\alpha}_\theta(\alpha_0, t) - \alpha_0|))|_{t=0}$$

$$v_{\epsilon m} = \frac{1}{2\pi}\int_0^{2\pi} \frac{d}{dt}(o_\pi(|\hat{\alpha}_\theta(\alpha_0, t) - \alpha_0|))|_{t=0}d\alpha_0 \tag{48}$$

$$v_{\epsilon-} = \inf_{\alpha_0} \frac{d}{dt}(o_\pi(|\hat{\alpha}_\theta(\alpha_0, t) - \alpha_0|))|_{t=0}$$

Notice then that since:

$$\epsilon^+(t) \leq \min(tv_{\epsilon+}, \pi)$$
$$\epsilon^-(t) \geq tv_{\epsilon-} \tag{49}$$

then,

$$\frac{1}{T}\int_0^T \epsilon^+(t)dt \leq \min\left(\frac{1}{T}\int_0^T tv_{\epsilon+}dt, \pi\right) = \min\left(\frac{Tv_{\epsilon+}}{2}, \pi\right)$$

$$\frac{1}{T}\int_0^T \epsilon^-(t)dt \geq \frac{1}{T}\int_0^T tv_{\epsilon-}dt = \frac{Tv_{\epsilon-}}{2} \tag{50}$$

and we get:

$$\frac{Tv_{\epsilon-}}{2} \leq \frac{1}{T}\int_0^T \epsilon^-(t)dt \leq \mathcal{L}(T) = \frac{1}{T}\int_0^T \epsilon^m(t)dt \leq \frac{1}{T}\int_0^T \epsilon^+(t)dt \leq \min\left(\frac{Tv_{\epsilon+}}{2}, \pi\right) \tag{51}$$

Finally, if the error is uniform enough we can expect $\epsilon^m(t) \approx tv_{\epsilon m}$ and

$$\mathcal{L}(T) = \frac{1}{T}\int_0^T \epsilon^m(t)dt \approx \frac{1}{T}\int_0^T tv_{\epsilon m}dt = \frac{Tv_{\epsilon m}}{2} \tag{52}$$

**Within manifold case** Let's assume that we have managed the system $F_\theta$ have a slow manifold $\mathcal{M} \in \mathbb{R}^N$ in bijection with $\mathcal{U}$, i.e. $f|_\mathcal{M}$ is not a mapping but a bijective function and:

$$\forall \boldsymbol{x} \in \mathcal{M} \quad \dot{\boldsymbol{x}} = \epsilon_\theta(\boldsymbol{x})\frac{\frac{\partial f}{\partial \alpha}(f^{-1}(\boldsymbol{x}))}{||\frac{\partial f}{\partial \alpha}(f^{-1}(\boldsymbol{x}))||} \tag{53}$$

Then we have a slow manifold in the form of a ring attractor, we have $f(0) = f(2\pi)$ and:

$$\hat{\alpha}_\theta(\alpha_0, t) = \left(\alpha_0 + \int_0^t \epsilon_\theta(\alpha_0, s)ds\right) \mod 2\pi \tag{54}$$

Then:

$$\hat{\alpha}_\theta(\alpha_0, t) = o_\pi\left(\left|\left(\alpha_0 + \int_0^t \epsilon_\theta(\alpha_0, s)ds\right) \mod 2\pi - \alpha_0\right|\right)$$

$$= o_\pi\left(\left|\left(\alpha_0 + \int_0^t \epsilon_\theta(\alpha_0, s)ds\right) \mod 2\pi - \alpha_0 \mod 2\pi\right|\right) \tag{55}$$

$$= o_\pi\left(\left|\int_0^t \epsilon_\theta(\alpha_0, s)ds \mod 2\pi\right|\right)$$

where we used that $\alpha_0 \in [0, 2\pi) \Rightarrow \alpha_0 = \alpha_0 \mod 2\pi$ and that $|x \mod 2\pi - y \mod 2\pi| = |(x - y) \mod 2\pi|$.

The final equation of the loss in this case has the form:

$$\mathcal{L}(T) = \frac{1}{2\pi}\int_0^{2\pi} \frac{1}{T}\int_0^T o_\pi\left(\left|\int_0^t \epsilon_\theta(\alpha_0, s)ds \mod 2\pi\right|\right) dt d\alpha_0. \tag{56}$$

**Slow manifold bounds**

In this case, if we have $N$ fixed points in the ring-like slow manifold, we know that:

$$\epsilon^+(t) \leq \min\left(\frac{2\pi}{N}, \pi\right), \tag{57}$$

and therefore:

$$\mathcal{L}(T) \leq \min\left(\frac{2\pi}{N}, \pi\right). \tag{58}$$

# S7 Slow manifold in trained RNNs

We will provide a detailed description of the tasks, architectures, training methods, and analysis techniques used in our numerical experiments with trained RNNs.

## S7.1 Tasks

**Memory guided saccade task**    The total time length of a trial is 512 steps. The time delay to the output cue was sampled from

$$T_{delay} \sim \mathcal{U}(50, 400). \tag{59}$$

We applied a mask $m_{i,t} = 0$ for 5 time steps ($t = T_{delay} + j$ for $j = 0, \ldots 4$) after the go cue (Eq. 65).

**Angular velocity integration task**    The time length of a trial is 256 steps. The input is an angular velocity and the target output is the sine and cosine of the integrated angular velocity. Velocity at every timestep is sampled from as a Gaussian Process (GP) for smooth movement trajectories, consistent with the observed animal behavior in flies and rodents.

$$k(x, y) = \exp\left(-\frac{\|x - y\|}{2\ell^2}\right), \tag{60}$$

with length scale $\ell$. The length scale of the kernel was fixed at 1.

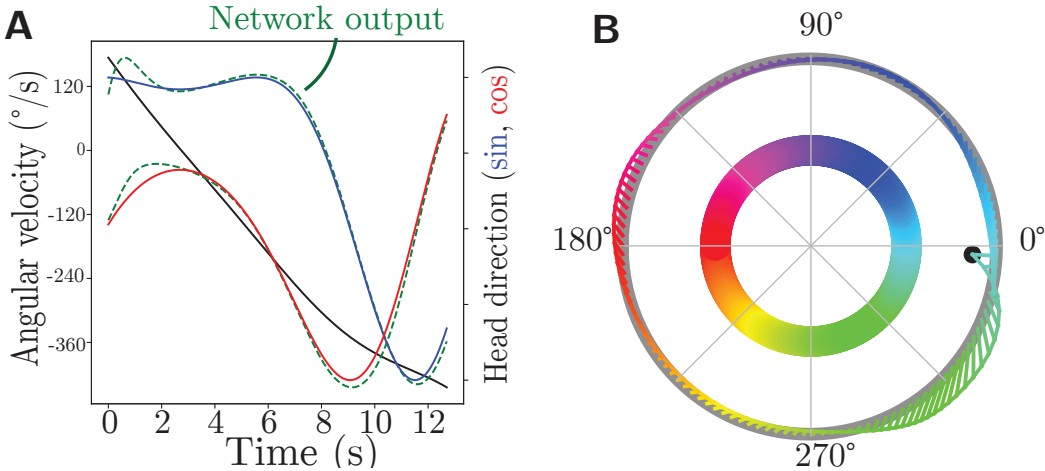

Figure S14: Description of the angular velocity integration task. (A) The angular velocity integration task. (B) The output of the angular velocity integration in the output space, color coded according to the integrated angle. An example of an input is shown with constant velocity and it is provided until one turn is completed.

**Double angular velocity integration task**    The Double Angular Velocity Integration Task is an extension of the Angular Velocity Integration Task, where two independent instances of the task are performed simultaneously. In this case, you have two separate angular velocities, each sampled from its own Gaussian Process, representing two distinct movement trajectories. For each of the two angular velocities, the integration over time is performed separately, resulting in two sets of outputs: one for each angular velocity, making the output space four dimensional.

Grid cells are known to exhibit periodic firing patterns that form a hexagonal grid across an environment, and these patterns are often modeled as existing on a toroidal surface (a doughnut-shaped surface)[140]. The reason for this is that the activity patterns of grid cells are continuous and wrap around seamlessly, meaning that if you move far enough in one direction, the grid pattern will repeat itself. This toroidal structure allows for the continuous representation of space without boundaries, which is crucial for efficient path integration. In the context of the Double Angular Velocity Integration Task, where two independent angular velocities are integrated, the resulting four-dimensional output space can be considered as two 2D subspaces (one for each angular velocity).

**Unbounded tasks**    Regarding tasks with an unbounded range, such as navigation tasks, two points bear mentioning. For planar attractors are diffeomorphic to $\mathbb{R}^2$, note that they do not conform to the assumptions on normally hyperbolic invariant manifolds, since $\mathbb{R}^2$ is not compact. There are suitable generalizations of this theory to noncompact manifolds[74], but we do not pursue them since they require more refined tools, which would only obscure the point that we are trying to make. Tangentially, we would also like to point out that we

assume that neural dynamics are naturally bounded (e.g. by energy constraints) and hence sufficiently well described by compact invariant manifolds.

## S7.2 Output projection of the invariant manifold

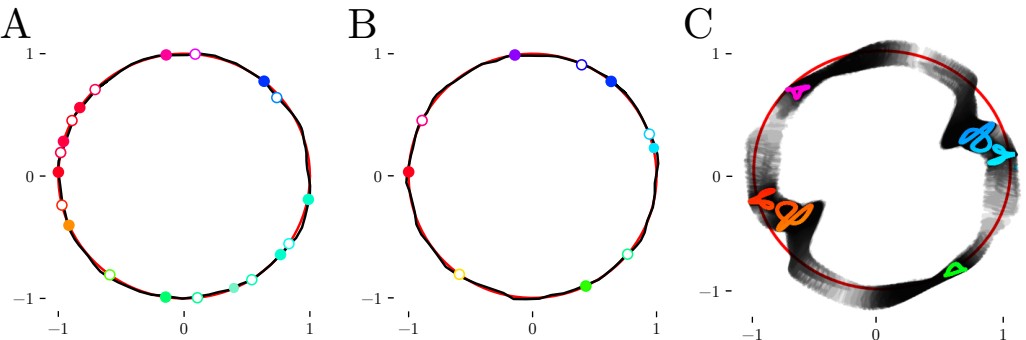

Figure S15: Output projection of slow manifold approximation of the trained networks of Fig. 4. All stability strutures are colored according to the decoded angle shown in Fig. S14B, target output circle shown in red. (A) An example fixed-point type solution to the memory-guided saccade task (Fig. 4B). (B) An example of a found solution to the angular velocity integration task (Fig. 4C). (C) An example slow-torus type solution to the memory-guided saccade task. The colored curves indicate stable limit cycles of the system (Fig. 4D).

The output projection of the invariant manifolds and stability structures (fixed points and limit cycles) is very close to the target output circle, as shown in Fig. S15.

## S7.3 Discretization

Computational neuroscientists often train RNNs as models of neural computation and interpret them as dynamical systems[36,87,101]. Our experiments connect to existing literature. We examined vanilla continuous-time RNNs for this work. The time-discretized version of RNN network activity is given by

$$
\begin{aligned}
\mathbf{x}_t &= \phi(\mathbf{W}_{\text{in}}\mathbf{I}_t + \mathbf{W}\mathbf{x}_{t-1} + \mathbf{b}) + \zeta_t \\
\mathbf{y}_t &= \mathbf{W}_{\text{out}}\mathbf{x}_t + \mathbf{b}_{\text{out}}
\end{aligned}
\tag{61}
$$

where $\mathbf{x}_t \in \mathbb{R}^d$ is the hidden state, $\mathbf{y}_t$ is the readout, $\mathbf{I}_t \in \mathbb{R}^K$ is the input, $\phi \colon \mathbb{R} \to \mathbb{R}$ is an activation function that acts on each of the hidden dimensions, $\zeta_t \overset{iid}{\sim} \mathcal{N}(\mathbf{0}, \sigma^2 = 1/100\mathbb{I})$ is a state noise variable, and $\mathbf{W}, \mathbf{b}, \mathbf{W}_{\text{in}}, \mathbf{W}_{\text{out}}, \mathbf{b}_{\text{out}}$ are parameters. We will shortly explain the discretization procedure, i.e., the steps for going from Eq. 64 to Eq. 61. Let $t_n = n\Delta t$.

The Euler-Maruyama method for a stochastic differential equation (Eq. 64)

$$
d\mathbf{x} = (-\mathbf{x} + \phi(\mathbf{W}_{\text{in}}\mathbf{I}(t) + \mathbf{W}\mathbf{x} + \mathbf{b}))\, dt + \sigma d W_t
$$

is given by :

$$
\mathbf{x}_{n+1} = \mathbf{x}_n + (-\mathbf{x}_n + \phi(\mathbf{W}_{\text{in}}\mathbf{I}_n + \mathbf{W}\mathbf{x}_n + \mathbf{b}))\, \Delta t + \sigma \Delta W_n,
$$

with $\Delta W_n = W_{(n+1)\Delta t} - W_{n\Delta t} \sim \mathcal{N}(0, \Delta t)$.

Now subsitute $\Delta t = 1$:

$$
\begin{aligned}
\mathbf{x}_{t+1} &= \mathbf{x}_t + (-\mathbf{x}_t + \phi(\mathbf{W}_{\text{in}}\mathbf{I}_t + \mathbf{W}\mathbf{x}_t + \mathbf{b})) + \sigma \Delta W_t, \tag{62} \\
&= \phi(\mathbf{W}_{\text{in}}\mathbf{I}_t + \mathbf{W}\mathbf{x}_t + \mathbf{b}) + \sigma \Delta W_t. \tag{63}
\end{aligned}
$$

If we introduce the noise term $\zeta_t = \sigma \Delta W_t$, which represents the discrete-time noise, we have derived the discrete-time equation:

$$
\mathbf{x}_t = \phi(\mathbf{W}_{\text{in}}\mathbf{I}_t + \mathbf{W}\mathbf{x}_{t-1} + \mathbf{b}) + \zeta_t.
$$

So, assuming Euler-Maruyama integration with unit time step, the discrete-time RNN of (61) corresponds to the stochastic differential equation:

$$
d\mathbf{x} = -\mathbf{x}\, dt + \phi(\mathbf{W}_{\text{in}}\mathbf{I}(t) + \mathbf{W}\mathbf{x} + \mathbf{b})\, dt + \sigma\, dW. \tag{64}
$$

where $dW$ is a Wiener process that models the intrinsic state noise in the brain. See for more detail on correspondences between discrete- and continuous-time RNNs in [141] and [142]. Our experiments connect to existing literature. In future studies, it would be interesting to perform experiments with Neural SDEs [143]

## S7.4 Network architectures

In all network architectures a linear output is used. Furthermore, for the angular velocity integration tasks we used an additional mapping from the output to the hidden layer to initialize the hidden state on the initial position along the ring from which the network needed to integrate from.

**Vanilla**    We used vanilla RNNs with different nonlinearities (ReLU, tanh and rectified tanh) for the recurrent layer.

**LSTM**    The number of units for the trained LSTMs was half of that of vanilla RNNs to match the number of paramters [144].

**GRU**    We also trained Gated Recurrent Units (GRU) [145] for which we used the same number of hidden units as the vanilla RNNs.

## S7.5 Training methods

We trained RNNs with PyTorch [146] on the three tasks Fig. S14. For the vanilla RNNs, we used a time step of $\Delta t = 0.1$. The parameters were initialized were initialized using the Xavier normal distribution [147]. For the recurrent weights we used $W_{ij} \sim \mathcal{N}(0, g/\sqrt{N})$ with a high gain $g = 1.5$. The initial hidden state was initialized using the output to recurrent mapping matrix $W_{otr} \colon \mathbb{R}^2 \to \mathbb{R}^N$ which was trained together with the other parameters.

Adam optimization with $\beta_1 = 0.9$ and $\beta_2 = 0.999$ was employed with a batch size of 64 and training was run for 5000 gradient updates. The batches were generated on-line, similar to how animals are trained with a new trial instead of iterating through a dataset of trials.

The best learning rate $10^{-2}$ was chosen from a set of values $\{10^{-2}, 10^{-3}, 10^{-4}, 10^{-5}\}$ by 5 initial runs for all nonlinearity and size pairings with the lowest average loss after 100 gradient steps. Training a single network took around 10 minutes on a CPU and occupied 10 percent of an 8GB RAM.

We numerically minimized the loss $L$ which was the mean squared error (MSE) between the network output $\mathbf{y}(t)$ and the target output $\hat{\mathbf{y}(t)}$:

$$L_{MSE} \coloneqq \langle m_{i,t}(y_{i,t} - \hat{y}_{i,t})^2 \rangle_{i,t}, \tag{65}$$

with a mask $m_{i,t}$ with $i$ the index of the output units and $t$ the index for time. We implemented a mask, $m_{i,t}$, for modulating the loss with respect to certain time intervals for the memory guided saccade task (see Sec. S7.1).

Although some of the models did not learn the task, most networks converged to a loss below $10^{-2}$ (Fig. S16).

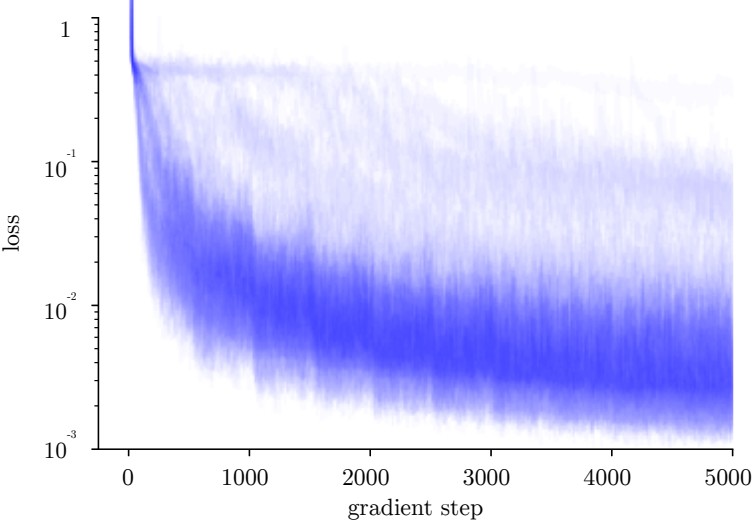

Figure S16: Training loss across gradient steps.

## S7.6 RNN analysis methods

### S7.6.1 Evaluation Metric

After training, we report the normalized mean squared error (NMSE) to asses the how good a found solution is:

$$\text{NMSE} = \frac{\mathbb{E}[(y - \hat{y})^2]}{\mathbb{E}[y]}, \tag{66}$$

where $y$ is the target and $\hat{y}$ is the prediction.

### S7.6.2 Asymptotic behavior and memory capacity

We determine the location of the fixed points through the local flow direction criterion as described in Sec. 4.2 and determine the basin of attraction

$$\text{Basin}(x^*) \coloneqq \{x \in \mathcal{M} \mid \lim_{t \to \infty} \varphi(t, x) = \{x^*\}\}. \tag{67}$$

through assesing the local flow direction for 1024 sample points in the found invariant manifold.

We construct a probability distribution of what part of state space we end up in an infinite time through the calculation of the size of the basins of attraction of stable fixed points as a proportion of the ring. We characterize the memory capacity of the network by calculating the entropy of this probability distribution of the network.

This follows from the following observation. If we assume that the angular variable that needs to be encoded $X$ is uniformly distributed and the encoding $Y$ is distributed according to the histogram given by the asymptotic behavior of the networks (i.e., the fixed points), then the *memory capacity* as the negative conditional entropy of the continuous memory given the asymptotic state, i.e.,

$$-H(X|Y) = \sum_{y \in Y} p(y) \int_{x \in \text{Basin}(y)} p(x|y) \log p(x|y) = \sum_{y \in Y} \text{vol Basin}(y) \log(\text{vol Basin}(y)). \tag{68}$$

### S7.6.3 Fast-slow decomposition

We simulated 1024 trajectories without noise with inputs from the task and let the networks evolve for 16 times the task definition lengths. We took the cutoff to identify the slow manifold to be $10^{-3}$ of the highest speed along each trajectory. We believe that this guarantees the identification of the slow manifold in a system that has a fast-slow decomposition. We sampled 1024 points from these points to fit a periodic, cubic spline (black line in Fig. 4, S19, S20).

**Finding fixed points** We then find fixed points by identifying where the flow reverses by sampling the direction of the local flow for 1024 sample points along the found invariant manifold. We assess the direction through projection onto the output mapping and calculate the angular flow. If the flow is pointing towards a point where the flow reverses then we consider there to be a stable fixed point. If the flow s pointing away from a reversal point then we consider the fixed point there to be a saddle. We find that long integrated trajectories of the network converge to the found stable fixed points through this independent method.

**Eigenspectrum along the invariant manifold** We use the eigenvalue spectrum as evidence for normal hyperbolicity. Normal hyperbolicity of an attractive ring invariant manifold implies that the eigenvalue spectrum has a gap in its eigenvalue spectrum. To measure this, we linearize at reference points on the invariant manifold (calculate the Jacobian) and calculate the eigenvalues. The largest eigenvalue (real part) for such a manifold needs to be much closer to zero than the second largest. For LSTMs and GRUs the eigenspectrum was approximated by autodifferentiation of the networks w.r.t. the states on the identified invariant manifold.

For a stable system, where the eigenvalues have negative real parts, the time constant $\tau$ is given by the negative inverse of the eigenvalue's real part: $\tau = -\frac{1}{\Re(\lambda)}$, where $\Re(\lambda)$ denotes the real part of the eigenvalue $\lambda$. For the two example networks in Fig.4, there is a time scale separation between the dynamics on and off the invariant manifold because there is only one eigenvalue close to zero.

**Vector field on invariant manifold** We assess the vector field for the ODE (Eq. 64) without noise and input) on the found invariant manifold $\mathcal{M}$ by calculating it in the state space and then projecting it onto the output space:

$$\dot{\alpha} = \mathbf{W}_{\text{out}} f(\hat{\alpha}) \tag{69}$$

for sampled points $\hat{\alpha} \in \mathcal{M}$. These points $\hat{\alpha} \in \mathcal{M}$ on the manifold are associated with the points on the ring through the mapping $\alpha = \mathbf{W}_{\text{out}} \hat{\alpha}$.

This vector field in the output space captures in what direction and how quickly angular memory will decay. The vector field suggests that the system indeed has an invariant manifold (Fig. S17). Furthermore, the vector field and fixed points are consistent with each other, as the vector field flips direction around found fixed points.

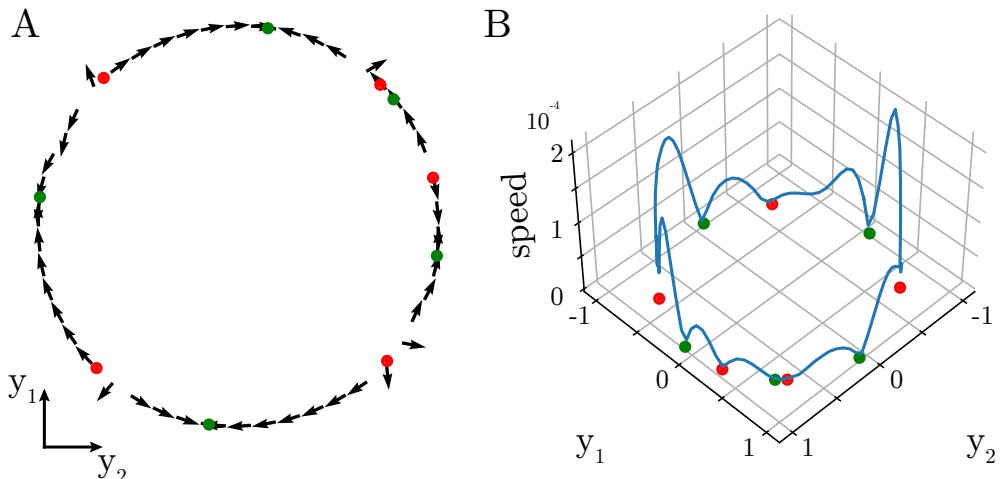

Figure S17: The projected vector field on the found invariant manifold for the system Fig 4C. (A) The found vector field aligns well with the ring in the projected output space. (B) The norm of the vector field is low around found fixed points as expected, but is higher for points that are just slow points.

There are some inconsistencies around saddle nodes, where the vector field seems to point off of the manifold. This is probably just inaccuracies coming from numerically calculating the vector field and the exact location of the invariant manifold. For the bound discussed in Sec. 3.2, we calculate the uniform norm of the found vector field

$$\|f\|_\infty = \sup_\alpha \mathbf{W}_{\text{out}} f(\alpha), \tag{70}$$

see also Sec. S6.

For LSTMs and GRUs the vector field was approximated by taking the difference the initial and the next state after initializing the network from states on the identified invariant manifold.

## S7.7 LSTM and GRU results

The trained LSTMs and GRUs share the same pattern observed in the trained vanilla RNNs: a ring slow invariant manifold (Fig. S18C and D). The fixed point topologies in the LSTM and GRU networks show a lot of variation in the number of fixed points, paralleling the systems adjacent to continuous attractors from Fig. 1, as seen in Figure 5D. These variations are similar to those discussed in Figures S18B. Additionally, the angular error and memory capacity measures across different time scales are comparable to those illustrated in Figure S18A, highlighting the generalization properties influenced by the topology of the solutions. These results underline the universality of our findings beyond vanilla RNNs.

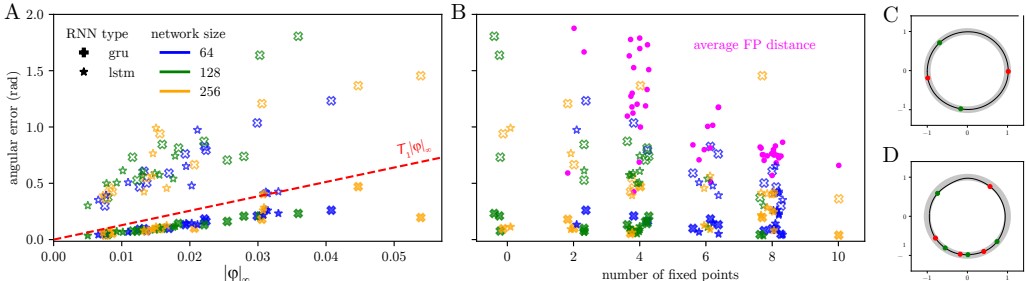

Figure S18: The different measures for memory capacity reflect the generalization properties implied by the topology of the found solution. **(A)** The average accumulated angular error vs. the uniform norm on the vector field shown. Angular error at $T_1 =$ trial length (filled markers) and $\lim T_1 \to \infty$ (hollow markers). Points are jittered to aid legibility. **(B)** The number of fixed points vs. average accumulated angular error, with the average distance between neighboring fixed points (magenta). **(C,D)** Invariant manifold (black) of a trained LSTM (C) and GRU (D) with stable fixed points (green) and saddle nodes (red).

## S7.8 Identified invariant manifold output projections

The identified invariant manifolds in the trained networks (Fig. S19 and Fig. S20). However, not all solutions can be meaningfully analyzed with the slow-fast decomposition method. For example, the solution at the center of the $\tanh$, $N = 256$ block, the found invariant manifold is not correctly captured. This is true for the networks that have not learned the task correctly (networks with a NMSE higher than -20dB).

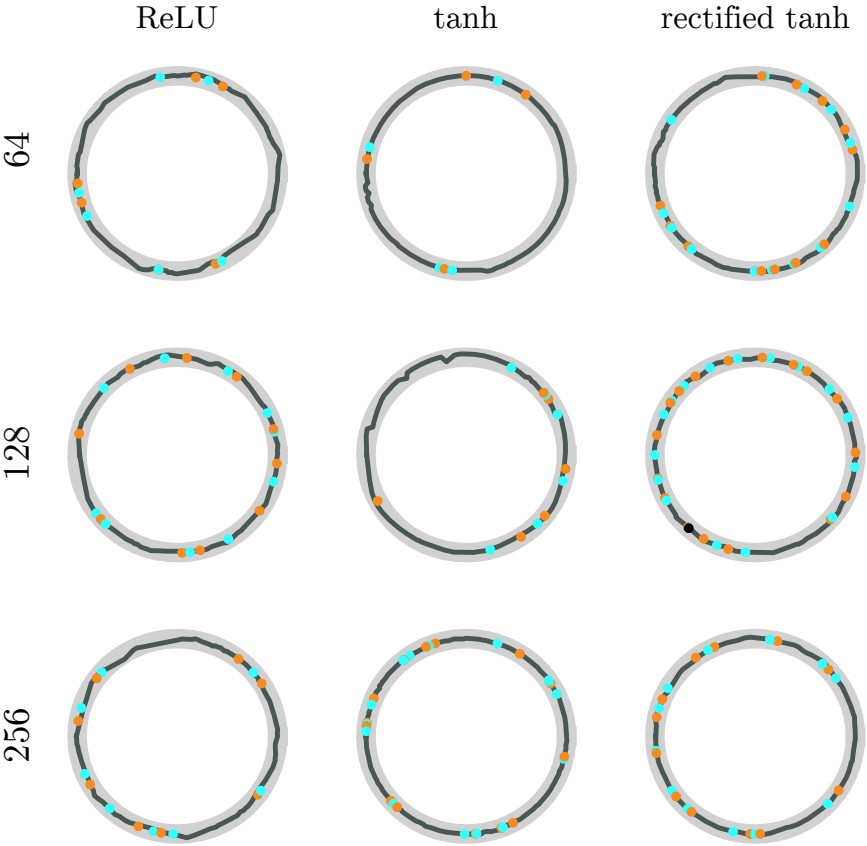

Figure S19: Representative identified invariant manifolds (projected onto the output space, in black) with the fixed points (cyan for stable, orange for saddle and black for unstable). The reference target ring is shown in grey.

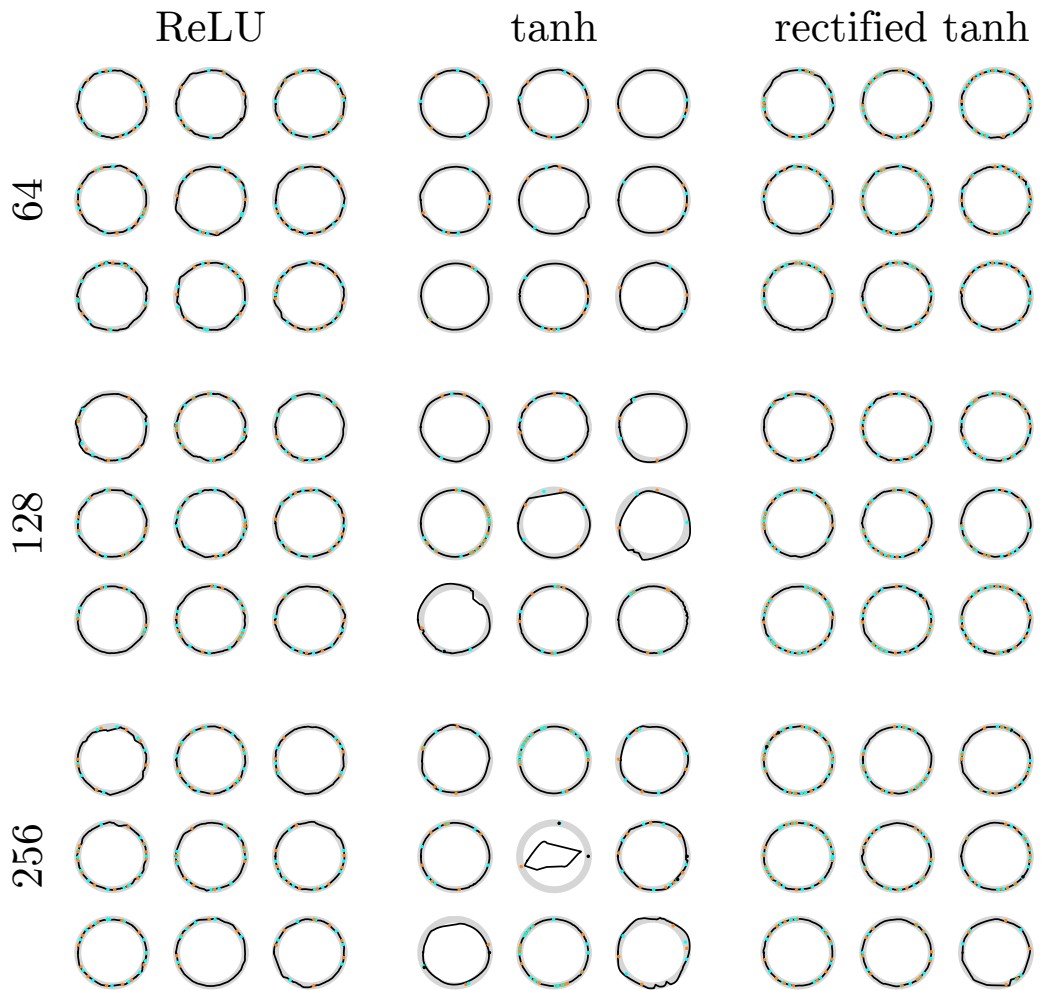

Figure S20: The identified invariant manifolds with the fixed points (cyan for stable, orange for saddle and black for unstable) for all inferred networks (except the ones in Fig S19).

## S7.9 Double angular integration task

The analysis of networks trained on the double angular velocity integration task indicates a torus shaped invariant slow manifold as predicted by our theory (Fig.S21A and B). Furthermore, the angular memory error (measured as the sum of the two separate angular errors) of the trained networks show the same conformity to the theoretical bound as defined by the uniform norm of the vector field on the identified invariant manifold. These results underline the universality of our findings beyond 1D tasks.

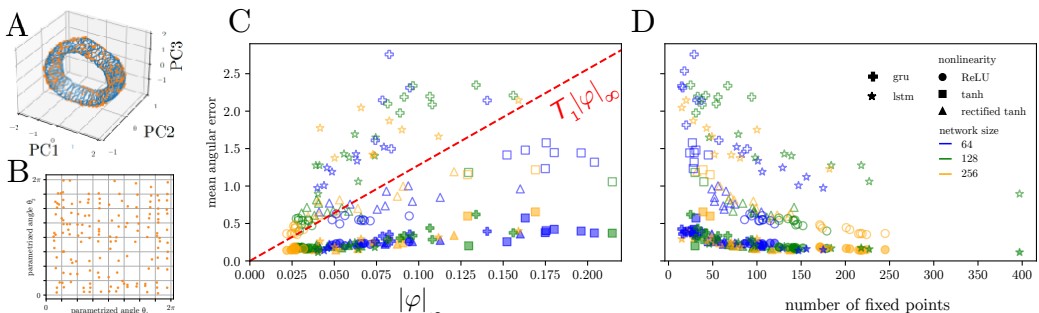

Figure S21: Networks trained on a double angular velocity integration task. **(A)** Initializations (blue) and fixed points (orange) of an example network. **(B)** Fixed points on a 2D parametrization of the torus for the example network. **(C)** The sum of the total mean angular error (sum of the two seperate angular errors over the two rings) is bounded by the uniform norm of the vector field. **(D)** Generalization for longer memory depends on the number of fixed points in the network.

Two dimensional attractors have been proposed to simultaneously represent heading direction and uncertainty of it [82,148].

### S7.9.1 Methods

**Fixed point search** In our study, we implemented a method to analyze the convergence and uniqueness of fixed points within our data. Specifically, we considered a convergence threshold of $10^{-4}$, meaning that the iterative process was halted when the change in the solution between consecutive iterations fell below this value, indicating convergence. Additionally, to assess the uniqueness of the fixed points, we applied a broader threshold of $10^{-2}$, ensuring that any fixed points identified within this margin were considered distinct.

**Double Mean Angular Error** The Mean Angular Error (MAE) was computed as the sum of the individual angular errors observed in the data. This measure provides an aggregate view of the angular errors for the two separate subtasks.

**Uniform Norm of the (Projected) Vector Field** To evaluate the uniform norm of the (projected) vector field, we calculated the sum of the individual uniform norms of the vector field components.

## S7.10    Comparison to other methods

**Fixed point analysis** [87] and [149] are primarily concerned with a pointwise definition of slowness, by comparison normal hyperbolicity requires a uniform separation of timescales over the entire invariant manifold. In [87], it was observed that structural perturbations (random gaussian noise in the parameters with zero mean and standard deviation) still leads to the same approximate plane attractor dynamical structure is still in place, however no explanation is provided for these observations. Our theory can explain why perturbations to the trained RNN. The Persistence Manifold Theorem (Theorem 1) guarantees that for small perturbations (of size $\epsilon$) the persistent invariant manifold will be at the approximate same place (it will be at a distance of order $\mathcal{O}(\epsilon)$).

**Piecewise linear recurrent neural network** [37] identifies asymptotic behaviors in dynamical systems, fixed point dynamics and more general cases cycles and chaos. We look beyond asymptotic behavior and characterize attractive invariant manifolds, thereby also identifying connecting orbits (or heteroclinic orbits) between fixed points. Although we developed new analysis methods for dynamical systems to find slow manifolds in them, we do not propose a new general framework for analysis of all dynamical systems. Finally, [37] provides analysis tools for Piecewise-Linear Dynamical Systems, while our methods are generally applicable to RNNs with any activation function.

