# OpenReview forum: "Back to the Continuous Attractor"
_NeurIPS.cc/2024/Conference — NeurIPS 2024 poster_

### Official Review · Reviewer_TfFn · 2024-06-24

**Soundness:** 3
**Presentation:** 2
**Contribution:** 3
**Rating:** 6
**Confidence:** 3

**Summary:**

The authors study continuous attractor networks, and their famous instability under noise. They show that continuous attractors, despite being unstable, are functionally robust, and analyse some behaviours when noise is introduced.

**Strengths:**

The main thrust of the paper was very interesting and very novel. I am a theoretical neuroscientist, and the studied noise-instability of CANs is a classic problem that we are taught in lectures. This paper appears to resolve this problem, using the very clever proof technique of 'knowing a relatively esoteric piece of maths (to me!) that solves the problem of interest as a corollary'. If I've understood this right, it should be applauded!

The rest of the paper was a smorgasbord of somewhat interesting characterisation of the behaviour of noise perturbed CANs, illustrating some interesting phenomenology.

A lot of the exposition was very clear, especially on the micro-scale: individual ideas were explained well. Figures 1 and 3 were good.

The bound on error in figure 5A was cool (if true).

**Weaknesses:**

I got very confused by what most of the paper was trying to show. The main result, section 3, is, almost draw-droppingly, nice. (Though I'm partly trusting that the authors are applying the results correctly; they seem to be, because the step from the stated theorem to their claims is not large) The rest is, ..., a bit more meh, especially in how it was presented. I would have liked a lot more signposting: why are these sections the right ones to care about? What are you showing, and why?

For example, section 2 shows that noise degrades CANs - a nice literature review - and then shows a categorisation of perturbed fixed point dynamics. I guess the idea was to show that noise-instability was a problem for all CAN types in the literature? If so you could do with signposting that. If not, why did you include all of that? Why was it important that some patterns appear and not others in the perturbed dynamics?

Then there were a lot of quite confusing things, especially in the figures:

Fig 1) A was a great figure, B was never mentioned in the text, despite being very pretty.

Fig 2) Why were there hexagons everywhere? I never could find any reason for there to be hexagonal figures, did you just make the ring a hexagon for fun? If so, tell the reader! Further, in B and C, are the blue lines numerical? How did you choose what to colour blue? Should I be using it as evidence for your big theoretical claim? Or is it just illustrative?

Fig 3) What am I looking at in figure 4A2: it says example trajectory? But there are many trajectories, no? The dots in B and C are presumably fixed points, why then is it called a limit cycle (line 246)? It doesn't look like that? Why is 4D described as around a repulsive ring invariant manifold? Are you describing the ring inside the torus? (rather than the presumably attractive limit cycles that are marked on the figure) What does the colouring of the torus (the greys) denote? I didn't get told what the task was for figure A1 so had to go to the appendix to see that this is apparently the output of the integration? Why are you plotting the output space, and not the recurrent space as in all the other figures?

On that last point, why include details of the (standard) discretisation scheme, MSE loss, and network details, when key steps to understand what I am looking at (e.g. figure A1 = output space) are missing?

Figure 5A) Why did one of the finite time results go above the line? Shouldn't this be an exact theoretical result, yet it appears not to be true?

Seemed obtuse to claim a linear relationship then show a log-linear plot, fig 5C? How should I see this?

Did you define the memory capacity?

Did you need to introduce omega-limit set, especially given the likely audience for this paper?

Finall, some other points:

You should definitely cite and discuss the relationship to this paper: Information content in continuous attractor neural networks is preserved in the presence of moderate disordered background connectivity, Kuhn & Monasson, 2023.

What was section 5.2 trying to show? First it claims that 2.2 presents a theory of approximate solutions in the neighbourhood of continuous attractors (news to me, as far as I could tell, that section showed me that all CAN models were unstable to noise and turn into a variety of different fixed points under noise, that doesn't sound like a theory at all? Section 3 seems to be the theory?) Then you list four conditions on what sounds like exactly the same problem? What is the difference between dynamical systems having robust manifolds, and the working memory task being solved, isn't the whole point of the model that these two are the same? (i.e. you can solve a working memory task with a CAN). Is this supposed to be a concluding section that says when working memory can be solved? Then why have you suddenly defined state and dynamical noise that haven't been used before, I thought we had a perfectly nice definition of perturbations on the network (equation 2)? This section seemed... strange, in my eyes the paper would be improved by removing it

Smaller things:
- line 107 - comment left in doc
- Figure 2 caption line 1, missing 'of' and pluralisation of 'implementation'

So all in all, I think the exposition, despite, as I said, often being very clear when describing a single idea, is, on a macro scale, a mess. I had a very hard time following, and the figures were quite tough-going. I think this paper should probably get in, but I think it should be given a good clean up first, at least for a humble neuro-theorist, rather than a dynamical systems guru, to understand.

**Questions:**

My, many, questions and confusions ended up being in the weaknesses section.

**Limitations:**

The limitations were somewhat discussed.

---

> ### Author Rebuttal · Authors · 2024-08-07
>
> We would like to thank the reviewer for their valuable comments. They have significantly enhanced the quality of our manuscript.
>
> > signposting
>
> We thank the reviewer for and agree with this feedback. We improved the text by:
> - Changing the last paragraph of the introduction, summarizing each section
> - Highlighting the main questions arising from the discussed examples
> - Adding sentences at the start of each (sub)section to point out the main message
>
> > Fig 1) A was a great figure...
>
> We now added the needed reference to Fig.1B.
>
> > Fig 2) Why were there hexagons everywhere?
>
> This is a quality of the ring attractor in [1]. The attractor implemented by 6 neurons is made up of 6 line attractors, fused at their ends. At these fusion points there is a lack of smoothness (similar to the function abs(x)). Therefore, this piecewise straight "ring" attractor, when projected onto two dimensions looks like a hexagon. We have added clarification to this on the main text.
>
> > Further, in B and C...
>
> We find the connecting orbits between the fixed points by identifying the slow part of simulated trajectories. From these slow trajectories, we choose the trajectory that is closest to two fixed points. These trajectories go from a saddle node to a stable fixed point. We do this for every pair of neighbouring fixed points.
>
> > Should I be using it as evidence for your big theoretical claim?
>
> In this section, we aim to motivate the relevance of our theory by demonstrating two key points: 1) that continuous attractors are inherently fragile to noise in the parameters (a well-known fact), and 2) that all bifurcations from and approximations of continuous attractors share a common feature: a slow invariant attractive manifold.
> In the subsequent sections, we provide an explanation of these universal features.
>
> > What am I looking at in figure 4A2...
>
> We appreciate the remarks and have revised the text accordingly.
> 1. You are correct; this figure shows multiple example trajectories. We have corrected this in the text and caption.
> 1. We have corrected the mistake of referring to Fig. 4B and C as a limit cycle.
> 1. We did not include the slow repulsive points that we found as it decreased the interpretability of the figure and this structure is not relevant for how the network solves the task.
> 1. The grey lines on the torus represent simulated trajectories, indicating the network dynamics after a saccade.
> 1. Fig. 4A1 shows the output of an integrated angular velocity, illustrating the task in addition to the solution type in Fig. 4C.
> Fig. 4A1 and 4A2 are different in that they show how the networks behave (in output space).
> The other subfigures in Fig.4 illustrate the stability structures of the networks, i.e., to which part of state space they tend over time in the absence of input.
>
> > On that last point...
>
> We included these details to support reproducibility.
>
> > Figure 5A) Why did one of the finite time results go above the line?
>
> This is indeed an exact theoretical result; however, our numerical methods are not exact. Because we numerically approximate the invariant manifold of each trained RNN, on which we calculate the uniform norm of the vector field, we cannot guarantee the vector field simulated trajectories follow to be exact. Additionally, the network initialization along the invariant manifold is approximate due to our parametrization (using a cubic spline). Nevertheless, it is important to note that this method has only a single violation of our theory among approximately 250 networks tested.
>
> > Seemed obtuse...
>
> We appreciate the reviewer pointing this out. We have corrected the mistake and now observe the log-linear relationship, which aligns with the theoretical expectation. The angular error should asymptotically approach zero as the number of fixed points increases, making the log-linear plot the appropriate representation for this relationship.
>
> > Did you define the memory capacity?
>
> We determine the location of the fixed points through points of reversal of the flow direction (the system evolves along a 1D subspace). We calculate the probability of a point converging to a stable fixed point by assessing the local flow direction (which allows us to characterize the basin of attraction). The memory capacity is the entropy of this probability distribution. The definition is in S.4.3.1., referenced in the main text. We hope these definitions are clearer in the new version of Sec. 5.
>
> > Did you need to introduce omega-limit set...
>
> We believe that this definition is supporting the definition of memory capacity.
> As we explained above, the memory capacity is calculated from the omega-limit set of each of the points on the invariant ring.
> This idea can be more generally applied to systems with other omega-limit sets, like limit cycles or chaotic orbits and therefore included this definition.
>
> > You should definitely cite...
>
> [2] analyzes an Ising network perturbed with a specially structured noise at the thermodynamic limit.
> Although their analysis elegantly shows that the population activity of the perturbed system does not destroy the Fisher information about the input to study instantaneous encoding, they do not consider a scenario where the ring attractor is used as a working memory mechanism. In contrast, our analysis involves understanding how the working memory content degrades over time due to the dynamics. We are not aware of any mean field analysis that covers this aspect.
> We include this work to discuss continuous attractors in mean field approaches.
>
> > What was section 5.2 trying to show?
>
> See the shared rebuttal for our clarification of Sec. 5.2.
>
> ### References
> [1] Noorman, M. et al. (2022). Accurate angular integration with only a handful of neurons. bioRxiv, 2022-05.
>
> [2] Kühn, T., & Monasson, R. (2023). Information content in continuous attractor neural networks is preserved in the presence of moderate disordered background connectivity. Physical Review E, 108(6), 064301.

---

> > ### Comment · Reviewer_TfFn · 2024-08-11
> > **Response**
> >
> > I thank the authors for their detailed and impressive responses.
> >
> > Broadly, my comments have been answered. That said, they were largely about clarity, so it is hard to verify if the paper does indeed make more sense.
> >
> > Certainly including the details of the network is important for reproducibility (sec 4.1) and apologies for making comments that sound like they're suggesting these details should be removed. That was not my intention! Rather, it seemed strange to put such standard details in the main text rather than the appendix, when the space could be used to explain so many other things.
> >
> > I continue to think the paper should get in, and will raise my score to 6 on the assumption that the resulting paper will indeed be much clearer. (I agree with reviewer 8M6u that this paper in its first version has an audience problem)

---

> > > ### Author Response · Authors · 2024-08-12
> > >
> > > We would like to thank the reviewer again.
> > >
> > > For our proposal to increase the readability of the paper and to ameliorate the audience problem, we would like to point the Reviewer to our comment to reviewer 8M6u: https://openreview.net/forum?id=fvG6ZHrH0B&noteId=GM02jaKd5z.

---

### Official Review · Reviewer_8M6u · 2024-07-12

**Soundness:** 4
**Presentation:** 2
**Contribution:** 4
**Rating:** 8
**Confidence:** 4

**Summary:**

The manuscript investigates the stability and robustness of continuous attractors to small deformations in vanilla recurrent neural networks. Continuous attractors were once heavily studied in the context of working memory, but they are inherently fragile and susceptible to even small perturbations and lead to very different topologically distinct attractor landscapes. This study, on the other hand, focuses on finite time behavior around these distinct attractor landscapes. Specifically, the authors theoretically discuss and empirically show that continuous attractors exhibit slow manifold structures under perturbations, which persist despite their theoretical instability.

**Strengths:**

- The study of continuous attractors in the finite time limit, opposite of what is usually studied with attractor landscapes, is a fresh look, novel, original, and interesting.
- The authors provide a superb theoretical motivation, which convinced me of their results correctness.

**Weaknesses:**

- The presentation in section 5 becomes unclear, and perhaps too dense. The authors may want to expand on this section significantly.
- Some more control experiments need to be added for proving the generality (See below).

**Questions:**

I believe the work warrants a borderline accept as is, yet I would feel very supportive of its publication (up to a strong accept) if the authors performed the following changes:

- I believe the slow manifold picture the authors are introducing here is not too different from [1], specifically Fig. 7. Can the authors please clarify the differences in the main text?

- The fast-slow decomposition does not seem to be specific to vanilla RNNs. Can the authors please include experiments with LSTMs and GRUs, which would support their claims on generality.

- Similarly, the experiments on Section 4 are centered around ring-like attractors. Can you please show an example with other structures? For example, you can consider the delayed addition/multiplication tasks discussed in [2]. Relatedly, similar slow dynamics arguments are made in [2], which I believe the authors should list as a close related work and explain the differences in their approach.

- A more detailed discussion of Section 5 is desired. I was able to understand the main takeaways, but could not evaluate the validity of the claims. Perhaps the authors may want to explain in simpler terms the evidence presented in Fig. 5.

- The title is not descriptive of the manuscript's content and feels like as if it belongs to a blog post. Could you please update the title to be representative of the paper's conclusions?

Citations

[1] Opening the Black Box: Low-Dimensional Dynamics in High-Dimensional Recurrent Neural Networks, David Sussillo and Omri Barak, Neural Computation, 2013

[2] Schmidt, D., Koppe, G., Monfared, Z., Beutelspacher, M., & Durstewitz, D. (2019). Identifying nonlinear dynamical systems with multiple time scales and long-range dependencies. arXiv preprint arXiv:1910.03471.


**Edit:** My main remaining concern is the presentation, which I find to be unnecessary complicated. That being said, this work is an important piece of contribution to neuroscience and I support its acceptance.

**Limitations:**

Yes.

---

> ### Author Rebuttal · Authors · 2024-08-07
>
> We would like to thank the reviewer for their valuable comments and suggestions (and appreciate the recent edit). They have significantly enhanced the quality of our manuscript.
>
> ### Weaknesses:
>
> > Some more control experiments need to be added for proving the generality (See below).
>
> We address the specific questions below, but would like to emphasize that the main contribution of this submission is theoretical.
> The numerical experiments are meant to illustrate our results, rather than prove them.
>
> ### Questions:
> > I believe the slow manifold picture the authors are introducing here is not too different from [1], specifically Fig. 7. Can the authors please clarify the differences in the main text?
>
> We appreciate the remark, and will revise the text to include the following:
> 1. The two approaches are indeed similar, however there are subtle technical differences.
> 1. In [1], Sussillo and Barak are primarily concerned with a pointwise definition of slowness, by comparison normal hyperbolicity requires a uniform separation of timescales over the entire invariant manifold.
> 1. Our theory can explain why perturbations to the trained RNN (with random gaussian noise with zero mean and standard deviation) still lead to the same approximate plane attractor dynamical structure is still in place. The Persistence Theorem guarantees that for small perturbations (of size $\epsilon$) the persistent invariant manifold will be at the approximate same place (it will be at a distance of order $\mathcal{O}(\epsilon))$. See Figure 9 for the experiments of the structural perturbations in [1]. They do not provide an explanation for their observations.
>
>
> > The fast-slow decomposition does not seem to be specific to vanilla RNNs. Can the authors please include experiments with LSTMs and GRUs, which would support their claims on generality.
>
> We have now trained and analyzed both LSTMs and GRUs on the angular integration task. We include our preliminary results in a separate document uploaded to OpenReview.
>
>
> > Similarly, the experiments on Section 4 are centered around ring-like attractors...
>
> We appreciate the comment, and have included an additional task where the approximate continuous attractor is of higher dimension, namely a double angular velocity integration task. Please see the shared reply to all reviewers on our new findings.
> Specifically regarding addition or multiplication tasks, an idealized solution to either would require that the RNN represent $G = (\mathbb{R},+)$ or $G = (\mathbb{R}_{+},\times)$, which are not compact.
> Because this contradicts our technical assumptions, we opt to focus on tasks where the invariant manifolds are naturally compact.
>
>
> > similar slow dynamics arguments are made in [2]...
>
> We thank the reviewer for pointing out this work, we will reference it accordingly.
> This work identifies asymptotic behaviors in dynamical systems, fixed point dynamics and more general cases cycles and chaos.
> We look beyond asymptotic behavior and characterize attractive invariant manifolds, thereby also identifying connecting orbits (or heteroclinic orbits) between fixed points.
> We would like to reiterate that we believe that the main contribution of the paper is a new theory of approximations of continuous attractors.
> Although we developed new analysis methods for dynamical systems to find slow manifolds in them, we do not propose a new general framework for analysis of all dynamical systems.
> Finally, [2] provides analysis tools for Piecewise-Linear Dynamical Systems, while our methods are generally applicable to RNNs with any activation function.
>
> > The presentation in section 5 becomes unclear, and perhaps too dense...
>
> We agree with the reviewer, the updated manuscript will include a substantial revision of section 5. In it, we will focus on clarity.
> We have split the section into generalization properties of trained RNNs (all relating to Fig.5 and in relation to our theoretical prediction on the error bound) and the four conditions that guarantee that a system that approximates an analog memory system will be near a continuous attractor (which was section 5.2).
> Please, see the shared rebuttal for our clarification of Section 5.2.
>
>
> > the evidence presented in Fig. 5.
>
> The main message of Fig.5 is to show the validity of our theoretical predictions about the bound to the memory based on the uniform norm of the flow.
> Besides, we can demonstrate that even though all networks learned a continuous attractor approximation, they are distinguished from one another by their fixed point topology, which determines their asymptotic behavior and hence generalization properties (Fig.5D and E).
> These results indicate the distance to a continuous attractor as was discussed in Sec.3.2 as measured by the uniform norm of the invariant slow manifold.
>
>
> > The title is not descriptive of the manuscript's content and feels like as if it belongs to a blog post. Could you please update the title to be representative of the paper's conclusions?
>
> The title references the reversal of the Persistence Manifold theorem, i.e., how to get back to a continuous attractor.
> Furthermore, it references that we can return to continuous attractors as a useful concept to describe neural computation because of the deep connection of all continuous attractor approximations.
> We can however propose to revise the title to: "A theory of analog memory approximation: Back to the continuous attractor."
>
>
> ### References
> [1] Opening the Black Box: Low-Dimensional Dynamics in High-Dimensional Recurrent Neural Networks, David Sussillo and Omri Barak, Neural Computation, 2013
>
> [2] Schmidt, D., Koppe, G., Monfared, Z., Beutelspacher, M., & Durstewitz, D. (2019). Identifying nonlinear dynamical systems with multiple time scales and long-range dependencies. arXiv preprint arXiv:1910.03471.

---

> > ### Comment · Reviewer_8M6u · 2024-08-07
> >
> > With the proposed changes, I believe this work is an important addition to the theoretical neuroscience literature. I will increase my score to match this belief. I do believe there is a way for authors to increase my score even further if the authors commit to addressing the following concern:
> >
> > I believe one of the main readers of this work, though the authors probably do not intend it that way, will be experimental neuroscientists. I believe the current writing is too heavy for them to be able to understand this work and start testing some of these ideas with experiments. Could you comment on how you would go about making changes to the writing to be more accessible to this audience? In my mind, some of the theoretical contributions can be toned down in favor of accessibility to a more general audience, which in turn would increase your impact.
> >
> > I believe your responses to all reviewers are satisfactory. I appreciated the new experiments in short time and I will be championing for the acceptance of this work. I do believe if you can provide a satisfactory answer to my concern above, I feel comfortable recommending this work for a spotlight.

---

> > > ### Comment · Reviewer_8M6u · 2024-08-11
> > >
> > > The silence from the authors is slightly concerning. I will be waiting for another 24 hours before finalizing my score, though I must admit that it is hard to not take this as an indication of low confidence by authors regarding the applicability of their work to experimental neuroscience.

---

> > > > ### Author Response · Authors · 2024-08-12
> > > >
> > > > Thank you for your patience. We just uploaded our proposed additions and changes to make the text more accessible.

---

> > > > > ### Comment · Reviewer_8M6u · 2024-08-12
> > > > >
> > > > > I believe this new addition will be an important step towards increasing the work's impact. I support the acceptance, though I will keep my score as is. I feel slightly concerned about author's commitment to making more changes, as it took a long while to answer a simple question. So, I make this judgement based on the submitted version and clarifications during the rebuttal.

---

> > > ### Author Response · Authors · 2024-08-12
> > >
> > > Thank you for your positive feedback. We greatly appreciate your thoughtful consideration of the potential impact on experimental neuroscientists, a crucial audience we indeed hope to reach!
> > >
> > > To address your concern, we propose to add a new section to the main text, simplify the language of our presentation, and add additional material in the appendix.
> > >
> > > ## We will include a new section titled: "Implications on experimental neuroscience"
> > >
> > > Following is a draft:
> > >
> > > Animal behavior exhibits strong resilience to changes in their neural dynamics, such as the continuous fluctuations in the synapses or slight variations in neuromodulator levels or temperature. Hence, any theoretical model of neural or cognitive function that requires fine-tuning, such as the continuous attractor model for analog working memory, raises concerns, as they are seemingly biologically irrelevant. Moreover, unbiased data-driven models of time series data and task-trained recurrent network models cannot recover such continuous attractor theories precisely. Our theory shows that this apparent fragility is not as devastating as previously thought: despite the "qualitative differences" in the phase portrait, the "effective behavior" of the system can be arbitrarily close, especially in the behaviorally relevant time scales. We show that as long as the attractive flow to the memory representation manifold is fast and the flow on the manifold is sufficiently slow, it represents an approximate continuous attractor. Our theory bounds the error in working memory incurred over time for such approximate continuous attractors. **Therefore, the concept of continuous attractors remains a crucial framework for understanding the neural computation underlying analog memory, even if the ideal continuous attractor is never observed in practice.** Experimental observations that indicate the slowly changing population representations during the "delay periods" where working memory is presumably required, do not necessarily contradict the continuous attractor hypothesis. Perturbative experiments can further measure the attractive nature of the manifold and their causal role through manipulating the memory content.
> > >
> > > ## Simplifying Language
> > > We will revise the manuscript to reduce the use of technical jargon and complex theoretical language where possible. This will involve clearly defining key terms and concepts upfront and using more intuitive explanations throughout the text.
> > > We will simplify Theorem 1 to only talk about invariant manifolds (locally invariant manifolds indeed unnecessarily complicate the statement).
> > >
> > > We will furthermore leave out concepts such as omega-limit sets and only include it in the supplementary material for the curious theoretician to show how to generalize certain concepts to other stability structures (in this case going from fixed points to considering limit cycles as well). Similarly, we will also relegate discussions of the Hausdorff distance to a less prominent section in the supplementary materials (as this part of the theory is still important to guarantee that the persistence invariant manifold is close to the original continuous attractor).
> > >
> > > Additionally, we include an additional supplementary section to explain through less technical and in more intuitive terms the dynamical systems terms that are essential. This includes compact, normally hyperbolic, invariant, diffeomorphism and heteroclinic/connecting orbits. We will further explain what the neighborhood of a vector field entails (see also our proposed intuitive definitions list below).
> > >
> > > Finally, in this new supplementary section, we will include a subsection with a visual illustration of our claims, aiming to tap into human geometric intuition. In this visualization, we show how models that behave almost like a perfect analog memory system correspond to a volume of models in the space of dynamical systems.
> > >
> > > We will furthermore provide intuitive definitions of several key concepts used in our paper:
> > >
> > >
> > > - Manifold: A part of the state-space that locally resembles a flat, ordinary space (such as a plane or a three-dimensional space, but more generally $n$-dimensional Euclidean space) but can have a more complicated global shape (such as a donut).
> > > - Invariant set: A property of a set of points in the state space where, if you start within the set, all future states remain within the set and all past states belong to the set as well.
> > > - Normally Hyperbolic Invariant Manifold: A behavior of a dynamical system where flow in the direction orthogonal to the manifold converges (or diverges) to the manifold significantly faster than the direction that remains on the manifold.
> > > - Diffeomorphism: A diffeomorphism is a stretchable map that can be used to transform one shape into another without tearing or gluing.
> > > $C^1$ neighborhood of a $C^1$ function: A set of functions that are close to the function in terms of both their values and their first derivatives.

---

### Official Review · Reviewer_CAVv · 2024-07-16

**Soundness:** 3
**Presentation:** 2
**Contribution:** 3
**Rating:** 7
**Confidence:** 3

**Summary:**

This paper studies the fragility of continuous attractors, which have been used to explain various computations or functions in the brain related to memory and navigation, to perturbations. The authors mainly focus their analyses on ring attractors, which have been used to model continuous-valued memory. Under perturbation, the authors find that the bifurcations of continuous attractors exhibit structurally stable forms with different asymptotic behaviors but similar finite-time behaviors as the original attractor. For example, a stable limit cycle arising from a bifurcation of a ring attractor would be functionally similar to the ring attractor in finite time. Thus, the authors posit that signatures of the continuous attractor persist even under perturbation in the form of a persistent, attractive manifold which serves as an approximate continuous attractor. Experiments and analyses on recurrent neural networks performing analog memory tasks show that the networks learn approximate continuous attractors. Thus, the authors conclude from their theory and numerical experiments that approximate continuous attractors do remain a good model of analog memory.

**Strengths:**

1. The authors' theoretically demonstrate the existence of a persistent attractive manifold in various bifurcations of a continuous attractor, and study the systems' finite time behaviors to show that they are functionally equivalent. To my knowledge, this is a novel contribution and an important result to bolster the continuous attractor hypothesis.
2. Apart from the theory, the authors also carry out numerical experiments on a working memory task, characterized by a 1-D ring attractor. Through their analyses they validate their theory.
3. The authors also study the generalization properties of the approximate attractors.

**Weaknesses:**

1. The experiments and associated analyses focus solely on networks that approximate 1D ring attractors. This is quite simplistic, and at least for the numerical expreiments, the authors could consider tasks like navigation where a planar 2D attractor is approximated by the networks.
2. The authors have only qualitatively characterized the variations in the topologies of the networks. It is perhaps possible to quantitatively characterize this by using Dynamical Similarity Analysis [1] on various trained networks.
3. For the generalization analysis, the authors could evaluate generalization performance by the nature/type of the approximate attractor as well. Furthermore, although I may have missed this, could the authors comment on what networks hyperparameters lead to which approximations?
4. The figures and presentation could be improved:
    1. On line 107 there is a comment that should be removed ("add link to details").
    2. Fig. 4C, caption should indicate the nature of the solution found.
    3. Fig. 5B, y-axis label is missing.
    4. Fig. 5D, could also show the mean $\pm$ std for classes of networks.
    5. Fig. 5E, y-axis label is missing. Also, the authors could just use the normalized MSE on the axis could just follow the convention used in Fig. 5A instead of using dB.
    6. Overall, the writing could be improved in several places to improve clarity. For example, the conclusions of the generalization analysis and their implications are not very clear, and how this connects to the various types of approximate attractors is not clear (related to W3).

**References:**
1. Ostrow et al. "Beyond geometry: Comparing the temporal structure of computation in neural circuits with dynamical similarity analysis." Advances in Neural Information Processing Systems 36 (2024).

**Questions:**

See the Weaknesses section. I have some additional questions:
1. How do the authors identify the various kinds of approximations of the attractors? Can this be automated, perhaps by using to DSA to cluster the various types?
2. At what level of performance are all trained networks compared? Are they all trained until the same loss value and how close is this MSE to 0?

**Rebuttal update:** Score increased from 5 to 7.

**Limitations:**

There is no limitations section. Tthe authors have mentioned in the discussion that they do not explicitly characterize all possible invariant manifolds identified. However, there are other limitations as well such as the lack of diversity in the tasks/attractors explored. I would also encourage the others to unpack the limitations related to the numerical fast-slow analysis, such as how sensitive the results are to the threshold hyperparameter, cases or specific attractors where it does not work as well, etc.

---

> ### Author Rebuttal · Authors · 2024-08-07
>
> We would like to thank the reviewer for their valuable comments and suggestions.
>
> ### Weaknesses
> > The experiments and associated analyses focus solely on networks that approximate 1D ring attractors.
> > This is quite simplistic, and at least for the numerical experiments, the authors could consider tasks like navigation where a planar 2D attractor is approximated by the networks.
>
> We appreciate the remark, and have included an additional task where the approximate continuous attractor is of higher dimension, namely a double angular velocity integration task. Please see the shared reply to all reviewers on our new findings.
> The networks develop corresponding approximate continuous attractors that have the same structure as the task requires (in this case a torus).
>
> That being said, we would like to reiterate that the primary contribution is theoretical and the numerical experiments are meant to illustrate the theory.
> Regarding navigation tasks, two points bear mentioning.
> 1. For planar attractors are diffeomorphic to $R^2$, note that they do not conform to the assumptions on normally hyperbolic invariant manifolds, since $R^2$ isn't compact.
> There are suitable generalizations of this theory to noncompact manifolds [1], but we do not pursue them since they require more refined tools, which would only obscure the point that we are trying to make.
> 2. Tangentially, we would also like to point out that we assume that neural dynamics are naturally bounded (e.g. by energy constraints) and hence sufficiently well described by compact invariant manifolds.
>
> In the revised version of the manuscript, we will include the above limitations and provide reference to [2].
>
> > The authors have only qualitatively characterized ...
>
> We thank the reviewer for pointing out the reference; we applied DSA to our numerical results.
> Our preliminary observations are that DSA reflects the fact that the geometry of the invariant manifold is preserved, but it cannot detect the emergence of fixed-points and saddles on the perturbed manifold.
> The DSA values clustered around two points regardless of the number of fixed points.
> This appears to be consistent with the results reported in the referenced paper, c.f. Figure 4 shows a gradual increase in DSA as $\alpha \to 1$ despite having a bifurcation at $\alpha = 1$.
>
> Lastly, we would like to note that the analysis using DSA cannot be trivially automated. As pointed out by the authors of DSA:
> 1. The DSA 'score' is relative; one needs to compare different dynamics.
> 1. DSA essentially requires 'learning' or fitting a separate model, which implicitly requires performing model selection with respect to the delay embedding, rank of the linear operator.
>
> For these reasons, we would like to adhere to our initial analyses.
>
>
> > For the generalization analysis, the authors could evaluate generalization performance by the nature/type of the approximate attractor as well.
>
> We looked at the generalization performance by the nature/type of the approximate attractor (Fig.5D MSE vs number of fixed points).
>
> >Furthermore, although I may have missed this, could the authors comment on what networks hyperparameters lead to which approximations?
>
> The only networks hyperparameters that we varied were the nonlinearity and the size.
> In all our figures we show which nonlinearity and size corresponds to which fixed point topology (which we characterize through the number of fixed points on the invariant ring).
>
>
> > The figures and presentation could be improved [...]
>
> We appreciate the comments, and changed the manuscript accordingly.
>
> > Overall, the writing could be improved in several places to improve clarity.
>
> We improved the writing, focusing on overall clarity. See the shared comments.
>
>
> >the conclusions of the generalization analysis and their implications are not very clear...
>
> In the revised version, we will make a stronger point that connects the inherent slowness of the invariant manifold to the generalizability of the approximate solutions.
> We also added a longer description of the implications of our numerical experiments to the main text.
>
> ### Questions:
>
> > How do the authors identify the various kinds of approximations of the attractors? Can this be automated, perhaps by using to DSA to cluster the various types?
>
> We identify approximations by their (1) attractive invariant manifold (as motivated by the theory) and (2) asymptotic behavior (as motivated by our analysis of perturbations and approximations of ring attractors).
> The invariant manifold in our examples typically take the structure of a ring with fixed points and transient trajectories on it.
> In the supplementary document (FigR.1) we show that the identified invariant manifold indeed reflects the fast-slow separation expected for a normally hyperbolic system.
> We find the fixed points and their stabilities by identifying where the flow reverses by sampling the direction of the local flow for 1024 sample points along the found invariant manifold.
> The only example we found that is of another type is the attractive torus (Fig.4D).
> For this network, instead of finding the fixed points, we identifies stable limit cycles where there was a recurrence of the simulated trajectories, i.e., where the flow returned back to an initial chosen number of time steps (up to a distance of $10^{-4}$).
>
> For the difficulties of using DSA, see above.
>
>
> > At what level of performance are all trained networks compared? Are they all trained until the same loss value and how close is this MSE to 0?
>
> All networks are trained for 5000 gradient steps.
> We exclude those networks from the analysis that are performing less than -20dB in terms of normalized mean squared error tested on a  version of the task that is 16 times as long as the task on which the networks were trained.
>
> [2] Eldering, J. (2013). Normally hyperbolic invariant manifolds: The noncompact case (Vol. 2). Atlantis Press.

---

> > ### Comment · Reviewer_CAVv · 2024-08-11
> >
> > I appreciate the authors' rebuttal, the responses mostly address my concerns. I particularly appreciate the authors' additional experimental results with the 2D toroidal attractor. I understand that the DSA score is relative – I was hoping it could quantitatively characterize the similarity between various approximate attractors or help in clustering approximations with different numbers of fixed points, but the authors' response is illuminating. Also, related to another reviewer's concern, it might help to explicitly show the steps in the Euler-Maruyama integration to show the relationship between (6) and (7).
> >
> > With the proposed changes and writing updates, I'm happy to increase my score to 7. I think this is a valuable contribution to theoretical neuroscience and dynamical systems theory, and hope to see it accepted to NeurIPS.

---

> > > ### Author Response · Authors · 2024-08-12
> > >
> > > We thank the reviewer for their additional comments.
> > > We would like to point the Reviewer to our response to reviewer xmvZ for our notes on how we will explicitly describe the steps in the Euler-Maruyama integration to demonstrate the relationship between (6) and (7): [see](https://openreview.net/forum?id=fvG6ZHrH0B&noteId=oMNXw7U8k7).

---

> > > > ### Comment · Reviewer_CAVv · 2024-08-12
> > > >
> > > > Thanks again for the updates – looks good to me!

---

### Official Review · Reviewer_xmvZ · 2024-07-18

**Soundness:** 3
**Presentation:** 2
**Contribution:** 3
**Rating:** 6
**Confidence:** 4

**Summary:**

The study explores some bifurcations from continuous attractors in neuroscience models, revealing various structurally stable forms. Through fast-slow decomposition analysis, they uncover the persistent manifold surviving destructive bifurcations. Additionally, they analyze RNNs trained on analog memory tasks, showing approximate continuous attractors with predicted slow manifold structures. An important takeaway of their work is that continuous attractors demonstrate functional robustness and serve as a valuable universal analogy for understanding analog memory.

**Strengths:**

- The main idea of the paper is interesting, and this work can have an important takeaway, as mentioned in "Summary".

- The connection to Fenichel’s theorem is very nice.

- The visualizations in Fig. 1 are very helpful to understand the overall message

**Weaknesses:**

1. They discussed some interesting theoretical techniques (e.g., Theorem 1, Proposition 1) in their study. However, their theoretical investigation and results are limited to a few very simple systems, low-dimensional systems either in Section S4 or low-dimensional RNNs with specific activation functions and restrictive settings, i.e., specific parameter values (e.g., equations (1) and (10)). The bifurcation analysis of the line attractors and fast-slow decomposition in Section S2 are also studied for very simple systems. Therefore, it is difficult to determine how general their theoretical discussion is and whether it can be applied to investigate and obtain results for more general and high-dimensional cases.

2. In Sect. 3.1, the perturbation p(x) is not clear enough. Specifically, it is unclear:1) Under what conditions the perturbation function p(x) induces a bifurcation?  2) What types of (generic) bifurcations can arise from the perturbation p(x)? Likewise, the functions h and g are also not clear enough. It is unclear how one can obtain/choose the functions h and g such that the two systems defined by Eq. (2) and Eqs. (3) & (4) are equivalent.

3. What does **sufficiently smooth** mean in Theorem 1? As mentioned by the authors after this theorem, it applies to continuous piecewise linear systems. However, it cannot be applied to all piecewise smooth (PWS) systems , such as Filippov systems. In particular, for these systems, bifurcations involving non-hyperbolic fixed points can be analyzed using similar slow (center) manifold approaches, but only for part of the phase space. However, discontinuity-induced bifurcations cannot be examined in the same way, as there is no slow manifold in these cases.

4. It is unclear under what conditions RNN dynamics can be decomposed into slow-fast form to which we can apply Theorem 1.

5. In Sect. 4.1, line 213,  it is vague how assuming an Euler integration with unit time step, the discrete-time RNN of (6) transforms to eq. (7). Is this transformation independent of the function f and matrix W in eq. (6)?

6. In S4, the sentence "All such perturbations leave at least a part of the continuous attractor intact and preserve the invariant manifold, i.e. the parts where the fixed points disappear a slow flow appears." needs more clarification. Could you explain the mathematical reasoning behind this assertion?

**Questions:**

How does the selection of a specific threshold value influence the identification and characterization of slow manifolds in neural networks with continuous attractors as discussed in the first lines of section 4.2? Could you elaborate on how different threshold settings impact the dynamics of network states and the emergence of persistent manifolds?

**Limitations:**

- The authors have discussed most limitations of the analysis in the discussion section, but I suggest making them more explicit. This could be done by either incorporating a dedicated (sub)section on limitations or adding a bold title "**Limitations**" at the beginning of the relevant paragraph within the discussion section.

- As mentioned above, another important limitation is that it is difficult to determine how general their theoretical discussion is and whether it can be applied to investigate and obtain results for more general and high-dimensional cases.

---

> ### Author Rebuttal · Authors · 2024-08-07
>
> We would like to thank the reviewer for their valuable comments and suggestions.
>
> ### Weaknesses:
> >their theoretical investigation and results are limited to a few very simple systems, low-dimensional systems [...]
>
> We respectfully disagree with the stated limitation-- the role of the analysis, and numerical experiments is not to prove the generality of the theory but to illustrate it. Hence, we focus on visualizable low-dimensional systems and are more helpful in developing intuition.
> Nevertheless, we include additional results with RNNs trained on a 2D task.
> In the updated manuscript, we emphasize that the theory holds under broad, practically relevant conditions.
> 1. We added statements that assure that our theory is applicable regardless of the dimensionality or the invariant manifold (see shared rebuttal).
> 1. Furthermore, we will revise Theorem 1, to show that normal hyperbolicity is both **sufficient and necessary** for invariant manifolds to persist, see [1].
>
> > RNNs with specific activation functions and restrictive settings
>
> We appreciate the reviewer's insightful comment. In response, we have conducted additional experiments with LSTMs and GRUs, for which the results are included in the supplementary document and discussed in the shared rebuttal.
>
> > Under what conditions the perturbation function p(x) induces a bifurcation?
>
> Continuous attractors satisfy the first-order conditions for a local bifurcation; that is, they are equilibria, and their Jacobian linearization possesses a non-trivial subspace with eigenvalues having zero real parts. Consequently, any generic perturbation $p(x)$ will induce a bifurcation of the system.
> For a more comprehensive discussion on this topic, see [2].
>
> > What types of (generic) bifurcations can arise from the perturbation p(x)?
>
> We are working to characterize codimension-1 bifurcations for a ring attractor and believe a simple polynomial normal form can be derived.
> Characterizing bifurcations with codimension $n > 2$ is an open problem.
> Perturbations in the neighbourhood of a ring attractor (in $C^1$ topology) will result in no bifurcation, a limit cycle, or a ring slow manifold with fixed points.
>
> >the functions h and g are also not clear
>
> The essence of Theorem 1 can be restated as follows: **if** the function $f$ has a normally hyperbolic invariant manifold (NHIM), **then** there exist vector fields $h$ and $g$ that satisfy the conditions for equivalence between the systems defined by Eq.2 and Eqs.3&4.
> This means that the existence of these functions is guaranteed under the condition of having a NHIM, but their explicit forms is case specific.
>
> > What does sufficiently smooth mean in Theorem 1?
>
> We mean that a system needs to be at least continuous, but some extra conditions apply if a system is not differentiable (discontinuous systems are not considered in our theory). Since all theoretical models and activation functions for RNNs are at least continuous and piecewise smooth, our theory is broadly applicable.
> Center manifold are unique, and generally are local in **both** space and time and therefore invariance under the flow generally cannot be analyzed using them.
>
> > It is unclear under what conditions RNN dynamics can be decomposed into slow-fast form to which we can apply Theorem 1.
>
> Theorem 1 holds for all dynamical systems that have a normally hyperbolic continuous attractor. For example, RNNs with a ReLU activation functions can only have such continuous attractors. The continuous attractors and their approximations that we discuss are all normally hyperbolic. In fact, there is a huge benefit from having normal hyperbolicity as it can counteract state noise.
>
> > line 213 [...]
>
> The transformation from the continuous-time to the discrete-time is independent on the function $f$ anf the matrix $W$.
> However, it is important to note that the discretization process can result in significantly different system behavior depending on the activation functions used. For instance, discretization can introduce dynamics that are not present in the continuous-time system.
>
> > In S4, the sentence "All such [...]
>
> We will reformulate it as "all such perturbations leave the geometry of the continuous attractor intact as an attractive invariant slow manifold, i.e. the parts where the fixed points disappear a slow flow appears."
> The persistence of the invariant manifold under perturbations is a direct consequence of the normal hyperbolicity condition in Theorem 1.
> Therefore, for a normally hyperbolic continuous attractor there will remain an attractive slow invariant manifold.
>
> **Questions:**
> >How does the selection of a specific threshold value...
>
> We believe that the threshold value used to identify slow manifolds is robust- we used the same value for the newly added RNN models, without modification.
>
> >Could you elaborate [...] emergence of persistent manifolds?
>
> It is unclear for us to which threshold the reviewer is referring. The emergence of persistent manifolds happens under the 4 conditions we discuss.
> We demonstrate that all systems with a sufficiently good generalization property (in our case, defined as networks with NMSE lower than -20 dB) must have a NHIM. The persistence of these manifolds is a direct consequence of their normal hyperbolicity.
>
> **Limitations:**
> >The authors have discussed most limitations [...]
>
> We appreciate the reviewer's suggestion to make the limitations of our analysis more explicit. In response, we have included a dedicated Limitations subsection in the discussion section of the manuscript. Please refer to the shared rebuttal for further details.
>
> [1] Mané, R. (1978). Persistent manifolds are normally hyperbolic. Transactions of the American Mathematical Society, 246, 261-283.
>
> [2] Kuznetsov, Y. A., Kuznetsov, I. A., & Kuznetsov, Y. (1998). Elements of applied bifurcation theory (Vol. 112, pp. xx+-591). New York: Springer.

---

> ### Comment · Reviewer_xmvZ · 2024-08-11
>
> I really appreciate the authors' responses and new experiments, particularly the additional experiments with LSTMs, GRUs, and RNNs trained on a 2D task.
>
> - Regarding the first weakness I mentioned, my main concern was not about the role of numerical experiments in proving the generality of the theory. Rather, it was mostly about:
>
> 1) Whether and how the theory itself is applicable to higher-dimensional systems in the sense that the conditions of the theorems can be met for high-dimensional systems (especially as many systems in neuroscience are high-dimensional). Nevertheless, I appreciate the revision of Theorem 1 to show that normal hyperbolicity is both sufficient and necessary for invariant manifolds to persist.
>
> 2) Bifurcation and stability analysis in Appendix (S2) are not only restricted to a low-dimensional system but also to a system with specific parameters (eq. (9)). One can at least consider such analysis for a more general case, e.g., W = [w11 w12; w21 w22]. Then, one can mention that in some cases, even when having a theory for low-dimensional systems, it can be extended in a similar way to higher-dimensional systems, for instance, through center manifold theory, where the higher-dimensional system has the same low-dimensional center manifold.
>
>
> - Regarding the perturbation p(x) (the second mentioned weakness), thank you for the clarification. Certainly, dynamical systems with nonhyperbolic equilibria and/or nonhyperbolic periodic orbits are not structurally stable, meaning that any ϵ-perturbation  (in the sense of Definition 1.7.1 on p. 38 of [1]) will induce a bifurcation of the system. But, please note that in Sect. 3.1, based on your response, "$l$" must be nonzero. Otherwise, one can consider $l=0$, which implies there is no continuous attractor, which might lead to confusion, as it did for me. So, I suggest changing "$l$" to  "$l \neq 0$" in Sect. 3.1.
>
> - Finally, I agree with Reviewer CAVv that it might help to further clarify Sect. 4.1 to demonstrate the relationship between equations (6) and (7).
>
> ------------------------------------
>
> [1]  J. Guckenheimer and P. Holmes, Nonlinear Oscillations, Dynamical Systems, and Bifurcations of Vector Fields, Springer-Verlag, New York, 1983.

---

> > ### Author Response · Authors · 2024-08-12
> >
> > We thank the reviewer for the additional comments.
> >
> >
> > ## Bifurcation and stability analysis
> > > Bifurcation and stability analysis in Appendix (S2) are not only restricted to a low-dimensional system but also to a system with specific parameters
> >
> > ### Analytically tractable examples
> > We would first like to clarify that in Supplementary Sections 1 and 2 the provided examples are illustrative rather than exhaustive.
> > Recognizing that the supplementary material previously lacked clear structure and signposting, we will revise the text to better convey the motivation behind the examples and to provide a clearer explanation of the analysis.
> > The analysis can furthermore be easily extended to a more general form $W = [w\_{11} w\_{12}; w\_{21} w\_{22}]$ that has a bounded line attractor, through a coordinate transformation.
> >
> >
> > ### Extension of bounded line attractor analysis to higher dimensional systems
> > We agree that this analysis in S2 can be extended to higher dimensions, and we will incorporate this remark into the supplementary text.
> > An extension of these results of a low-dimensional system can be easily achieved by the 'addition' of dimensions that have an attractive flow normal to the low-dimensional continuous attractor or invariant manifold.
> >
> >
> > More generally, the results from a low-dimensional system can indeed be extended to higher-dimensional systems through reduction methods from center manifold theory.
> > On the center manifold the singular perturbation problem (as is the case for continuous attractors) restricts to a regular perturbation problem [1a].
> > Furthermore, relying on the Reduction Principle [2a], one can always reduce all systems (independent of dimension) to the same canonical form, given that they have the same continuous attractor. We thank the reviewer for pointing this out and will add a remark on this possibility to extend results.
> >
> >
> >
> >
> >
> >
> > ## Perturbation and dimensionality
> > > Regarding the perturbation ...
> > We understand how that may have lead to confusion. We changed the statement in Sec.3.1 to be $l\\neq 0$ for clarity.
> >
> >
> > ## Discretization of SDE
> > We appreciate the suggestion to better explain the discretization procedure, i.e., the steps for going from Eq.(7) to Eq.(6).
> >
> >
> > ### Steps
> > In the supplementary material, we will include a note on the discretization, which goes as follows.
> >
> > **Discretize the time variable:** Let $t\_n = n \\Delta t$.
> >
> > The Euler-Maruyama method for a stochastic differential equation
> >
> > $$\\mathrm{d}{\\mathbf{x}} = (-\\mathbf{x} + f( \\mathbf{W}\_{\\text{in}} \\mathbf{I}(t) + \\mathbf{W} \\mathbf{x} + \\mathbf{b} )) \\mathrm{d}{t} + \\sigma\\mathrm{d}{W}\_{t}$$
> >
> > is given by :
> >
> > $$\\mathbf{x}\_{n+1} = \\mathbf{x}\_n + ( - \\mathbf{x}\_n + f ( \\mathbf{W}\_{\\text{in}} \\mathbf{I}\_n + \\mathbf{W} \\mathbf{x}\_{n} + \\mathbf{b} ) ) \\Delta t + \\sigma \\Delta W\_{n}$$
> >
> > with $\\Delta W\_{n}=W\_{(n+1)\\Delta t}-W\_{n\\Delta t}\\sim \\mathcal{N}(0,\\Delta t).$
> >
> > **Subsitute $\\Delta t = 1$:**
> >
> > $$\\begin{aligned}
> >  \\mathbf{x}\_{t+1} &= \\mathbf{x}\_t + ( -\\mathbf{x}\_t + f(\\mathbf{W}\_{\\text{in}} \\mathbf{I}\_t + \\mathbf{W} \\mathbf{x}\_t + \\mathbf{b}) ) + \\sigma \\Delta W\_t, \\\\
> >  &= f(\\mathbf{W}\_{\\text{in}} \\mathbf{I}\_t + \\mathbf{W} \\mathbf{x}\_t + \\mathbf{b}) + \\sigma \\Delta W\_t.
> >  \\end{aligned}
> > $$
> >
> > **Introduce the noise term** $\\zeta\_t = \\sigma \\Delta W\_t$, which represents the discrete-time noise term.
> > Thus, we have derived the discrete-time equation:
> >
> > $$\\mathbf{x}\_t = f(\\mathbf{W}\_{\\text{in}} \\mathbf{I}\_t + \\mathbf{W} \\mathbf{x}\_{t-1} + \\mathbf{b}) + \\zeta\_t$$
> >
> >
> > We thought that $\\Delta t=1$ would simplify the presentation, however, it seems to be misleading the readers.
> > In our numerical experiments, we used a $\\Delta t<1$.
> > We will update our manuscript to include $\\Delta t$ for clarity.
> >
> >
> >
> > ### Integration scheme
> > Numerical integration of a stochastic differential equation is an extensive field by itself [3a].
> > We chose to use the simplest Euler-Maruyama discretization form because this leads to the standard RNN form.
> > Although it is generally inferior to other methods in terms of efficiency, systems with a fast-slow decomposition are stiff which presents additional challenges in their solution.
> >
> > Computational neuroscientists often train RNNs as models of neural computation [4a,5a]
> > and interpret them as dynamical systems.
> > Our experiments connect to existing literature.
> > In future studies, it would be interesting to perform experiments with Neural SDEs [6a].

---

> > > ### Comment · Reviewer_xmvZ · 2024-08-13
> > >
> > > I appreciate the authors' comprehensive responses. Assuming that the authors will make all mentioned changes to the revised version of the manuscript, I have increased my score to 6.

---

> ### Comment · Reviewer_xmvZ · 2024-08-12
>
> Dear Authors,
>
> Due to your new experiments and the fact that most of your responses were convincing to me, I will wait until the end of the discussion period. I will raise my score to 6 if you can address my concerns regarding the bifurcation and stability analysis; otherwise, I will raise it to 5.
>
> Kind regards,
> Reviewer

---

> ### Author Response · Authors · 2024-08-12
> **References**
>
> [1a] Fenichel, N. (1979). Geometric singular perturbation theory for ordinary differential equations. Journal of differential equations, 31(1), 53-98.
>
> [2a] Kirchgraber, U., & Palmer, K. J. (1990). Geometry in the neighborhood of invariant manifolds of maps and flows and linearization. (No Title).
>
> [3a] https://docs.sciml.ai/DiffEqDocs/stable/solvers/sde_solve/
>
> [4a] Mante, V., Sussillo, D., Shenoy, K. V., & Newsome, W. T. (2013). Context-dependent computation by recurrent dynamics in prefrontal cortex. nature, 503(7474), 78-84.
>
> [5a] Sussillo, D., & Barak, O. (2013). Opening the black box: low-dimensional dynamics in high-dimensional recurrent neural networks. Neural computation, 25(3), 626-649.
>
> [6a] Tzen, B., & Raginsky, M. (2019). Neural stochastic differential equations: Deep latent gaussian models in the diffusion limit. arXiv preprint arXiv:1905.09883.

---

### Author Rebuttal · Authors · 2024-08-07

We are grateful and encouraged that the reviewers found our work novel and interesting. Reviewers remarked that it is "a novel contribution and an important result to bolster the continuous attractor hypothesis", "fresh look, novel, original, and interesting [and] superb theoretical motivation", that "the main thrust of the paper was very interesting and very novel" and "it should be applauded."
We feel many of your suggestions have led us to changes and additions that better position the paper.

To respond to the reviewer's comments, we have performed the following analysis:

 * quantify the fast-slow time scale separation on the manifold found in task-trained RNNs (Fig R1)

 * trained LSTM and GRU networks (Fig R2)

 * trained RNNs on a 2D task where the continuous attractor manifold is a torus (Fig R3)

# Generality of the theory
While most bifurcation analyses in theoretical neuroscience and machine learning are based on a particular parameterization (e.g., pairwise weight matrix), our theory applies to any differentiable dynamical system and to continuous piecewise smooth systems (with a global continuous attractor). Hence, the robustness of most continuous attractors are covered. The only necessary condition is the *normal hyperbolicity* as demonstrated via separation of eigenvalues (Fig R1).
## Architecture
We tested our theory with LSTMs and GRUs to support our claim about the universality on trained RNNs.
These networks form the same normally hyperbolic invariant slow ring manifold just like Vanilla RNNs (Fig R2C,D) and on this manifold we find fixed points (Fig R2A,B). This consistency of structure across different RNN architectures provides further validation of our theoretical framework.
## Simple systems
The analysis of theoretical models and numerical experiments is intended to illustrate the theory's practical applicability rather than to prove its generality.
We focused on low-dimensional systems because they are easier to visualize and are a better guide to developing intuition.
We include results on RNNs trained on a 2D task (a double angular velocity integration task) to further demonstrate our theory's relevance. In the trained RNNs (Fig R3A,B), we find a slow, attractive, invariant manifold in the shape of a torus with a point topology. Additionally, we find evidence supporting the relevance of the error bound in these trained RNNs (Fig R3C,D).
## Broader impact
Approximate continuous attractor theory applies to dynamical systems inferred from data, from task-trained neuralODEs and RNNs, finite size effects on theoretical models, and due to lack of expressive power parametrized dynamics.
We believe our theory opens up a new way of grouping similar dynamical systems for understanding the essential computation.

# Clarity
We acknowledge the reviewers' concerns regarding clarity and add details on the main topics highlighted in the reviews. If accepted, our final manuscript will be updated.
## Section 5.2
Section 5.2 outlines the conditions under which approximations to an analog working memory problem are near a continuous attractor. This section is crucial for clarifying when a situation like Proposition 1 would occur. These conditions are met for RNNs:

 * C1: This translates to the existence of a manifold in the neural activity space with the same topology as the memory content. We formalize the dependence as the output mapping being a locally trivial fibration over the output manifold.
 * C2: Persistence, as per the reverse of the Persistent Manifold Theorem, requires the flow on the manifold to be slow and bounded.
 * C3+C4: Non-positive Lyapunov exponents correspond to negative eigenvalues of $\nabla_zh$. Along with dynamics robustness (corresponding to the persistence of the manifold), this implies normal hyperbolicity. We have expanded on this correspondence by building on [1].

## Parameter dependence for the analysis
The threshold parameter for identifying invariant slow manifolds was chosen to reflect the bimodal distributions of speeds along the integrated trajectories.
The supplementary document (Fig R1) shows that the identified invariant manifold accurately reflects the fast-slow separation expected for a normally hyperbolic system, thereby validating our method's legitimacy.
The number of fixed points (NFPS) identified depends on the number of points sampled for the angular flow on the invariant ring, but converges to the true NFPS as the grid of initial points is refined.
# Limitations
We will add a separate **Limitations** subsection:

Although our theory is general, since the magnitude of perturbation is measured in uniform norm, for specific parameterizations, further analysis is needed. If the parameters are not flexible enough, the theory may not apply, for example, RNNs with "built-in" continuous attractors such as LSTM without a forget gate cannot be destroyed. However, in biological systems, this is highly unlikely at least at the network level.

Our empirical analysis requires the fast-slow decomposition around the manifold. Not all dynamical system solutions to the tasks that require analog memory have this property (hence sec 5.2). Solutions such as the quasi-periodic toroidal attractors or constant spirals represent challenges to the current framework of analysis in understanding the general phenomena of analog memory without continuous attractors.

Our numerical analysis relies on identifying a time scale separation from simulated trajectories. If the separation of time scales is too small, it may inadvertently identify parts of the state space that are only forward invariant (i.e., transient). However, this did not pose a problem in our analysis of the trained RNNs, which is unsurprising, as the separation is guaranteed by state noise robustness (due to injected state noise during training).

[1] Mané, R. (1978). Persistent manifolds are normally hyperbolic. Transactions of the American Mathematical Society, 246, 261-283.

---

### Decision · Program_Chairs · 2024-09-25

**Decision:**

Accept (poster)

**Comment:**

This work revives the neuroscience idea of using manifold attractors for storing continuously varying quantities in long memory of recurrent systems. The common critique with such systems is that they are structurally unstable, but anchored in solid and elegant theoretical work and topological considerations, the authors show this actually may not be so much of a problem on functionally relevant time scales. These are important observations that may give new impetus to a powerful concept in theoretical neuroscience and beyond. While some of the referees initially had some reservations about this work, all of them after a strong rebuttal unanimously support publication.

I would like to add, however, that the idea that the vicinity of a manifold attractor may still be good enough to support behaviorally relevant time scales is not entirely new [1,2]. This should be mentioned and discussed in the revision.

[1] https://www.frontiersin.org/journals/computational-neuroscience/articles/10.3389/fncom.2011.00040/full

[2] https://www.cell.com/cell-reports/fulltext/S2211-1247(22)00360-6